# MEDFAIR: Benchmarking Fairness for Medical Imaging

**Yongshuo Zong**[1]**, Yongxin Yang**[1]**, Timothy Hospedales**[1,2]
[1] School of Informatics, University of Edinburgh, [2] Samsung AI Centre, Cambridge
{yongshuo.zong, yongxin.yang, t.hospedales}@ed.ac.uk

## Abstract

A multitude of work has shown that machine learning-based medical diagnosis systems can be biased against certain subgroups of people. This has motivated a growing number of bias mitigation algorithms that aim to address fairness issues in machine learning. However, it is difficult to compare their effectiveness in medical imaging for two reasons. First, there is little consensus on the criteria to assess fairness. Second, existing bias mitigation algorithms are developed under different settings, *e.g.*, datasets, model selection strategies, backbones, and fairness metrics, making a direct comparison and evaluation based on existing results impossible. In this work, we introduce MEDFAIR, a framework to benchmark the fairness of machine learning models for medical imaging. MEDFAIR covers eleven algorithms from various categories, ten datasets from different imaging modalities, and three model selection criteria. Through extensive experiments, we find that the under-studied issue of model selection criterion can have a significant impact on fairness outcomes; while in contrast, state-of-the-art bias mitigation algorithms do not significantly improve fairness outcomes over empirical risk minimization (ERM) in both in-distribution and out-of-distribution settings. We evaluate fairness from various perspectives and make recommendations for different medical application scenarios that require different ethical principles. Our framework provides a reproducible and easy-to-use entry point for the development and evaluation of future bias mitigation algorithms in deep learning. Code is available at https://github.com/ys-zong/MEDFAIR.

## 1 Introduction

Machine learning-enabled automatic diagnosis with medical imaging is becoming a vital part of the current healthcare system (Lee et al., 2017). However, machine learning (ML) models have been found to demonstrate a systematic bias toward certain groups of people defined by race, gender, age, and even the health insurance type with worse performance (Obermeyer et al., 2019; Larrazabal et al., 2020; Spencer et al., 2013; Seyyed-Kalantari et al., 2021). The bias also exists in models trained from different types of medical data, such as chest X-rays (Seyyed-Kalantari et al., 2020), CT scans (Zhou et al., 2021), skin dermatology images (Kinyanjui et al., 2020), etc. A biased decision-making system is socially and ethically detrimental, especially in life-changing scenarios such as healthcare. This has motivated a growing body of work to understand bias and pursue fairness in the areas of machine learning and computer vision (Mehrabi et al., 2021; Louppe et al., 2017; Tartaglione et al., 2021; Wang et al., 2020).

Informally, given an observation input $x$ (*e.g.*, a skin dermatology image), a sensitive attribute $s$ (*e.g.*, male or female), and a target $y$ (*e.g.*, benign or malignant), the goal of a diagnosis model is to learn a meaningful mapping from $x$ to $y$. However, ML models may amplify the biases and confounding factors that already exist in the training data related to sensitive attribute $s$. For example, data imbalance (*e.g.*, over 90% individuals from UK Biobank (Sudlow et al., 2015) originate from European ancestries), attribute-class imbalance (*e.g.*, in age-related macular degeneration (AMD) datasets, subgroups of older people contain more pathology examples than that of younger people (Farsiu et al., 2014)), label noise (*e.g.*, Zhang et al. (2022) find that label noises in CheXpert dataset (Irvin et al., 2019) is much higher in some subgroups than the others), etc. Bias mitigation algorithms

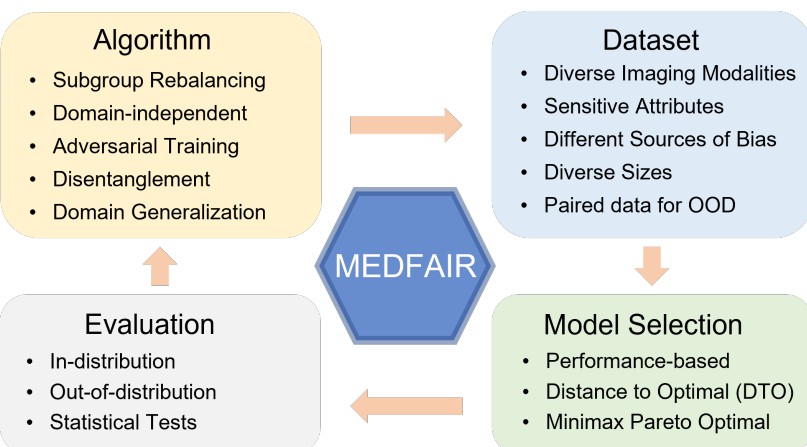

Figure 1: Components of MEDFAIR benchmark.

therefore aim to help diagnosis algorithms learn predictive models that are robust to confounding factors related to sensitive attribute $s$ (Mehrabi et al., 2021).

Given the importance of ensuring fairness in medical applications and the special characteristics of medical data, we argue that a systematic and rigorous benchmark is needed to evaluate the bias mitigation algorithms for medical imaging. However, a straightforward comparison of algorithmic fairness for medical imaging is difficult, as there is no consensus on a single metric for fairness of medical imaging models. Group fairness (Dwork et al., 2012; Verma & Rubin, 2018) is a popular and intuitive definition adopted by many debiasing algorithms, which optimises for equal performance among subgroups. However, this can lead to a trade-off of increasing fairness by decreasing the performance of the advantaged group, reducing overall utility substantially. Doing so may violate the ethical principles of *beneficence* and *non-maleficence* (Beauchamp, 2003), especially for some medical applications where all subgroups need to be protected. There are also other fairness definitions, including individual fairness (Dwork et al., 2012), minimax fairness (Diana et al., 2021), counterfactual fairness (Kusner et al., 2017), etc. It is thus important to consider which definition should be used for evaluations.

In addition to the use of differing evaluation metrics, different experimental designs used by existing studies prevent direct comparisons between algorithms based on the existing literature. Most obviously, each study tends to use different datasets to evaluate their debiasing algorithms, preventing direct comparisons of results. Furthermore, many bias mitigation studies focus on evaluating tabular data with low-capacity models (Madras et al., 2018; Zhao et al., 2019; Diana et al., 2021), and recent analysis has shown that their conclusions do not generalise to high-capacity deep networks used for the analysis of image data (Zietlow et al., 2022). A crucial but less obvious issue is the choice of model selection strategy for hyperparameter search and early stopping. Individual bias mitigation studies are divergent or vague in their model selection criteria, leading to inconsistent comparisons even if the same datasets are used. Finally, given the effort required to collect and annotate medical imaging data, models are usually deployed in a different domain than the domain used for data collection. (E.g., data collected at hospital A is used to train a model deployed at hospital B). While the maintenance of prediction quality across datasets has been well studied, it is unclear if fairness achieved within one dataset (in-distribution) holds under dataset shift (out-of-distribution).

In order to address these challenges, we provide the first comprehensive fairness benchmark for medical imaging – MEDFAIR. We conduct extensive experiments across eleven algorithms, ten datasets, four sensitive attributes, and three model selection strategies to assess bias mitigation algorithms in both in-distribution and out-of-distribution settings. We report multiple evaluation metrics and conduct rigorous statistical tests to find whether any of the algorithms is significantly better. Having trained over 7,000 models using 6,800 GPU-hours, we have the following observations:

- Bias widely exists in ERM models trained in different modalities, which is reflected in the predictive performance gap between different subgroups for multiple metrics.

- Model selection strategies can play an important role in improving the worst-case performance. Algorithms should be compared under the same model selection criteria.
- The state-of-the-art methods do not outperform the ERM with statistical significance in both in-distribution and out-of-distribution settings.

These results show the importance of a large benchmark suite such as MEDFAIR to evaluate progress in the field and to guide practical decisions about the selection of bias mitigation algorithms for deployment. MEDFAIR is released as a reproducible and easy-to-use codebase that all experiments in this study can be run with a single command. Detailed documentation is provided in order to allow researchers to extend and evaluate the fairness of their own algorithms and datasets, and we will also actively maintain the codebase to incorporate more algorithms, datasets, model selection strategies, etc. We hope our codebase can accelerate the development of bias mitigation algorithms and guide the deployment of ML models in clinical scenarios.

## 2 FAIRNESS IN MEDICINE

### 2.1 PROBLEM FORMULATION

We focus on evaluating the fairness of binary classification of medical images. Given an image, we predict its diagnosis label in a way that is not confounded by any sensitive attributes (age, sex, race, etc.) so that the trained model is fair and not biased towards a certain subgroup of people.

Formally, Let $D \in \{D_i\}_i^I$ be a set of domains, where $I$ is the total number of domains. A domain can represent a dataset collected from a particular imaging modality, hospital, population, etc. Consider a domain $D = (\mathcal{X}, \mathcal{Y}, \mathcal{S})$ to be a distribution where we have input sample $\mathbf{x} \in \mathbb{R}^d$ over input space $\mathcal{X}$, the corresponding binary label $y \in \{0, 1\}$ over label space $\mathcal{Y}$,[1] , and sensitive attributes $s \in \{0, 1, ..., m-1\}$ with $m$ classes over sensitive space $\mathcal{S}$. We train a model $h \in \mathcal{H}$ to output the prediction $\hat{y} \in \{0, 1\}$, *i.e.*, $h : \mathcal{X} \rightarrow \mathcal{Y}$, where $\mathcal{H}$ is the hypothesis class of models. Note that for each dataset $\mathcal{D}_i$, there may exist several sensitive attributes at the same time, *e.g.*, there are metadata of both patients' age and sex. We only consider one sensitive attribute at one time.

**In-distribution**    Given a domain $D_i$, assume the input samples $X_i$, their labels $Y_i$, and the sensitive attributes $S_i$ are identically and independently distributed (iid) from a joint probability distribution $P_i(X_i, Y_i, S_i)$. We define the evaluation where the training and testing on the same domain $D_i$ to be in-distribution, *i.e.*, the training and testing set are from the same distribution. We train models for each combination of *algorithms × datasets × sensitive attributes*.

**Out-of-distribution**    In clinical scenarios, due to the lack of training data, it is common to deploy a model trained in the original dataset to new hospitals/populations that have different data distributions. We define the training on one domain $D_i$ and testing on the other unseen domain $D_j$ to be out-of-distribution settings, where $D_i$ and $D_j$ may have different distribution $P_i(X_i, Y_i, S_i)$ and $P_j(X_j, Y_j, S_j)$. In this case, we assume domains $D_i$ and $D_j$ must have the same input space (*e.g.*, X-ray imaging), diagnosis labels, and sensitive attributes, but differ in their joint distributions due to collection from different locations or different imaging protocols. We evaluate if bias mitigation algorithms are robust to distribution shift by directly using the model selected from the in-distribution setting of the domain $D_i$ to test on the domain $D_j$.

### 2.2 FAIRNESS DEFINITION IN MEDICINE

Here we consider two most salient fairness definitions for healthcare, *i.e.*, group fairness and Max-Min fairness. We argue that one should focus on different fairness definitions depending on the specific clinical application.

**Group Fairness**    Metrics based on group fairness usually aim to achieve parity of predictive performance across protected subgroups. For resource allocation problems that can be considered a zero-sum game due to the limited resources, *e.g.*, prioritising which patients should be sent to a limited number of intensive care units (ICUs), it is important to consider group fairness to reduce the disparity among different subgroups (related discussions in Hellman (2008); Barocas & Selbst

---

[1]Our framework can be easily extended to non-binary classification.

(2016)). We measure the performance gap in diagnosis AUC between the advantaged and disadvantaged subgroups as an indicator of group fairness. This is in line with the "separability" criteria (Chen et al., 2021; Dwork et al., 2012) that algorithm scores should be conditionally independent of the sensitive attribute given the diagnostic label (*i.e.*, $\hat{Y} \perp S|Y$), which is also adopted by (Gardner et al., 2019; Fong et al., 2021). On the other hand, Zietlow et al. (2022) find that for high-capacity models in computer vision, this is typically achieved by worsening the performance of the advantaged group rather than improving the disadvantaged group, a phenomenon termed as *leveling down* in philosophy that has incurred numerous criticisms (Christiano & Braynen, 2008; Brown, 2003; Doran, 2001). Worse, practical implementations often lead to worsening the performance of both subgroups (Zietlow et al., 2022), making it *pareto inefficient* and comprehensively violating beneficence and non-maleficence principles (Beauchamp, 2003). Thus, we argue that group fairness alone is not sufficient to analyse the trade-off between fairness and utility.

**Max-Min Fairness**    It is another definition of fairness (Lahoti et al., 2020) following Rawlsian max-min fairness principle (Rawls, 2001), which is also studied as minimax group fairness (Diana et al., 2021) or minimax Pareto fairness (Martinez et al., 2020). Here, instead of seeking to equalize the error rates among subgroups, it treats the model that reduces the worst-case error rates as the fairer one. It may be a more appropriate definition than group fairness for some medical applications such as diagnosis, as it better satisfies the *beneficence* and *non-maleficence* principles (Beauchamp, 2003; Chen et al., 2018; Ustun et al., 2019), *i.e.*, do the best and do no harm. Formally, for a model $h$ in the hypothesis class $\mathcal{H}$, denote $U_s(h)$ to be a utility function for subgroup $s$. A model $h^*$ is considered to be Max-Min Fair if it maximizes (Max-) the utility of the worst-case (Min) group:

$$h^* = \underset{h \in \mathcal{H}}{\operatorname{argmax}} \ \min_{s \in S} U_s(h). \tag{1}$$

In practice, it is hard to quantify the maximum optimal utility, and therefore we treat a model $h_k$ to be fairer than the other model $h_t$ if

$$\min_{s \in S} U_s(h_k) > \min_{s \in S} U_s(h_t). \tag{2}$$

We measure both group fairness and Max-Min fairness to give a more comprehensive evaluation for fairness in medical applications.

## 3 MEDFAIR

We implement a reproducible and easy-to-use codebase MEDFAIR to benchmark fairness in machine learning algorithms for medical imaging. In our benchmark, we conduct large-scale experiments in ten datasets, eleven algorithms, up to three sensitive attributes for each dataset, and three model selection criteria, where all the experiments can be run with a single command. We provide source code and detailed documentation, allowing other researchers to reproduce the results and incorporate other datasets and algorithms easily.

### 3.1 DATASETS

Ten datasets are included in MEDFAIR: CheXpert (Irvin et al., 2019), MIMIC-CXR (Johnson et al., 2019), PAPILA (Kovalyk et al., 2022), HAM10000 (Tschandl et al., 2018), Fitzpatrick17k (Groh et al., 2021), OL3I (Chaves et al., 2021), COVID-CT-MD (Afshar et al., 2021), OCT (Farsiu et al., 2014), ADNI 1.5T, and ADNI 3T (Petersen et al., 2010), to evaluate the algorithms comprehensively, which are all publicly available to ensure the reproducibility. We consider five important aspects during dataset selection:

**Imaging modalities.**    We select datasets covering various 2D and 3D imaging modalities, including X-ray, fundus photography, computed tomography (CT), magnetic resonance imaging (MRI), spectral domain optical coherence tomography (SD-OCT), and skin dermatology images.

**Potential sources of bias.**    We involve datasets that may introduce bias from different sources, including label noise, data/class imbalance, spurious correlation, etc. Note that each dataset may contain more than one source of bias.

**Sensitive attributes.**    The selected datasets contain attributes that are commonly treated sensitively and may be biased in clinical practice, including age, sex, race, and skin type.

Table 1: Detailed statistics of the datasets. "# images/scans" listed here are the actual numbers used in this study after removing those missing sensitive attributes. For potential bias, LN, CI, DI, and SC represent label noise, class imbalance, data imbalance, and spurious correlation, respectively.

| Dataset | Modality | # Images | Sensitive Attribute | Bias Sources |
|---|---|---|---|---|
| CheXpert | Chest X-ray (2D) | 222,793 | Age, Sex, Race | LN, CI, DI |
| MIMIC-CXR | Chest X-ray (2D) | 370,955 | Age, Sex, Race | LN, CI, DI |
| PAPILA | Fundus Image (2D) | 420 | Age, Sex | DI, CI |
| HAM10000 | Skin Dermatology (2D) | 9,948 | Age, Sex | DI, CI |
| Fitzpatrick17k | Skin Dermatology (2D) | 16,012 | Skin type | LN, DI, CI |
| OL3I | Heart CT (2D) | 8139 | Age, Sex | DI, CI, SC |
| COVID-CT-MD | Lung CT (3D) | 308 | Age, Sex | DI, CI |
| OCT | SD-OCT (3D) | 384 | Age | DI, CI |
| ADNI 1.5T | Brain MRI (3D) | 550 | Age, Sex | SC |
| ADNI 3T | Brain MRI (3D) | 110 | Age, Sex | SC |

**Size of datasets.** As the sizes of medical imaging datasets are often limited by privacy issues, etc., it is important to inspect whether the fairness algorithms are effective with different sizes of datasets. The dataset sizes range from $420 \sim 370,955$ for 2D images and $110 \sim 550$ for 3D scans.

**Out-of-distribution evaluation.** We include two pairs of datasets with the same modality but collected from different locations or different imaging protocols for out-of-distribution evaluations. Specifically, we choose two 2D chest X-ray datasets *i.e.*, CheXpert and MIMIC-CXR, and two 3D brain MRI datasets *i.e.*, ADNI 1.5T and ADNI 3T.

Table 1 lists the basic datasets information, and more detailed statistics are provided in Appendix B.

## 3.2 ALGORITHMS

MEDFAIR incorporates 11 algorithms across 5 categories (related work in Appendix A): subgroup rebalancing, domain-independence, adversarial training, disentanglement, and domain generalization. We carefully re-implement the following algorithms based on the original official codes:

- **Baseline**
  - Empirical Risk Minimization (**ERM**) (Vapnik, 1999) minimizes the average error across the dataset without considering the sensitive attributes.
- **Subgroup Rebalancing**
  - **Resampling** method upsampled the minority groups so that all of the subgroups appear during training with equal chances.
- **Domain-independence**
  - Domain independent N-way classifier (**DomainInd**) (Wang et al., 2020) trains separate classifiers for different subgroups with a shared encoder.
- **Adversarial Training**
  - Learning Adversarially Fair and Transferable Representations (**LAFTR**) (Madras et al., 2018) de-biases the representation by minimizing the ability to recognize sensitive attributes.
  - Conditional learning of Fair representation (**CFair**) (Zhao et al., 2019) tries to enforce the balanced error rate and conditional alignment of representations during training.
  - Learning Not to Learn (**LNL**) (Kim et al., 2019) unlearns the bias information iteratively by minimizing the mutual information between feature representation and bias.
- **Disentanglement**
  - Entangle and Disentangle (**EnD**) (Tartaglione et al., 2021) disentangles confounders by inserting an "information bottleneck", while still passing the useful information.
  - Orthogonal Disentangled Representations (**ODR**) (Sarhan et al., 2020) disentangles the useful and sensitive representations by enforcing orthogonality constraints for independence.

- **Domain Generalization (DG)**
  - Group Distributionally Robust Optimization (**GroupDRO**) (Sagawa et al., 2019) minimizes the worst-case training loss with increased regularization.
  - Stochastic Weight Averaging Densely (**SWAD**) (Cha et al., 2021), a state-of-the-art method in DG, aims to find a robust flat minima by a dense stochastic weight sampling strategy.
  - Sharpness-Aware Minimization (**SAM**) (Foret et al., 2020) seeks parameters that lie in neighborhoods having uniformly low loss during optimization.

The hyper-parameter tuning strategy is described in Appendix B.2.2.

## 3.3 MODEL SELECTION

The trade-off between fairness and utility has been widely noted (Kleinberg et al., 2016; Zhang et al., 2022), making hyper-parameter selection criteria particularly difficult to define given the multi-objective nature of optimising for potentially conflicting fairness and utility. Previous work differs greatly in model selection. Some use conventional utility-based selection strategies, *e.g.*, overall validation loss, while others have no explicit specification. We provide a summary of model selection strategies across the literature in Table A1. To investigate the influence of model selection strategies on the final performance, we study three prevalent selection strategies in MEDFAIR.

**Overall Performance-based Selection**  This is one of the most basic and common strategies for model selection. It picks the model that has the smallest loss value or highest accuracy/AUC across the validation set of all sub-populations. However, this strategy tends to select the model with better performance in the majority group to achieve the best overall performance, leading to a potentially large performance gap among subgroups, which is illustrated in the red pentagon on the right side of Figure 2 (note that it is not necessarily Pareto optimal).

**Minimax Pareto Selection**  The concept of Pareto optimality was proposed by Mas-Colell et al. (1995) and utilized in fair machine learning to study the trade-off among subgroup accuracies (Martinez et al., 2020). Intuitively, for a model on the Pareto front, no group can achieve better performance without hurting the performance of other groups. In other words, it defines the set of best achievable trade-offs among subgroups (without introducing unnecessary harm). Based on this definition, we select the model that lies on the Pareto front and achieves the best worst-case AUC (the red star in the middle top of Figure 2). We present a formal definition of minimax Pareto selection in Appendix B.4.

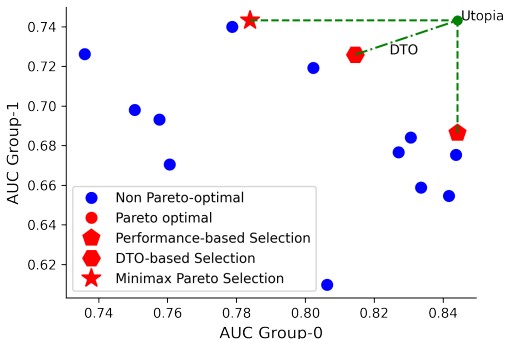

Figure 2: Illustration of three different model selection strategies. Each data point represents a different hyper-parameter combination for one algorithm, where the red points are the models lying on the Pareto front.

**DTO-based Selection**  Distance to optimal (DTO) (Han et al., 2021) is calculated by the normalized Euclidean distance between the performance of the current model and the optimal utopia point. Here, we construct the utopia point by taking the maximum AUC value of each subgroup among all models. The DTO strategy selects the model that has the smallest distance to the utopia point (the red hexagon in Figure 2).

## 3.4 EVALUATION AND IMPLEMENTATION

We apply the bias mitigation algorithms to medical image classification tasks and evaluate fairness based on the performance of different subgroups (sensitive attributes). The sensitive attributes are regarded as available during training (if needed). We consider two settings to evaluate fairness for medical imaging, *i.e.*, in-distribution and out-of-distribution.

**Evaluation Metrics**  We use the area under the receiver operating characteristic curve (AUC) as the major metric, which is a commonly used metric for medical binary classification. We evaluate

the algorithms from three aspects: (1) utility: overall AUC across all subgroups; (2) group fairness: AUC gap between the subgroups that have maximum AUC and minimum AUC; (3) Max-Min fairness: AUC of the worst-case group. Besides, we also report the values of binary cross entropy (BCE), expected calibration error (ECE), false positive rate (FPR), false negative rate (FNR), and true positive rate (TPR) at 80% true negative rate (TNR) of each subgroup, as well as the Equalized Odd (EqOdd). We provide detailed explanations of these metrics in Appendix B.3.

**Statistical Tests**    Prior work has empirically evaluated bias mitigation algorithms and occasionally claimed that some algorithm works well based on results from a couple of datasets. We note that to make a stronger conclusion that would be more useful to practitioners, e.g., '*algorithm A works better than B for medical imaging*' (i.e., $A$ is better general, rather than better for dataset $C$ specifically), one needs to evaluate performance across several datasets and perform significance tests that check for consistently good performance that can not be explained by overfitting to a single dataset. This is where the MEDFAIR benchmark suite comes in. To rigorously compare the relative performance of different algorithms, we perform the Friedman test (Friedman, 1937) followed by Nemenyi post-hoc test (Nemenyi, 1963) for both settings to identify if any of the algorithms is significantly better than the others, following the authoritative guide of Demšar (2006). We first calculate the relative ranks among all algorithms on each dataset and sensitive attribute separately, and then take the average ranks for the Nemenyi test if significance is detected by Friedman test. We consider a p-value lower than $0.05$ to be statistically significant. The testing results are visualized by Critical Difference (CD) diagrams (Demšar, 2006). In CD diagrams, methods that are connected by a horizontal line are in the same group, meaning they are not significantly different given the p-value, and methods that are in different groups (not connected by the same line) have statistically significant difference.

**Implementation Details**    We adopt 2D and 3D ResNet-18 backbone (He et al., 2016; Hara et al., 2018) for 2D and 3D datasets, respectively. The light backbone is used to avoid overfitting as there are datasets with small sizes, and also to remain consistent with the backbone used in the original literature (Kim et al., 2019; Wang et al., 2020; Tartaglione et al., 2021; Sarhan et al., 2020). Binary cross entropy loss is used as the major objective. To ensure the stability of randomness, for each experiment, we report the mean values and the standard deviations for three separate runs with three randomly selected seeds. Further implementation details for all datasets and algorithms can be found in Appendix B.

## 4    RESULTS

### 4.1    BIAS WIDELY EXISTS IN ML MODELS TRAINED IN DIFFERENT MODALITIES AND TASKS

Firstly, we train ERM on different datasets and sensitive attributes, and select models using the regular overall performance-based strategy. For each dataset and sensitive attribute, we calculate the maximum and minimum AUC and underdiagnosis rate among subgroups, where we use FNR for the malignant label and FPR for "No Finding" label as the underdiagnosis rate. As shown in Figure 3, most points are to the side of the equality line, showing that the performance gap widely exists. This confirms a problem that has been widely discussed (Seyyed-Kalantari et al., 2021) but, until now, has never been systematically quantified for deep learning across a comprehensive variety of modalities, diagnosis tasks, and sensitive attributes.

### 4.2    MODEL SELECTION SIGNIFICANTLY INFLUENCES WORST-CASE GROUP PERFORMANCE

We study the impact of model selection strategies on ERM using our three metrics of interest: The AUC of the worst-case group, the AUC gap, and overall AUC with ERM. We first conduct a hyper-parameter sweep for ERM while training on all the datasets, and then compute the metrics and the relative ranks of the three model selection strategies. The results, including statistical significance tests are summarised in Figure 4, and the raw data in Table A8. Each sub-plot corresponds to a metric of interest (worst-case AUC, AUC Gap, Overall AUC), and the average rank of each selection strategy (Pareto, DTO, Overall) is shown on a line. Selection strategies not connected by the bold bars have significantly different performances. The results show that for the worst-case AUC metric (left), the Pareto-optimal model selection strategy has the highest average rank of around 1.5, which is statistically significantly better than the overall AUC model selection strategy's average rank of around 2.5. Meanwhile, in terms of the overall AUC metric (right) the Pareto selection strategy is

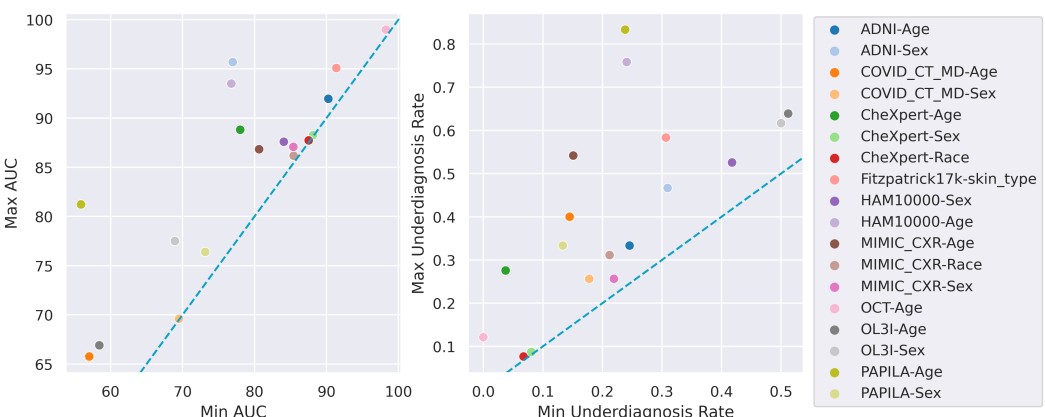

Figure 3: The AUC (left) and underdiagnosis rates (right) for the advantaged and disadvantaged subgroups across each dataset and sensitive attribute, when training with ERM. Most points are off the blue equality line, showing that bias widely exists in conventional ERM-trained models.

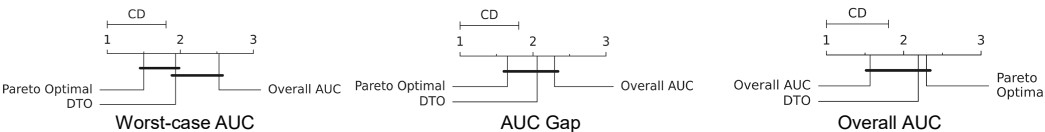

Figure 4: Influence of model selection strategies on ERM, illustrated in CD diagrams. The higher the rank of the AUC Gap, the smaller the gap.

not significantly worse than the overall model selection strategy. Thus, *even without any explicit bias mitigation algorithm, max-min fairness can be significantly improved simply by adopting the corresponding model selection strategy in place of the standard overall strategy - and this intervention need not impose a significant cost to the overall AUC.*

## 4.3 NO METHOD OUTPERFORMS ERM WITH STATISTICAL SIGNIFICANCE

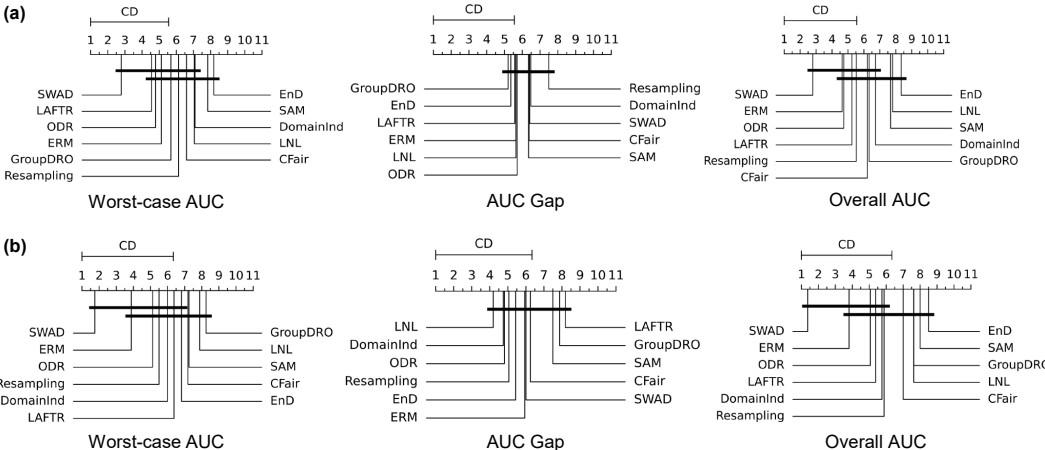

Figure 5: Performance of bias mitigation algorithms summarised across all datasets as average rank CD diagrams. (a) in-distribution, (b) out-of-distribution. SWAD is the highest ranked method for worst- and overall-AUC metrics, but it is still not significantly better than ERM.

We next ask whether any of the purpose-designed bias mitigation algorithms is significantly better than ERM, and which algorithm is best overall? To answer these questions, we evaluate the perfor-

mance of all methods using the Pareto model selection strategy. We report the Nemenyi post-hoc test results on worst-group AUC, AUC gap, and overall AUC in Figure 5 for in-distribution (top row) and out-of-distribution (bottom row) settings with raw data in Tables A9 and A10. For in-distribution, while there are some significant performance differences, no method outperforms ERM significantly for any metric: ERM is always in the highest-rank group of algorithms without significant differences. The conclusion is the same for the out-of-distribution testing, and some methods that rank higher than ERM in the in-distribution setting perform worse than ERM when deployed to an unseen domain, suggesting that preserving fairness across domain-shift is challenging.

It is worth noting there are some methods that consistently perform better, *e.g.*, SWAD ranks a clear first for the worst-case and overall AUC for both settings, and thus could be a promising method for promoting fairness. However, from a statistical significance point of view, SWAD is still not significantly better despite the fact that we use a much larger sample size (number of datasets) than most of the previous fairness studies. This shows that many studies do not use enough number of datasets to justify their desired claims. Our benchmark suite provides the largest collection of medical imaging datasets for fairness to date, and thus provides the best platform for future research to evaluate the efficacy of any method works in a rigorous statistical way.

## 5    DISCUSSION

**Source of bias**    There are multiple confounding effects that can lead to bias, rather than any single easy-to-isolate factor. As summarised in Table 1 and discussed further in Appendix A.4 and E, these include both measurable and unmeasurable factors spanning imbalance in subgroup size, imbalance in subgroup disease prevalence, difference in imaging protocols/time/location, spurious correlations, the intrinsic difference in difficulty of diagnosis for different subgroups, unintentional bias from the human labellers, etc. It is difficult or even impossible to disentangle all of these factors, making algorithms that specifically optimise for one particular factor to succeed.

**Failure of the bias mitigation algorithms**    Although most bias mitigation algorithms are not consistently effective across our benchmark suite, we are certainly not trying to disparage them. It is understandable because some are not originally designed for medical imaging, which contains characteristics distinct from those of natural images or tabular data, and more work may be necessary to design medical imaging specific solutions. More fundamentally, different algorithms may succeed if addressing solely the specific confounding factors for which they are designed to compensate, but fail when presented with other confounders or a mixture of multiple confounders. For example, resampling specifically targets data imbalance, while disentanglement focuses more on removing spurious correlations. But real datasets may also simultaneously contain other potential sources of bias such as label noise. This may explain why SWAD is the most consistently high-ranked algorithm, as it optimises a general notion of robustness without any specific assumption on confounders or sensitive attributes, and thus may be more broadly beneficial to different confounding factors.

**Relation of domain generalization and fairness**    The aim of domain generalization (DG) algorithms is to maintain stable performance on unseen sub-populations, while the fairness-promoting algorithms try to ensure that no known sub-populations are poorly treated. Despite this difference, they share the eventual goal — being robust to changes in distribution across different sub-populations. As shown in section 4, some domain generalization methods, such as SWAD, consistently improve the performance of *all* subgroups, and thus overall utility. However, we also notice that they may also enlarge the performance gap among subgroups. It introduces a question of whether a systematically better algorithm (*i.e.*, improving Max-Min fairness) is fairer if it increases the disparity (*i.e.*, not satisfying group fairness)? This question goes beyond machine learning and depends on application scenarios. We suggest that a relevant differentiator may be between diagnosis and zero-sum resource allocation problems where Max-Min and group fairness could be prioritised respectively.

**Are the evaluations enough for now?**    Although we have tried our best to include a diverse set of algorithms and datasets in our benchmark, it is certainly not exhaustive. There are methods to promote fairness from other perspectives, *e.g.*, self-supervised learning may be more robust (Liu et al., 2021; Azizi et al., 2022). Also, datasets from other medical data modalities (*e.g.*, cardiology, digital pathology) should be added. Beyond image classification, other important tasks in medical imaging, such as segmentation, regression, and detection, are underexplored. We will keep our codebase alive and actively incorporate more algorithms, datasets, and even other tasks in the future.

## REPRODUCIBILITY STATEMENT

We report the data preprocessing in Appendix B.1.2, hyper-parameter space in Appendix B.2.2. All of the datasets we use are publicly available, and we provide the download links in Table A6. Source code and documentation are available at https://ys-zong.github.io/MEDFAIR/. Running all the experiments required ∼ 0.77 NVIDIA A100-SXM-80GB GPU years.

## ACKNOWLEDGMENT

Yongshuo Zong is supported by the United Kingdom Research and Innovation (grant EP/S02431X/1), UKRI Centre for Doctoral Training in Biomedical AI at the University of Edinburgh, School of Informatics. For the purpose of open access, the author has applied a creative commons attribution (CC BY) licence to any author accepted manuscript version arising. We thank Dr. Maria Valdés Hernández for the discussion of data preprocessing. We acknowledge the data resources providers at Appendix F.

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

# A    RELATED WORK

In Appendix A, We present a broad review of the model selection strategies adopted by existing literature, current bias mitigation algorithms, existing fairness benchmarks, and domain generalization methods and their relationship to fairness.

## A.1    MODEL SELECTION STRATEGY IN FAIRNESS

Considering the trade-off between fairness and the utility, how to select the appropriate model among hyper-parameters remains an important problem. We broadly review and summarize the model selection strategies of recent work in Table A1. N/A means they do not explicitly specify their model selection strategies in their papers (although they may have implemented it in their open-source code). It can be seen that the model selection strategies differ greatly among existing literature, making a direct comparison infeasible. Hence, we investigate the influence of model selection strategies in our study.

Table A1: Summary of model selection strategies of fairness-aware methods and benchmarks.

| Reference | Paper Type | Selection Strategy | Dataset Type | Model Type |
|---|---|---|---|---|
| Locatello et al. (2019) | Benchmark | N/A | Images | Deep |
| Reddy et al. (2021) | Benchmark | Report best test results for each metric | Tabular / Images | Shallow |
| Zhang et al. (2022) | Benchmark | Best validation worst-group AUC | Images | Deep |
| Han et al. (2022) | Benchmark | Best validation loss / DTO | NLP | Deep |
| Friedler et al. (2019) | Benchmark | N/A | Tabular | Shallow |
| Wang et al. (2020) | Method | Best validation weighted mAP | Images | Deep |
| Madras et al. (2018) | Method | Report fairness/accuracy trade-off | Tabular | Shallow |
| Zhao et al. (2019) | Method | Report fairness/accuracy trade-off | Tabular | Shallow |
| Creager et al. (2019) | Method | Report fairness/accuracy trade-off | Tabular/Images | Shallow/Deep |
| Kim et al. (2019) | Method | N/A | Images | Deep |
| Tartaglione et al. (2021) | Method | Optimized on validation set | Images | Deep |
| Sarhan et al. (2020) | Method | Best validation loss | Images | Deep |
| Park et al. (2022) | Method | N/A | Images | Deep |
| Martinez et al. (2020) | Method | Best validation worst-group risk | Images/Tabular | Shallow/Deep |
| Idrissi et al. (2022) | Method | Best validation worst-group accuracy | Images | Deep |
| Lahoti et al. (2020) | Method | Best validation overall AUC | Tabular | Shallow |
| Lee et al. (2021) | Method | N/A | Images | Deep |

## A.2    BIAS MITIGATION ALGORITHMS

According to the stage when bias mitigation methods are introduced, they can be mainly classified into three categories, namely pre-processing, in-processing, and post-processing. The pre-processing methods aim to curate the data in the dataset to remove the potential bias before learning a model (Khodadadian et al., 2021), while the post-processing methods seek to adjust the predictions given a trained model according to the sensitive attributes (Pleiss et al., 2017). In this paper, we focus on benchmarking in-processing methods, which aim to mitigate the bias during the training of the models. Below, we review four popular categories of bias mitigation algorithms.

**Subgroup Rebalancing**    For imbalanced data, synthetic minority oversampling technique (SMOTE) (Chawla et al., 2002) is a classic resampling method that over-samples the minority class and under-samples the majority class. Recent work found that simply using data balancing can effectively improve the worst-case group accuracy (Idrissi et al., 2022).

**Domain-Independence**    Wang et al. (2020) develop a domain-independent training strategy that applies different classification heads to different subgroups. (Royer & Lampert, 2015) propose

classifier adaption strategies in the prediction time to reduce the error rates when the distribution of testing domain is different.

**Adversarial Learning**   There are two major categories of adversarial learning: (1) it plays a minimax game that the classification head tries to achieve the best classification performance while minimizing the ability of the discriminator to predict the sensitive attributes (Zhang et al., 2018; Kim et al., 2019). After training, the sensitive attributes in the representation are expected to be indistinct. (2) it enforces the fairness constraints (*e.g.*, group fairness) on the representation during trianing, and the representation is used for downstream tasks later (Xie et al., 2017; Madras et al., 2018; Zhao et al., 2019).

**Disentanglement**   Disentanglement methods (Tartaglione et al., 2021; Sarhan et al., 2020; Creager et al., 2019; Lee et al., 2021) isolates independent factors of variation into different and independent components of a representation vector. In other words, these methods disentangle the sensitive attributes and task-meaningful information in the representation level. Then, the classification tasks can be conducted on representation containing only task-specific representation so that the sensitive attributes and other unobserved factors will not make an impact.

In our benchmark, we select typical in-processing algorithms from these categories to provide a comprehensive comparison.

## A.3   Fairness Benchmarks

There are efforts in general machine learning communities to benchmark the performance of the fairness-aware algorithms, inspecting their effectiveness in different aspects. AIF360 (Bellamy et al., 2019) implements a wide range of fairness metrics and debiasing algorithms, which is available in both Python and R language. Fairlearn (Bird et al., 2020) also provides debiasing algorithms and fairness metrics to evaluate ML models. Friedler et al. (2019) benchmark a series of fairness-aware machine learning techniques. However, they only study the traditional machine learning models. For deep learning methods, Reddy et al. (2021) benchmark algorithms from the perspective of representation learning on tabular and synthetic datasets, and find they can successfully remove spurious correlation. Locatello et al. (2019) specifically benchmark the disentanglement fair representation learning methods on 3D shape datasets, suggesting disentanglement can be a useful property to encourage fairness.

In the area of computational medicine, machine learning models have been found to demonstrate a systematic bias toward a wide range of attributes, such as race, gender, age, and even the health insurance type (Obermeyer et al., 2019; Larrazabal et al., 2020; Spencer et al., 2013; Seyyed-Kalantari et al., 2021). The bias also exists in different types of medical data, such as chest X-ray (Seyyed-Kalantari et al., 2020), CT scans (Zhou et al., 2021), skin dermatology (Kinyanjui et al., 2020), health record (Obermeyer et al., 2019), etc.

The most relevant work to ours is Zhang et al. (2022), which compares a series of algorithms on chest X-ray images, and finds no method outperforms simple data balancing. However, it is unclear whether the conclusion can be generalized to other medical imaging modalities, and whether the selection of methods is comprehensive. In contrast, we benchmark a wider range of algorithms on different data modalities, study the ultimately more significant issue of model selection, and provide further analysis of the cause of the bias and the explanation of the effective algorithms. To the best of our knowledge, we are the first to provide a comprehensive benchmark for a wide range of algorithms and datasets, and a comprehensive analysis of different model selection criteria.

## A.4   Source of Bias

**Data imbalance across subgroups** is one of the most common sources of bias for medical imaging, as many biomedical datasets lack demographic diversity. For example, many datasets are developed with individuals originating from European ancestries, such as UK Biobank and The Cancer Genome Atlas (TCGA) (Sudlow et al., 2015; Liu et al., 2018). Also, more data samples in total will be from older people if the dataset is collected to study a specific disease that occurs more in older people, *e.g.* dataset for Age-related macular degeneration (AMD) (Farsiu et al., 2014). Overall,

when there are more samples in one subgroup than the other, it can be expected that the machine learning model may have different prediction accuracy for different subgroups.

**Class imbalance** can occur along with the data imbalance, where one subgroup has more samples of some specific classes while the other subgroup has more samples from other classes. For example, in the AMD dataset, subgroups of older people contain more pathology examples than subgroups of younger people. Also, the problem of class imbalance happens for rare diseases, which is generally due to genetic mutations that occur in a very limited number of people (Lee et al., 2020), *e.g.*, 10 in 1,000,000. So, the class imbalance is severe and there would never be enough data (even worldwide) for a balanced representation in the training set.

**Spurious correlation** can be learned by machine learning models undesirably from imaging devices. For example, a model can learn to classify skin diseases by looking at the markings placed by dermatologists near the lesions, instead of really learning the diseases (Winkler et al., 2019). It is also related to the data imbalance and class imbalance because the spurious correlations may occur more frequently in a subgroup with a smaller number of examples by simply remembering all the data points, *i.e.*, overfitting. Moreover, class imbalance itself may lead to a spurious correlation. For example, the model may try to use age-related features to predict pathology in a dataset where most of the older patients are unhealthy.

**Label noise** can also be a source of bias for medical imaging datasets. As the labeling of medical imaging datasets is labor-intensive and time-consuming, some large-scale datasets are labeled by automatic tools (Irvin et al., 2019; Johnson et al., 2019), which may not be precisely accurate and thus introduce noises. Zhang et al (Zhang et al., 2022) recruit a board-certified radiologist to relabel a subset of the chest X-ray dataset CheXpert, and find that the label noises are much higher in some subgroups than the others.

**Inherent characteristics** of the data of certain subgroups can lead to different performance for different subgroups even if the dataset is balanced, *i.e.*, the tasks in some subgroups are inherently difficult even for humans. For example, in skin dermatology images, the lesions are usually more difficult to recognize for darker skin than that for light skin due to the low contrast (Wen et al., 2021). Thus, even with balanced datasets, a trained ML model can still give lower accuracy to patients with darker skin. Considering this, other measures beyond algorithms should be adopted to promote fairness, such as collecting more representative samples and improving imaging devices.

In summary, the bias is usually not from a single source, and different sources of bias also usually correlate with each other, leading to the unfairness of the machine learning model.

### A.5 DOMAIN GENERALIZATION AND FAIRNESS

Domain generalization (DG) algorithms aim to maintain good performance on unseen sub-population, while fairness-promoting algorithms try to ensure that no known sub-population is poorly treated. Generally, though differing in detail, their eventual goal is the same — being robust to distribution changes across different sub-populations, which is also discussed in (Creager et al., 2020). Hence, in this work, we also explore fairness from the perspective of domain generalization.

One line of work for DG is to treat it as a robust optimization problem (Ben-Tal et al., 2009), where the goal is to try to minimize the worst-case loss for subgroups of the training set. Duchi et al. (2016) propose to minimize the worst-case loss for constructed distributional uncertainty sets with Distributionally Robust Optimization (DRO). GroupDRO (Sagawa et al., 2019) extends this idea by adding increased regularization to overparameterized networks and achieves good worst-case performance.

Another line of work focuses on finding flat minima in the loss landscape during optimization. As flat minima in the loss landscape is considered to be able to generalize better to different domains (Hochreiter & Schmidhuber, 1997), methods have been proposed to optimize for flatness (Izmailov et al., 2018; Cha et al., 2021; Keskar et al., 2016; Foret et al., 2020). Stochastic weight averaging (SWA) (Izmailov et al., 2018) finds flat minima by averaging model weights during parameters updating every $K$ epoch. Stochastic weight averaging densely (SWAD) adopts a similar weight ensemble strategy to SWA by sampling weights densely, *i.e.*, for every iteration. Also, SWAD searches the start and end iterations for averaging by considering the validation loss to avoid overfitting. Sharpness-aware minimization (SAM) (Foret et al., 2020) finds flat minima by seeking parameters

that lie in neighborhoods having uniformly low loss. We select three popular methods of them in our benchmark — GroupDRO, SAM, and SWAD.

# B   IMPLEMENTATION AND EVALUATION DETAILS

## B.1   DATA

### B.1.1   DATASET

We summarize the statistics of the subgroups and class labels in Table A2 to A5. The numbers with percentages out of the brackets are the percentage of the appearing prevalence, and the numbers in the brackets are the percentage of being unhealthy (class label). The used datasets are all publicly available, but we cannot directly include the downloading links in our benchmark. Thus we provide the access links in Table A6 provides.

Table A2: The statistics of the subgroups and class labels. The numbers with percentages out of the brackets are the percentage of the appearing prevalence, and the numbers in the brackets are the percentage of being unhealthy (class label).

|  | CheXpert | MIMIC-CXR | HAM10000 |
|---|---|---|---|
| **# Images** | 222,793 | 370,955 | 9,948 |
| **# Patients** | 64,428 | 222,793 | Unknown |
| **Male** | 59.34% (90.12%) | 52.16% (62.64%) | 54.28% (16.81%) |
| **Female** | 40.65% (89.78%) | 47.83% (57.11%) | 45.72% (11.65%) |
| **White** | 56.39% (90.02%) | 60.56% (65.2%) | Unknown |
| **non-White** | 43.61% (89.92%) | 39.44% (52.01%) | Unknown |
| **Age 0-20** | 0.8% (78.62%) | 0.75% (25.37%) | 2.42% (0.41%) |
| **Age 20-40** | 16.05% (79.56%) | 13.3% (36.66%) | 45.01% (5.71%) |
| **Age 40-60** | 31.07% (87.6%) | 15.2% (53.46%) | 6.98% (9.51%) |
| **Age 60-80** | 39.01% (93.0%) | 31.13% (67.24%) | 29.2% (23.92%) |
| **Age 80+** | 13.08% (96.3%) | 39.63% (76.65%) | 16.39% (32.13%) |

Table A3: The statistics of the subgroups and class labels. The numbers with percentages out of the brackets are the percentage of the appearing prevalence, and the numbers in the brackets are the percentage of being unhealthy (class label). Age group 0 and age group 1 range from 0-60 and 60+ except for the OCT dataset, whose age groups are from 55-75 and 75+.

|  | PAPILA | OCT | ADNI | ADNI3T |
|---|---|---|---|---|
| **# Images** | 420 | 384 | 550 | 182 |
| **# Patients** | 210 | 269 | 417 | 80 |
| **Male** | 34.76% (23.97%) | Unknown | 52.55% (44.98%) | 34.62% (42.86%) |
| **Female** | 65.24% (18.98%) | Unknown | 47.45% (43.30%) | 65.38% (39.50%) |
| **Age Group 0** | 42.38% (8.43%) | 40.89% (52.23%) | 49.82% (44.16%) | 46.70% (45.36%) |
| **Age Group 1** | 57.62% (29.75%) | 59.11% (14.54%) | 50.18% (44.20%) | 53.30% (35.29%) |

### B.1.2   DATA PREPROCESSING

Data splitting for experiments: unless otherwise specified, we randomly split the whole dataset into training/validation/testing sets with a proportion of 80/10/10 for 2D datasets and 70/10/20 for 3D datasets.

Table A4: The statistics of the Fitzpatrick17k dataset.

| Skin Type | I | II | III | IV | V | VI |
|---|---|---|---|---|---|---|
| % Images | 18.40% | 30.03% | 20.66% | 17.37% | 9.57% | 3.97% |
| % Malignant | 15.37% | 15.43% | 13.78% | 10.82% | 10.82% | 9.61% |

Table A5: The statistics of the subgroups and class labels. The numbers with percentages out of the brackets are the percentage of the appearing prevalence, and the numbers in the brackets are the percentage of being unhealthy (class label). Age group 0 and age group 1 range from 0-60 and 60+.

| | COVID-CT-MD | OL3I |
|---|---|---|
| # Images | 305 | 8139 |
| # Patients | 305 | 8139 |
| Male | 60.00% (59.02%) | 40.46% (5.53%) |
| Female | 40.00% (50.00%) | 59.54% (3.57%) |
| Age Group 0 | 73.11%(56.50%) | 67.91% (2.26%) |
| Age Group 1 | 26.89%(52.44%) | 32.09% (8.81%) |

**CheXpert**    We first incorporate ethnicity labels (Gichoya et al., 2022) and the original data (Links in Table A6), dropping those images without sensitive attribute labels. The "No Finding" label is used for training and testing. We use all of the available frontal and lateral images, and images of the same patient do not share across train/validation/test split.

**MIMIC-CXR**    The race data is available via MIMIC-IV (Johnson et al., 2020) dataset, which is also deposited in the PhysioNet database (Goldberger et al., 2000). We merge it to the original MIMIC-CXR metadata based on "subject ID". Other preprocessing steps are similar to that of CheXpert.

**PAPILA**    We exclude the "suspect" label class and use images with labels of glaucomatous and non-glaucomatous for binary classification tasks. The dataset contains right-eye and left-eye images of the same patient. We split the train/validation/test in a proportion of 70/10/20, and images of the same patient do not share across the train/validation/test split.

**HAM10000**    We split 7 diagnostic labels into binary labels, *i.e.*, benign and malignant following Maron et al. (2019). Benign contains basal cell carcinoma (bcc), benign keratosis-like lesions (solar lentigines / seborrheic keratoses and lichen-planus like keratoses, bkl), dermatofibroma (df), melanocytic nevi (nv), and vascular lesions (angiomas, angiokeratomas, pyogenic granulomas and hemorrhage, vasc). malignant contains Actinic keratoses and intraepithelial carcinoma / Bowen's disease (akiec), and melanoma (mel). We discard images whose sensitive attributes are not recorded, resulting in 9948 images in total.

**Fitzpatrick17k**    We split the three partition labels into binary labels *i.e.*, benign and malignant. We treat "non-neoplastic" and "benign" as the benign label, and use "malignant" as the malignant label. Fitzpatrick skin type labels are used as sensitive attributes.

**OL3I**    Opportunistic L3 computed tomography slices for Ischemic heart disease risk assessment (OL3I) dataset provides 8,139 axial computed tomography (CT) slices at the third lumbar vertebrae (L3) level of individuals. We design the task to predict whether the individual would be diagnosed with ischemic heart disease one year after the scan according to the labels provided (i.e. prognosis). Sex and age are treated as sensitive attributes.

Table A6: Access to the datasets.

| Dataset | Access |
|---|---|
| CheXpert | Original data: `https://stanfordmlgroup.github.io/competitions/chexpert/`
Demographic data: `https://stanfordaimi.azurewebsites.net/datasets/192ada7c-4d43-466e-b8bb-b81992bb80cf` |
| MIMIC-CXR | `https://physionet.org/content/mimic-cxr-jpg/2.0.0/` |
| PAPILA | `https://www.nature.com/articles/s41597-022-01388-1#Sec6` |
| HAM10000 | `https://dataverse.harvard.edu/dataset.xhtml?persistentId=doi:10.7910/DVN/DBW86T` |
| OCT | `https://people.duke.edu/~sf59/RPEDC_Ophth_2013_dataset.htm` |
| OL3I | `https://stanfordaimi.azurewebsites.net/datasets/3263e34a-252e-460f-8f63-d585a9bfecfc` |
| Fitzpatrick17k | `https://github.com/mattgroh/fitzpatrick17k` |
| COVID-CT-MD | `https://doi.org/10.6084/m9.figshare.12991592` |
| ADNI-1.5T
ADNI-3T | `https://ida.loni.usc.edu/login.jsp?project=ADNI` |

**OCT** The task is to predict if the patients have age-related macular degeneration (AMD). The dataset only contains people whose ages are over 55. Thus, we treat age as a binary attribute splitting from 75 years old, *i.e.*, group-0: 55-75, group-1: 75+. Scans are resized to $224 \times 224 \times 100$, and random cropping of the size of $192 \times 192 \times 96$ is used for training.

**COVID-CT-MD** We design the task to predict whether the patients are infected by COVID-19, *i.e.*, COVID-19 in one class, Community Acquired Pneumonia (CAP) and Normal in the other class. We resize each slice to a resolution of $256 \times 256$, and then take the central $80$ slices for training, as the crucial signal associated with infection of COVID and CAP is often present in the middle slices (Chaudhary et al., 2021). random cropping of size of $224 \times 224 \times 80$ is used for training. We split train/validation/test in a proportion of 70/10/20.

**ADNI-1.5T/ANDI-3T** We design the task to predict if patients have Alzheimer's disease (AD). MRI scans from ADNI have been preprocessed. We resize the height and width of scans of both 1.5T and 3T to the same size of $224 \times 224 \times 144$ to reduce the variance for cross-domain testing. Random cropping of the size of $196 \times 196 \times 128$ is used for training.

## B.2 TRAINING

### B.2.1 IMPLEMENTATION DETAILS

The experiments are conducted on a Scientific Linux release version 7.9 with one NVIDIA A100-SXM-80GB GPU. We trained over 7,000 models using $\sim 0.77$ GPU year. The implementation is based on Python 3.9 and PyTorch 1.10. We adapt the source code released by the original authors to our framework. We use ResNet-18 for 2D images and 3D ResNet-18 for 3D images as the backbone network for all experiments except otherwise specified.

For 2D datasets, we resize the images to the size of $256 \times 256$, and apply standard data augmentation strategies, *i.e.,* random cropping to $224 \times 224$, random horizontal flipping, and random rotation of up to 15 degrees. The backbone network is initialized using ImageNet (Deng et al., 2009) pretrained weights and images are normalized with the ImageNet mean and standard deviation values. For 3D datasets, we resize 3D images according to the original imaging characteristics as described in B.1.2.

The backbone network is initialized using kinetics (Carreira & Zisserman, 2017) pretrained weights and images are normalized with the kinetics mean and standard deviation values. Dataset-specific preprocessing and hyper-parameters can be found below.

### B.2.2 HYPER-PARAMETERS

To achieve the optimal performance of each algorithm for fair comparisons, we perform a Bayesian hyper-parameter optimization search for each algorithm and each combination of datasets and sensitive attributes using a machine learning platform Weights & Bias (Biewald, 2020). We use the batch size 1024 and 8 for 2D and 3D images respectively. SGD optimizer is used for all methods and we apply early stopping if the validation worst-case AUC does not improve for 5 epochs. The following hyper-parameter space is searched (20 runs for each method per *dataset × sensitive attribute*), where [ ] means the value range, and {} means the discrete values:

**ERM/Resampling/DomainInd**  Learning rate $lr \in [1e-3, 1e-5]$.

**LAFTR**  Learning rate $lr \in [1e-3, 1e-4]$. Adversarial coefficients $\eta \in [0.01, 5]$.

**CFair**  Learning rate $lr \in [1e-3, 1e-4]$. Adversarial coefficients $\eta \in [0.01, 5]$.

**LNL**  Learning rate $lr \in [1e-3, 1e-4]$. Adversarial coefficients $\eta \in [0.01, 5]$.

**EnD**  Learning rate $lr \in [1e-3, 1e-4]$. Entangling term coefficients $\alpha \in [0.01, 5]$. Disentangling term coefficients $\beta \in [0.01, 5]$.

**ODR**  Learning rate $lr \in [1e-3, 1e-4]$. Entropy Weight coefficients $\lambda_E \in [0.01, 5]$. Orthogonal-Disentangled loss coefficients $\lambda_{OD} \in [0.01, 5]$. KL divergence loss coefficients $\gamma_{OD} \in [0.01, 5]$. Entropy Gamma $\gamma_E \in [0.1, 5]$.

**GroupDRO**  Learning rate $lr \in [1e-3, 1e-4]$. Group adjustments $\eta \in [0.01, 5]$. Weight decay $L_2 \in \{1e-1, 1e-2, 1e-3, 1e-4, 1e-5\}$.

**SWAD**  Learning rate $lr \in [1e-3, 1e-4]$. Starting epoch $E_s \in \{3, 5, 7, 9\}$. Tolerance epochs $E_t \in \{3, 5, 7, 9\}$. Tolerance ratio $T_r \in [0.01, 0.3]$.

**SAM**  Learning rate $lr \in [1e-1, 1e-4]$. Neighborhood size $rho \in [0.01, 5]$.

Empirically, we find the best learning rate for all methods is around $1e-4$ except SAM, which requires a higher learning rate at around $[1e-1, 1e-3]$ depending on the datasets.

### B.2.3 METHODS

We summarize the benchmarked methods in Table A7. We show their categories, whether they can accept multiple sensitive attributes for training, and whether they require the information of sensitive attributes for training and testing.

### B.3 METRICS

We explain the metrics used in this study in detail below:

**AUC**  Area under the receiver operating characteristic curve (AUROC) is a standard metric to measure the performance of binary classification tasks, whose value is not affected by the imbalance of class labels. We use the name AUC in our text for simplicity. We measure the average AUC and the AUC of each subgroup. We pay particular attention to the AUC gap and the worst-case AUC to evaluate group fairness and max-min fairness.

**ECE**  Expected calibration error (ECE) (Guo et al., 2017; Nixon et al., 2019) is an indicator of group sufficiency (Castelnovo et al., 2022). A high ECE value may result in a different optimal best decision threshold.

**BCE**  Binary cross entropy (BCE) is the objective function we optimize for the classification task.

Table A7: A list of methods used in the benchmark. **SA** is short for Sensitive Attributes. Y and N represent Yes and No, respectively.

| Method | Category | Multiple SA | SA for Training | SA for Testing |
|---|---|---|---|---|
| ERM (Vapnik, 1999) | Baseline | Y | N | N |
| Resampling (Idrissi et al., 2022) | Subgroup Rebalancing | Y | Y | N |
| DomainInd (Wang et al., 2020) | Domain-Independence | Y | Y | N |
| LAFTR (Madras et al., 2018) | Adversarial | N | Y | N |
| CFair (Zhao et al., 2019) | Adversarial | N | Y | N |
| LNL (Kim et al., 2019) | Adversarial | Y | Y | N |
| EnD (Tartaglione et al., 2021) | Disentanglement | Y | Y | N |
| ODR (Sarhan et al., 2020) | Disentanglement | Y | Y | N |
| GroupDRO (Sagawa et al., 2019) | Domain Generalization | Y | Y | N |
| SWAD (Cha et al., 2021) | Domain Generalization | Y | N | N |
| SAM (Foret et al., 2020) | Domain Generalization | Y | N | N |

**TPR, FPR, FNR, FPR** Define true positive (TP) and true negative (TN) are values that are actually positive (negative) and correctly predicted positive (negative). Define false positive (FP) and false negative (FN) are values that are actually negative (positive) but wrongly predicted positive (negative). Then, we define true positive rate (TPR), true negative rate (TNR), false positive rate (FPR), and false negative rate (FNR) as

$$TPR = \frac{TP}{TP + FN} = 1 - FNR \tag{3}$$

$$TNR = \frac{TN}{TN + FP} = 1 - FPR. \tag{4}$$

We report the overall FPR, FNR, and their values for each subgroup. The threshold is selected based on the minimum F1 score following (Seyyed-Kalantari et al., 2021). We also report the value of TPR at 80% TNR, which indicates the true positive rate of a given desirable true negative rate.

**EqOdd** Equalized Odds is a widely used group fairness metric that the true positive and false positive rates should be equalized across subgroups. Denote the input, label, and sensitive attribute as $x, y, s$, the prediction and the output probability as $\hat{y}, p$. Following (Reddy et al., 2021), we define Equality of Opportunity w.r.t. $y = 0$ and 1, *i.e.,* EqOpp0 and EqOpp1, as

$$\text{EqOpp0} = 1 - |p(\hat{y} = 1|y = 0, s = \text{group-0}) - p(\hat{y} = 1|y = 0, s = \text{group-1})|, \tag{5}$$

$$\text{EqOpp1} = 1 - |p(\hat{y} = 1|y = 1, s = \text{group-0}) - p(\hat{y} = 1|y = 1, s = \text{group-1})|. \tag{6}$$

Then, we have EqOdd $= 0.5 \times$ (EqOpp0 + EqOpp1).

### B.4 MINIMAX PARETO SELECTION

Following (Martinez et al., 2020) and the dominance definition (Miettinen, 2008), we give a formal definition of the Pareto optimal regarding AUC.

**Dominant vector** A vector $\mathbf{t}' \in \mathbb{R}^k$ is said to dominate $t \in \mathbb{R}^k$ if $t'_i \geq t_i, \forall i = 1, ..., k$ and $\exists j : t'_j > t_j$ (namely strictly inequality on at least one component), denoted as $t' \succ t$.

**Dominant Classifier** Given a set of group-specific metrics function $T(h)$, *i.e.* AUC in our case, a model $h^{'}$ is said to dominate $h^{''}$ if $T(h^{'}) \succ T(h^{''})$, denote $h^{'} \succ h^{''}$. Likewise, a model $h^{'} \succeq h^{''}$ if $T(h^{'}) \succeq T(h^{''})$.

**Pareto Optimality** Given a set of models $H$, and a set of group-specific metrics function $T(h)$, a Pareto front model is $P_{S,H} = \{h \in H : \nexists h^{'} \in H | h^{'} \succ h\} = \{h \in H : h \succeq h^{'} \forall h^{'} \in H\}$. We call a model $h$ a Pareto optimal solution iff $h \in P_{A,H}$.

Finally, a model $h^*$ is a minimax Pareto fair classifier if it maximizes the worst-group AUC among all Pareto front models. For example, in Figure 2, each data point represents a different hyperparameter combination for one algorithm, where the red points are models lying on the Pareto front. As we can see from the figure, the Pareto optimal points cannot improve the AUC of group 0 without hurting the AUC of group 1, and vice versa, indicating the best trade-off between different groups. The worst-case group is group 1, as the best AUC achieved for group 1 is lower than the best AUC achieved for group 0. In this case, we select the model that achieves the best AUC of group 1 (red star point) – the disadvantaged group, to make the selection as fair as possible.

## C    ADDITIONAL RESULTS

Table A8 are the results of ERM under different model selection strategies behind Figure 4. Table A9 presents in-distribution results of the maximum and minimum AUC, and the gap between them for all datasets, as well as their average ranks. Table A10 presents out-of-distribution results. The highest maximum and minimum AUC values, and the smallest value of the performance gap are in bold.

We also report the BCE, ECE, TPR @80 TNR, FPR, FNR and EqOdd (for binary attributes) of each subgroup for a complete evaluation in the in-distribution setting in Table A11 to A13, and out-of-distribution setting in Table A14 and A15.

Table A8: Performance of different model selection strategies of ERM. The numbers separated by slashes are the mean values and ranks of overall AUC, the worst-case AUC, and the AUC gap, respectively. Mean values are reported.

| Dataset/Attr. | Overall Performance-based | DTO | Minimax Pareto Optimal |
|---|---|---|---|
| ADNI 1.5T / Age | 91.83 / 87.53 / 4.70 | 91.55 / 91.37 / 1.43 | 91.20 / 92.38 /1.04 |
| ADNI 1.5T / Sex | 82.77 / 76.97 / 18.72 | 81.57 / 75.33 / 22.73 | 81.36 / 78.52 / 24.90 |
| COVID-CT-MD / Age | 73.89 / 57.03 / 8.73 | 72.51 / 60.02 / 7.99 | 71.93 / 65.00 / 8.91 |
| COVID-CT-MD / Sex | 73.41 / 69.48 / 0.11 | 74.26 / 73.30 / 0.17 | 74.49 / 75.34 / 0.09 |
| CheXpert / Age | 89.50 / 78.01 / 10.80 | 89.62 / 77.92 / 10.92 | 88.09 / 78.36 / 10.33 |
| CheXpert / Sex | 89.01 / 88.12 / 0.13 | 88.43 / 88.01 / 0.08 | 88.90 / 88.70 / 0.02 |
| CheXpert / Race | 88.64 / 87.53 / 0.20 | 88.23 / 88.07 / 0.29 | 87.92 / 87.84 / 0.16 |
| Fitzpatrick17k / Skin Type | 90.38 / 91.37 / 3.72 | 90.71 / 90.42 / 4.05 | 91.51 / 90.67 / 3.70 |
| HAM10000 / Sex | 84.15 / 84.07 / 3.52 | 85.74 / 84.21 / 2.99 | 85.20 / 83.12 / 3.31 |
| HAM10000 / Age | 89.23 / 76.77 / 16.73 | 88.67 / 78.42 / 15.82 | 90.00 / 77.53 / 14.72 |
| MIMIC-CXR / Age | 86.13 / 80.64 / 6.19 | 85.98 / 81.97 / 5.78 | 86.40 / 81.06 / 5.32 |
| MIMIC-CXR / Race | 87.04 / 85.44 / 0.73 | 87.52 / 85.02 /1.26 | 86.26 / 85.52 / 0.85 |
| MIMIC-CXR / Sex | 87.10 / 85.40 / 2.37 | 86.39 / 85.49 / 1.32 | 86.45 / 85.62 / 1.41 |
| OCT / Age | 98.37 / 98.25 / 0.74 | 97.88 / 91.24 / 5.88 | 95.92 / 91.44 / 7.31 |
| PAPILA / Age | 66.27 / 55.88 / 25.34 | 64.07 / 56.31 / 23.48 | 64.98 / 53.22 / 24.90 |
| PAPILA / Sex | 80.97 / 73.14 / 3.25 | 80.04 / 74.39 /2.49 | 81.36 / 78.52 / 3.02 |
| OL3I / Sex | 74.25 / 67.41 / 9.27 | 74.01 / 68.55 / 9.13 | 73.87 / 68.93 / 8.57 |
| OL3I / Age | 69.13/ 57.94 / 9.27 | 67.99 / 58.22/ 8.93 | 66.58 / 58.43 / 8.48 |
| Avg. Rank | **1.56** / 2.53 / 2.29 | 2.19 / 1.93 / 2.05 | 2.29 / **1.49** / **1.65** |

Table A9: Results of the in-distribution evaluation.

| Dataset | Attr. | Metrics | ERM | Resample | DomainInd | LAFTR | CFair | LNL | EnD | ODR | GroupDRO | SWAD | SAM |
|---|---|---|---|---|---|---|---|---|---|---|---|---|---|
| HAM10000 | Sex | Overall | 85.20 | 86.94 | 86.63 | 86.48 | 86.24 | 86.58 | 86.92 | 86.36 | 87.03 | **91.44** | 89.90 |
| | | Min. | 83.12 | 85.04 | 84.93 | 86.00 | 83.56 | 86.03 | 84.96 | 83.54 | 85.92 | **91.14** | 89.55 |
| | | Gap | 3.31 | 3.09 | 2.79 | 0.48 | 5.12 | 0.93 | 3.29 | 4.59 | 1.80 | **0.44** | 0.73 |
| | Age | Overall | 90.00 | 90.10 | 89.03 | 89.26 | 89.06 | 88.78 | 86.92 | **90.23** | 89.32 | 89.61 | **90.23** |
| | | Min. | 77.53 | 82.73 | 78.23 | 80.03 | 82.95 | 81.18 | 71.21 | 79.56 | 80.00 | **86.66** | 80.54 |
| | | Gap | 14.72 | 9.68 | 13.95 | 11.59 | 9.31 | 11.43 | 21.96 | 13.61 | 13.01 | **5.72** | 12.77 |
| CheXpert | Age | Overall | 88.09 | 86.64 | 87.85 | 87.69 | 86.85 | 85.62 | 86.73 | 87.19 | 84.93 | **88.75** | 86.73 |
| | | Min. | 78.36 | 61.16 | 59.06 | 83.46 | 82.32 | 54.61 | 59.39 | 82.83 | 79.91 | **84.54** | 81.22 |
| | | Gap | 10.33 | 18.88 | 21.86 | 4.99 | 5.69 | 24.30 | 21.25 | **4.56** | 6.13 | 4.79 | 5.75 |
| | Sex | Overall | 88.09 | 87.72 | 88.13 | 87.77 | 87.97 | 86.87 | 86.54 | 87.98 | 86.27 | **88.79** | 88.09 |
| | | Min. | 88.07 | 87.51 | 88.11 | 87.64 | 88.01 | 86.78 | 86.48 | 87.96 | 86.24 | **88.72** | 87.99 |
| | | Gap | 0.02 | 0.48 | 0.03 | 0.20 | 0.13 | 0.20 | 0.50 | **0.01** | 0.06 | 0.13 | 0.15 |
| | Race | Overall | 87.92 | 87.76 | 88.01 | 88.01 | 87.77 | 86.65 | 86.84 | 87.82 | 86.30 | **88.75** | 87.75 |
| | | Min. | 87.84 | 87.66 | 87.88 | 87.98 | 87.73 | 86.57 | 86.82 | 87.80 | 85.99 | **88.65** | 87.64 |
| | | Gap | 0.16 | 0.16 | 0.23 | 0.04 | 0.06 | 0.12 | **0.01** | 0.03 | 0.65 | 0.21 | 0.23 |
| MIMIC-CXR | Age | Overall | 86.40 | 85.53 | 86.22 | 86.05 | 85.69 | 84.13 | 85.12 | 86.12 | 83.49 | **87.08** | 82.80 |
| | | Min. | 81.06 | 70.97 | 70.87 | 81.11 | 81.05 | 66.60 | 73.18 | 70.78 | 78.62 | **82.15** | 77.97 |
| | | Gap | 5.32 | 6.99 | 7.42 | 4.87 | 4.73 | 9.18 | 5.16 | 7.76 | 5.38 | 5.10 | **3.95** |
| | Sex | Overall | 86.45 | 86.21 | 86.31 | 86.28 | 86.36 | 85.18 | 85.97 | 86.42 | 85.89 | **87.05** | 83.65 |
| | | Min. | 85.62 | 85.31 | 85.53 | 85.36 | 85.55 | 84.22 | 85.35 | 85.64 | 84.96 | **86.23** | 82.69 |
| | | Gap | 1.41 | 1.53 | 1.35 | 1.60 | 1.41 | 1.70 | **1.19** | 1.34 | 1.61 | 1.39 | 1.75 |
| | Race | Overall | 86.26 | 86.07 | 86.24 | 86.36 | 86.20 | 85.04 | 85.57 | 86.26 | 85.65 | **87.10** | 83.06 |
| | | Min. | 85.52 | 85.31 | 85.52 | 85.67 | 85.53 | 84.27 | 84.95 | 85.58 | 84.90 | **86.38** | 82.54 |
| | | Gap | 0.85 | 0.88 | 0.82 | 0.75 | 0.70 | 0.87 | 0.53 | 0.69 | 0.80 | 0.88 | **0.12** |
| OCT | Age | Overall | 95.92 | 98.35 | 98.83 | 99.16 | 98.74 | 97.93 | **99.71** | 99.35 | 98.02 | 94.88 | 97.44 |
| | | Min. | 91.44 | 96.26 | 96.79 | 97.50 | 96.97 | 96.08 | **99.11** | 98.93 | 97.60 | 91.98 | 96.79 |
| | | Gap | 7.31 | 3.01 | 3.21 | 2.50 | 2.93 | 3.71 | 0.89 | 1.07 | **0.26** | 3.64 | 0.71 |
| ADNI 1.5T | Age | Overall | 91.20 | 92.51 | 84.89 | 84.47 | **94.07** | 87.08 | 77.61 | 88.97 | 83.96 | 85.98 | 66.54 |
| | | Min. | **92.38** | 90.79 | 89.21 | 84.13 | 88.89 | 87.57 | 71.43 | 83.17 | 72.70 | 81.90 | 64.33 |
| | | Gap | 1.04 | 3.65 | **0.56** | 4.76 | 8.19 | 4.81 | 14.10 | 9.52 | 17.21 | 7.28 | 7.42 |
| | Sex | Overall | 81.36 | 92.17 | 84.81 | 84.39 | 81.82 | 66.75 | 84.76 | **92.26** | 92.05 | 91.33 | 66.54 |
| | | Min. | 78.52 | 88.52 | 75.93 | 81.85 | 78.97 | 69.63 | 84.82 | 85.56 | **90.74** | 88.15 | 62.90 |
| | | Gap | 3.02 | 6.32 | 10.38 | 4.08 | 4.73 | 2.49 | **1.85** | 9.58 | 2.22 | 6.59 | 7.47 |
| PAPILA | Age | Overall | 64.98 | 74.07 | 61.50 | 77.78 | 77.10 | 69.36 | 75.42 | 75.87 | 75.65 | **82.27** | 72.62 |
| | | Min. | 53.22 | 56.02 | 47.34 | 59.94 | 60.78 | 56.58 | 54.90 | 56.58 | 58.26 | **66.39** | 56.30 |
| | | Gap | **24.90** | 43.98 | 34.95 | 39.02 | 31.93 | 31.96 | 45.10 | 43.42 | 41.74 | 33.61 | 36.41 |
| | Sex | Overall | 78.40 | 67.45 | 67.23 | 74.86 | 66.78 | **82.10** | 80.70 | 77.33 | 79.69 | 76.66 | 72.95 |
| | | Min. | 78.04 | 64.58 | 63.54 | 70.31 | 61.98 | **81.76** | 78.12 | 76.56 | 79.17 | 66.67 | 64.58 |
| | | Gap | **0.08** | 6.01 | 5.87 | 6.55 | 7.82 | 1.05 | 4.23 | 1.87 | 1.22 | 16.08 | 13.07 |
| Fitz17k | Skin Type | Overall | 91.51 | 90.00 | 87.88 | 90.04 | 89.69 | 90.55 | 85.82 | 89.46 | 91.98 | **93.31** | 90.24 |
| | | Min. | 90.67 | 88.33 | 86.53 | 86.51 | 86.25 | 86.64 | 80.29 | 85.20 | 89.97 | **90.94** | 87.93 |
| | | Gap | **3.70** | 6.24 | 7.05 | 8.00 | 6.74 | 6.88 | 11.46 | 6.69 | 5.71 | 6.75 | 5.06 |
| COVID-CT-MD | Age | Overall | 71.93 | 69.95 | 70.20 | 62.01 | 64.86 | 70.60 | 68.11 | 74.68 | 75.94 | **78.43** | 71.39 |
| | | Min. | 65.00 | 62.78 | 57.78 | 56.67 | 59.44 | 65.56 | 61.67 | 63.89 | 66.67 | **71.67** | 68.53 |
| | | Gap | 8.91 | 7.27 | 13.17 | 5.03 | 3.36 | 3.25 | 3.34 | 11.20 | 10.97 | 6.66 | **1.47** |
| | Sex | Overall | 74.49 | 71.50 | 72.22 | 73.99 | 71.14 | 69.41 | 67.39 | 73.48 | 67.82 | **76.44** | 71.50 |
| | | Min. | **75.34** | 66.24 | 72.91 | 72.59 | 73.93 | 67.52 | 64.97 | 73.02 | 67.09 | 74.92 | 70.90 |
| | | Gap | **0.09** | 13.03 | 1.45 | 8.39 | 1.52 | 5.28 | 8.75 | 3.48 | 3.64 | 7.99 | 6.88 |
| OL3I | Age | Overall | 66.58 | 67.76 | 58.96 | 62.12 | 68.74 | 64.43 | 52.43 | 67.25 | 62.63 | **72.73** | 52.03 |
| | | Min. | 58.43 | 52.58 | 54.24 | 55.97 | 51.02 | 58.51 | 49.49 | 55.41 | 57.83 | **61.59** | 46.65 |
| | | Gap | 24.90 | 24.30 | 4.79 | 12.00 | 27.04 | 8.12 | 10.36 | 16.75 | **1.87** | 11.91 | 15.59 |
| | Sex | Overall | 73.87 | **74.47** | 58.98 | 66.27 | 62.61 | 70.17 | 51.38 | 67.54 | 69.06 | 71.78 | 61.15 |
| | | Min. | **68.93** | 68.34 | 56.07 | 61.70 | 56.82 | 65.45 | 47.31 | 64.37 | 61.36 | 64.23 | 58.22 |
| | | Gap | 8.57 | 13.49 | 10.29 | 9.05 | 12.62 | 8.80 | 8.43 | 4.00 | 16.23 | 15.88 | **3.08** |
| Avg. Overall Rank | | | 4.63 | 5.52 | 6.72 | 5.25 | 6.22 | 7.80 | 8.33 | 4.74 | 6.32 | **2.80** | 7.67 |
| Avg. Min. Rank | | | 5.12 | 6.13 | 7.10 | 4.55 | 6.60 | 7.05 | 8.19 | 4.79 | 5.69 | **2.79** | 7.83 |
| Avg. Gap. Rank | | | 5.64 | 7.49 | 6.48 | 5.58 | 6.38 | 5.64 | **5.35** | 5.71 | 5.20 | 6.41 | 6.35 |

Table A10: Results of the out-of-distribution evaluation. In the dataset column, the dataset in the first row is the training domain, and the second row is the testing domain.

| Dataset | Attr. | Metrics | ERM | Resample | DomainInd | LAFTR | CFair | LNL | EnD | ODR | GroupDRO | SWAD | SAM |
|---|---|---|---|---|---|---|---|---|---|---|---|---|---|
| CheXpert MIMIC-CXR | Age | Overall | 83.05 | 80.71 | 79.85 | 83.10 | 82.44 | 80.80 | 79.87 | 79.75 | 79.32 | **83.56** | 82.26 |
| | | Min. | 76.50 | 75.09 | 72.43 | 76.06 | 76.11 | 74.83 | 74.05 | 74.28 | 70.35 | **76.66** | 75.91 |
| | | Gap | 6.92 | 5.15 | 7.01 | 7.13 | 6.25 | 5.24 | 5.97 | **5.02** | 8.67 | 7.00 | 6.01 |
| | Sex | Overall | 82.73 | 82.92 | 82.86 | 82.51 | 82.73 | 81.96 | 82.65 | 83.34 | 81.94 | **83.91** | 83.68 |
| | | Min. | 81.63 | 81.97 | 81.93 | 81.49 | 81.84 | 81.25 | 81.84 | 82.44 | 80.83 | **83.04** | 82.73 |
| | | Gap | 1.96 | 1.67 | 1.65 | 1.79 | 1.59 | **1.21** | 1.53 | 1.54 | 1.94 | 1.50 | 1.67 |
| | Race | Overall | 82.50 | 82.39 | 83.30 | 82.63 | 81.68 | 81.46 | 82.33 | 82.91 | 81.72 | **83.50** | 83.25 |
| | | Min. | 81.40 | 81.49 | 82.50 | 81.76 | 80.98 | 80.60 | 81.60 | 82.20 | 80.46 | **82.55** | 82.43 |
| | | Gap | 1.28 | 0.90 | 0.70 | 0.76 | **0.43** | 0.69 | 0.50 | 0.50 | 1.63 | 1.02 | 0.73 |
| MIMIC-CXR CheXpert | Age | Overall | 85.87 | 84.10 | 86.06 | 85.17 | 84.60 | 84.71 | 83.68 | 85.33 | 85.10 | **86.80** | 83.43 |
| | | Min. | 78.84 | 77.29 | 78.70 | 76.89 | 76.84 | 77.58 | **79.27** | 77.77 | 76.61 | 79.18 | 77.97 |
| | | Gap | 8.57 | 7.68 | 8.80 | 10.43 | 9.95 | 8.00 | 7.05 | 9.49 | 10.18 | 9.32 | **6.18** |
| | Sex | Overall | 86.26 | 85.67 | 85.42 | 86.04 | 85.77 | 85.47 | 85.44 | 85.77 | 85.26 | **86.95** | 84.89 |
| | | Min. | 85.72 | 85.34 | 85.09 | 85.56 | 85.49 | 85.16 | 84.96 | 85.25 | 84.51 | **86.56** | 84.29 |
| | | Gap | 0.90 | 0.58 | 0.55 | 0.86 | **0.50** | 0.69 | 0.99 | 0.89 | 1.26 | 0.66 | 1.01 |
| | Race | Overall | 85.91 | 85.64 | 85.69 | 85.92 | 84.99 | 84.86 | 82.94 | 84.62 | 85.90 | **86.87** | 83.78 |
| | | Min. | 85.85 | 85.55 | 85.68 | 85.76 | 84.62 | 84.77 | 81.94 | 84.46 | 85.82 | **86.77** | 83.61 |
| | | Gap | 0.13 | 0.16 | **0.02** | 0.29 | 0.67 | 0.16 | 1.82 | 0.29 | 0.12 | 0.20 | 0.34 |
| ADNI 1.5T ADNI 3T | Age | Overall | **89.52** | 87.46 | 84.55 | 79.89 | 78.04 | 89.30 | 76.06 | 86.65 | 80.19 | 88.16 | 52.92 |
| | | Min. | **91.22** | 89.03 | 85.06 | 80.80 | 75.29 | 87.45 | 75.63 | 87.99 | 82.54 | 88.36 | 48.86 |
| | | Gap | **0.18** | 0.52 | 6.22 | 2.44 | 9.27 | 6.76 | 3.78 | 0.52 | 0.91 | 1.31 | 11.63 |
| | Sex | Overall | 84.33 | 86.36 | 83.05 | 79.40 | 83.75 | 62.05 | 86.16 | **89.91** | 86.45 | 86.47 | 52.92 |
| | | Min. | 84.31 | 83.74 | 82.63 | 77.63 | 82.68 | 62.21 | 85.66 | **89.05** | 85.96 | 85.45 | 49.87 |
| | | Gap | 0.24 | 4.03 | **0.18** | 2.73 | 1.92 | 0.24 | 1.05 | 1.35 | 0.41 | 1.46 | 8.92 |
| Avg. Overall Rank | | | 3.81 | 5.88 | 5.75 | 5.38 | 7.0 | 7.62 | 8.5 | 5.06 | 7.62 | **1.38** | 8.0 |
| Avg. Min. Rank | | | 3.88 | 5.5 | 6.0 | 6.38 | 7.19 | 7.88 | 6.81 | 5.12 | 8.25 | **1.75** | 7.25 |
| Avg. Gap. Rank | | | 5.94 | 5.06 | 4.75 | 8.19 | 6.25 | **4.19** | 5.44 | 4.81 | 7.88 | 6.0 | 7.5 |

Table A11: Results of other metrics for HAM10000, CheXpert, and ADNI 1.5T dataset

| Dataset | Attr. | Metrics | Group | ERM | Resample | DomainInd | LAFTR | CFair | LNL | EnD | ODR | GroupDRO | SWAD | SAM |
|---|---|---|---|---|---|---|---|---|---|---|---|---|---|---|
| HAM10000 | Sex | BCE | Grp. 0 | 0.30±0.01 | 0.28±0.02 | 0.43±0.07 | 0.31±0.01 | 0.29±0.02 | 0.30±0.01 | 0.52±0.00 | 0.30±0.01 | 0.28±0.01 | 0.27±0.01 | 0.27±0.01 |
| | | | Grp. 1 | 0.28±0.01 | 0.28±0.04 | 0.42±0.02 | 0.27±0.04 | 0.29±0.02 | 0.27±0.01 | 0.51±0.00 | 0.31±0.02 | 0.26±0.01 | 0.23±0.02 | 0.23±0.01 |
| | | ECE | Grp. 0 | 0.03±0.01 | 0.03±0.01 | 0.10±0.01 | 0.04±0.01 | 0.05±0.04 | 0.04±0.01 | 0.26±0.01 | 0.05±0.01 | 0.03±0.01 | 0.05±0.01 | 0.05±0.01 |
| | | | Grp. 1 | 0.04±0.01 | 0.04±0.01 | 0.09±0.00 | 0.04±0.01 | 0.05±0.01 | 0.04±0.01 | 0.27±0.00 | 0.06±0.01 | 0.03±0.01 | 0.04±0.01 | 0.03±0.01 |
| | | TPR@80 TNR | Grp. 0 | 73.45±2.3 | 76.65±3.4 | 77.71±4.7 | 75.23±3.9 | 78.51±0.9 | 78.00±4.2 | 77.20±4.5 | 78.80±3.2 | 76.93±3.4 | 85.51±4.1 | 79.59±1.7 |
| | | | Grp. 1 | 71.23±4.4 | 70.31±14.2 | 71.02±5.5 | 72.30±7.4 | 69.91±7.9 | 73.21±2.5 | 69.37±5.0 | 70.71±1.3 | 74.70±6.1 | 87.00±3.5 | 81.10±3.6 |
| | | FPR | Grp. 0 | 9.68±4.5 | 13.74±3.8 | 10.88±3.5 | 10.36±1.6 | 9.91±1.0 | 12.24±5.1 | 17.04±4.4 | 10.21±3.8 | 10.28±3.7 | 7.43±2.5 | 8.78±3.7 |
| | | | Grp. 1 | 7.14±2.8 | 10.92±4.7 | 8.73±3.9 | 6.80±1.5 | 7.81±3.9 | 8.82±4.2 | 10.58±2.4 | 7.56±2.2 | 8.06±2.5 | 5.21±1.8 | 5.29±2.1 |
| | | FNR | Grp. 0 | 41.77±10.8 | 30.80±9.5 | 35.02±8.1 | 40.93±9.0 | 32.49±1.9 | 32.49±8.2 | 25.32±12.7 | 38.82±10.3 | 37.55±7.0 | 32.49±3.9 | 34.60±6.5 |
| | | | Grp. 1 | 52.56±7.3 | 44.23±18.3 | 52.56±13.5 | 44.23±7.7 | 58.97±11.1 | 45.52±12.8 | 45.51±6.2 | 52.56±10.6 | 39.74±2.9 | 41.03±6.2 | 44.87±7.8 |
| | | EqOdd | - | 93.13±4.0 | 90.89±5.8 | 90.12±4.5 | 91.87±0.8 | 85.20±5.4 | 91.78±1.9 | 86.67±5.4 | 91.80±3.2 | 96.06±1.5 | 94.62±0.8 | 93.12±1.8 |
| | Age | BCE | Grp. 0 | 0.17±0.02 | 0.19±0.03 | 0.65±0.29 | 0.16±0.04 | 0.18±0.01 | 0.24±0.04 | 0.48±0.03 | 0.14±0.01 | 0.25±0.08 | 0.16±0.01 | 0.18±0.02 |
| | | | Grp. 1 | 0.19±0.01 | 0.22±0.01 | 0.66±0.23 | 0.26±0.01 | 0.22±0.03 | 0.21±0.02 | 0.48±0.01 | 0.26±0.07 | 0.25±0.03 | 0.21±0.01 | 0.19±0.01 |
| | | | Grp. 2 | 0.40±0.04 | 0.47±0.02 | 1.76±0.78 | 0.46±0.03 | 0.42±0.06 | 0.39±0.03 | 0.54±0.01 | 0.42±0.02 | 0.76±0.29 | 0.36±0.03 | 0.37±0.01 |
| | | | Grp. 3 | 0.43±0.03 | 0.65±0.13 | 2.89±1.58 | 0.58±0.19 | 0.47±0.07 | 0.43±0.02 | 0.56±0.03 | 0.58±0.29 | 0.80±0.21 | 0.47±0.10 | 0.42±0.03 |
| | | ECE | Grp. 0 | 0.04±0.01 | 0.05±0.01 | 0.09±0.01 | 0.04±0.01 | 0.04±0.01 | 0.08±0.02 | 0.32±0.02 | 0.04±0.02 | 0.07±0.01 | 0.04±0.01 | 0.06±0.02 |
| | | | Grp. 1 | 0.03±0.00 | 0.04±0.00 | 0.08±0.01 | 0.06±0.01 | 0.05±0.02 | 0.05±0.01 | 0.29±0.01 | 0.05±0.02 | 0.06±0.01 | 0.04±0.01 | 0.03±0.00 |
| | | | Grp. 2 | 0.09±0.01 | 0.09±0.00 | 0.16±0.02 | 0.09±0.01 | 0.09±0.03 | 0.06±0.03 | 0.23±0.01 | 0.09±0.02 | 0.11±0.02 | 0.07±0.03 | 0.05±0.01 |
| | | | Grp. 3 | 0.10±0.02 | 0.18±0.05 | 0.28±0.05 | 0.16±0.03 | 0.12±0.04 | 0.13±0.01 | 0.26±0.00 | 0.14±0.05 | 0.20±0.04 | 0.13±0.04 | 0.13±0.03 |
| | | TPR@80 TNR | Grp. 0 | 76.60±10.5 | 65.70±13.8 | 63.70±7.5 | 81.09±3.4 | 57.43±10.1 | 66.68±3.0 | 46.79±7.5 | 78.11±6.5 | 70.86±11.6 | 69.67±8.6 | 78.15±3.1 |
| | | | Grp. 1 | 91.55±2.0 | 90.47±2.9 | 90.37±6.4 | 87.06±3.2 | 89.48±2.7 | 89.62±5.2 | 92.51±2.7 | 89.29±2.1 | 94.76±2.3 | 93.17±3.0 | 89.84±0.6 |
| | | | Grp. 2 | 73.86±4.9 | 78.09±1.2 | 76.87±1.3 | 76.40±3.5 | 69.35±2.8 | 73.01±3.1 | 70.92±1.4 | 78.91±2.9 | 76.95±1.9 | 77.81±2.5 | 73.48±2.2 |
| | | | Grp. 3 | 75.22±2.2 | 69.10±1.1 | 77.69±5.0 | 74.07±11.6 | 72.22±5.6 | 66.90±19.4 | 59.52±32.7 | 72.96±8.4 | 76.28±5.5 | 71.85±10.3 | 75.47±2.9 |
| | | FPR | Grp. 0 | 4.36±1.0 | 4.36±3.3 | 4.36±4.7 | 6.32±1.0 | 7.41±2.9 | 6.10±3.1 | 10.89±2.1 | 6.97±3.1 | 5.66±4.7 | 4.14±1.0 | 5.45±1.6 |
| | | | Grp. 1 | 5.37±0.8 | 5.02±2.2 | 3.12±2.5 | 6.84±2.0 | 5.89±2.5 | 5.20±2.6 | 6.40±0.7 | 6.06±2.3 | 5.20±3.1 | 5.20±0.7 | 5.36±0.2 |
| | | | Grp. 2 | 10.73±3.2 | 10.73±6.2 | 10.58±3.9 | 10.00±4.3 | 10.14±1.8 | 11.30±3.9 | 11.45±1.4 | 9.28±4.9 | 11.16±6.2 | 9.71±0.9 | 13.91±0.8 |
| | | | Grp. 3 | 18.79±4.2 | 23.03±6.4 | 23.03±8.4 | 17.58±9.0 | 18.79±5.2 | 16.97±11.0 | 25.45±5.5 | 21.82±4.8 | 18.79±10.7 | 17.58±5.8 | 24.24±4.6 |
| | | FNR | Grp. 0 | 75.83±7.2 | 54.17±7.2 | 70.83±28.9 | 29.17±19.1 | 66.67±7.2 | 50.00±12.5 | 58.33±7.2 | 29.17±14.4 | 50.00±12.5 | 54.17±7.2 | 33.33±7.2 |
| | | | Grp. 1 | 36.01±4.8 | 35.66±12.8 | 53.49±18.5 | 37.21±10.1 | 42.64±5.9 | 40.31±2.7 | 34.11±3.6 | 40.31±3.6 | 40.31±12.8 | 44.96±3.6 | 37.21±4.0 |
| | | | Grp. 2 | 38.71±4.8 | 33.87±9.8 | 32.26±4.3 | 38.17±10.5 | 42.48±5.7 | 38.17±5.2 | 34.41±0.9 | 33.34±8.1 | 30.64±11.3 | 31.72±4.1 | 34.95±3.4 |
| | | | Grp. 3 | 24.08±6.4 | 25.93±11.6 | 18.52±3.2 | 35.19±22.5 | 29.63±11.6 | 27.78±9.6 | 24.07±3.2 | 25.93±8.5 | 25.92±6.4 | 27.78±5.6 | 18.52±3.2 |
| CheXpert | Age | BCE | Grp. 0 | 0.37±0.00 | 0.52±0.01 | 0.74±0.14 | 0.40±0.01 | 0.41±0.05 | 0.45±0.04 | 0.44±0.08 | 0.40±0.04 | 0.81±0.10 | 0.35±0.01 | 0.43±0.01 |
| | | | Grp. 1 | 0.33±0.00 | 0.38±0.01 | 0.79±0.06 | 0.33±0.01 | 0.36±0.02 | 0.39±0.01 | 0.36±0.02 | 0.34±0.01 | 0.59±0.08 | 0.32±0.00 | 0.35±0.00 |
| | | | Grp. 2 | 0.26±0.00 | 0.27±0.00 | 0.70±0.07 | 0.26±0.00 | 0.27±0.01 | 0.30±0.01 | 0.28±0.00 | 0.27±0.00 | 0.57±0.08 | 0.25±0.00 | 0.27±0.00 |
| | | | Grp. 3 | 0.20±0.00 | 0.21±0.00 | 0.61±0.06 | 0.21±0.01 | 0.21±0.00 | 0.23±0.01 | 0.24±0.01 | 0.21±0.00 | 0.56±0.07 | 0.20±0.00 | 0.21±0.00 |
| | | | Grp. 4 | 0.12±0.00 | 0.13±0.00 | 0.39±0.02 | 0.13±0.00 | 0.14±0.01 | 0.15±0.01 | 0.16±0.00 | 0.13±0.01 | 0.48±0.07 | 0.12±0.00 | 0.13±0.00 |
| | | ECE | Grp. 0 | 0.07±0.01 | 0.12±0.02 | 0.16±0.02 | 0.11±0.02 | 0.09±0.03 | 0.12±0.04 | 0.08±0.01 | 0.09±0.01 | 0.34±0.07 | 0.07±0.01 | 0.09±0.01 |
| | | | Grp. 1 | 0.03±0.01 | 0.05±0.01 | 0.13±0.01 | 0.02±0.01 | 0.07±0.02 | 0.06±0.01 | 0.03±0.01 | 0.03±0.01 | 0.25±0.06 | 0.01±0.00 | 0.02±0.00 |
| | | | Grp. 2 | 0.02±0.00 | 0.02±0.00 | 0.09±0.01 | 0.02±0.00 | 0.03±0.01 | 0.04±0.02 | 0.03±0.01 | 0.02±0.01 | 0.28±0.05 | 0.01±0.00 | 0.01±0.00 |
| | | | Grp. 3 | 0.01±0.00 | 0.01±0.00 | 0.06±0.00 | 0.02±0.01 | 0.02±0.00 | 0.04±0.03 | 0.07±0.01 | 0.02±0.01 | 0.30±0.05 | 0.01±0.00 | 0.01±0.00 |
| | | | Grp. 4 | 0.01±0.00 | 0.01±0.01 | 0.03±0.00 | 0.02±0.01 | 0.02±0.01 | 0.02±0.01 | 0.06±0.00 | 0.02±0.00 | 0.29±0.05 | 0.01±0.00 | 0.01±0.00 |
| | | TPR@80 TNR | Grp. 0 | 76.37±3.1 | 61.24±7.8 | 73.52±2.6 | 62.81±7.2 | 67.20±6.4 | 62.76±2.2 | 69.67±4.1 | 66.28±8.9 | 61.80±8.7 | 79.14±1.5 | 61.70±3.9 |
| | | | Grp. 1 | 82.30±0.6 | 77.76±1.6 | 83.20±1.7 | 83.53±0.2 | 82.76±1.7 | 71.81±1.8 | 80.31±1.8 | 81.51±1.3 | 76.11±2.3 | 85.37±1.0 | 78.67±1.0 |
| | | | Grp. 2 | 83.26±1.5 | 80.65±0.9 | 82.37±0.7 | 83.00±0.9 | 83.38±1.5 | 74.68±2.8 | 80.75±0.5 | 80.54±1.1 | 78.05±1.2 | 85.29±0.4 | 80.03±0.4 |
| | | | Grp. 3 | 72.69±1.7 | 71.49±2.1 | 73.12±0.7 | 73.27±1.5 | 72.50±0.8 | 67.30±2.2 | 71.11±0.5 | 72.98±0.7 | 68.95±1.0 | 74.59±0.7 | 73.49±0.9 |
| | | | Grp. 4 | 75.18±3.0 | 69.80±3.2 | 71.50±1.0 | 73.73±2.0 | 67.72±1.5 | 66.73±1.9 | 69.13±0.3 | 73.02±1.8 | 65.09±2.0 | 77.76±0.9 | 69.93±0.4 |
| | | FPR | Grp. 0 | 27.56±7.2 | 23.40±3.9 | 33.66±3.3 | 26.92±9.5 | 19.87±1.1 | 37.50±12.6 | 19.23±2.5 | 29.49±5.8 | 27.24±2.0 | 25.00±1.9 | 44.87±1.1 |
| | | | Grp. 1 | 18.66±3.0 | 18.95±0.9 | 19.40±1.2 | 18.63±2.9 | 14.87±0.8 | 22.96±3.0 | 15.27±0.7 | 18.28±0.8 | 18.01±1.1 | 18.36±0.5 | 21.87±0.8 |
| | | | Grp. 2 | 13.05±1.6 | 13.47±0.9 | 11.93±1.3 | 11.95±0.9 | 11.49±0.7 | 15.30±1.0 | 13.53±1.8 | 11.51±1.1 | 13.12±1.0 | 11.51±0.6 | 12.09±0.4 |
| | | | Grp. 3 | 7.48±1.3 | 8.02±0.6 | 6.59±1.1 | 6.74±1.4 | 9.53±1.7 | 8.13±2.2 | 11.62±2.8 | 6.37±0.6 | 10.60±1.7 | 5.99±0.6 | 6.10±0.5 |
| | | | Grp. 4 | 3.74±0.4 | 3.15±0.3 | 2.17±0.4 | 2.26±0.8 | 5.45±2.0 | 3.15±2.8 | 6.18±2.7 | 2.36±0.6 | 6.40±1.9 | 1.72±0.3 | 1.54±0.2 |
| | | FNR | Grp. 0 | 16.66±5.8 | 31.11±5.1 | 11.11±5.1 | 23.34±5.8 | 34.45±6.9 | 18.89±21.2 | 33.33±13.3 | 15.55±3.9 | 27.78±6.9 | 14.44±1.9 | 17.78±1.9 |
| | | | Grp. 1 | 19.13±3.5 | 23.95±3.0 | 17.59±2.2 | 18.90±4.1 | 26.80±2.1 | 23.11±5.9 | 30.30±1.3 | 20.20±1.3 | 27.63±0.6 | 16.87±1.1 | 18.66±0.7 |
| | | | Grp. 2 | 32.67±4.1 | 30.49±2.9 | 31.63±3.0 | 31.74±2.5 | 32.19±2.2 | 33.04±1.7 | 31.52±3.3 | 34.81±3.1 | 33.89±2.1 | 27.72±1.2 | 34.29±1.8 |
| | | | Grp. 3 | 60.77±3.2 | 54.70±4.6 | 56.69±5.2 | 57.99±3.5 | 49.66±5.4 | 60.77±5.5 | 44.56±5.5 | 60.14±2.3 | 50.17±2.3 | 58.73±3.8 | 63.78±1.5 |
| | | | Grp. 4 | 77.78±4.4 | 70.96±3.4 | 74.49±2.4 | 76.01±4.3 | 64.90±9.1 | 78.03±15.3 | 58.34±7.7 | 76.01±5.6 | 67.17±11.4 | 74.75±6.1 | 83.84±1.9 |
| | Sex | BCE | Grp. 0 | 0.23±0.00 | 0.23±0.00 | 0.28±0.01 | 0.23±0.00 | 0.23±0.01 | 0.25±0.00 | 0.24±0.01 | 0.23±0.00 | 0.49±0.04 | 0.22±0.00 | 0.23±0.00 |
| | | | Grp. 1 | 0.22±0.00 | 0.22±0.00 | 0.27±0.00 | 0.22±0.00 | 0.22±0.00 | 0.23±0.00 | 0.23±0.00 | 0.22±0.00 | 0.43±0.03 | 0.21±0.00 | 0.22±0.00 |
| | | ECE | Grp. 0 | 0.01±0.01 | 0.01±0.01 | 0.05±0.03 | 0.01±0.00 | 0.01±0.00 | 0.02±0.01 | 0.03±0.00 | 0.01±0.01 | 0.23±0.03 | 0.01±0.00 | 0.01±0.00 |
| | | | Grp. 1 | 0.02±0.01 | 0.01±0.00 | 0.04±0.01 | 0.01±0.00 | 0.02±0.01 | 0.01±0.01 | 0.02±0.01 | 0.02±0.01 | 0.20±0.03 | 0.01±0.00 | 0.01±0.00 |
| | | TPR@80 TNR | Grp. 0 | 83.17±1.0 | 80.99±0.6 | 83.10±0.3 | 83.06±0.5 | 83.38±0.4 | 78.62±0.6 | 81.26±0.1 | 82.16±0.2 | 79.13±1.3 | 84.04±0.2 | 83.09±0.2 |
| | | | Grp. 1 | 81.35±0.4 | 81.21±1.4 | 81.13±0.5 | 81.16±1.3 | 81.35±1.5 | 77.99±1.1 | 79.73±0.8 | 80.78±0.4 | 78.96±0.7 | 82.58±0.4 | 81.35±0.3 |
| | | FPR | Grp. 0 | 7.65±1.1 | 8.41±0.6 | 9.10±0.8 | 9.68±0.7 | 8.05±0.8 | 11.63±0.9 | 10.17±3.6 | 7.55±0.3 | 12.46±0.3 | 8.45±0.6 | 8.25±1.0 |
| | | | Grp. 1 | 6.75±0.7 | 7.88±0.8 | 7.96±0.6 | 8.16±0.4 | 9.02±1.2 | 8.17±3.4 | 7.44±2.7 | 6.60±0.1 | 9.61±0.4 | 7.50±0.3 | 6.76±1.0 |
| | | FNR | Grp. 0 | 42.39±3.2 | 41.50±3.2 | 37.28±2.6 | 36.92±2.0 | 42.44±3.1 | 37.07±2.0 | 37.10±7.2 | 41.71±0.5 | 34.66±2.4 | 36.66±2.2 | 39.48±3.6 |
| | | | Grp. 1 | 40.91±2.2 | 37.21±1.3 | 36.85±1.5 | 37.72±0.8 | 35.02±2.6 | 43.69±9.2 | 41.93±6.9 | 41.35±1.4 | 36.33±1.9 | 36.84±1.0 | 40.58±3.0 |
| | | EqOdd | - | 98.54±0.5 | 97.59±1.4 | 98.43±0.4 | 98.50±0.7 | 95.80±2.8 | 93.48±5.8 | 96.21±1.0 | 99.05±0.2 | 97.74±0.7 | 99.08±0.4 | 98.71±0.5 |
| | Race | BCE | Grp. 0 | 0.22±0.00 | 0.22±0.00 | 0.27±0.01 | 0.22±0.00 | 0.22±0.00 | 0.23±0.01 | 0.23±0.00 | 0.22±0.01 | 0.43±0.06 | 0.22±0.00 | 0.22±0.00 |
| | | | Grp. 1 | 0.23±0.00 | 0.23±0.00 | 0.28±0.01 | 0.23±0.00 | 0.23±0.00 | 0.24±0.01 | 0.24±0.00 | 0.23±0.00 | 0.44±0.06 | 0.23±0.00 | 0.23±0.00 |
| | | ECE | Grp. 0 | 0.01±0.00 | 0.01±0.00 | 0.04±0.00 | 0.01±0.00 | 0.01±0.01 | 0.01±0.00 | 0.02±0.01 | 0.02±0.00 | 0.20±0.04 | 0.01±0.00 | 0.01±0.00 |
| | | | Grp. 1 | 0.01±0.00 | 0.02±0.00 | 0.04±0.00 | 0.01±0.00 | 0.01±0.00 | 0.01±0.00 | 0.02±0.01 | 0.02±0.01 | 0.20±0.04 | 0.01±0.00 | 0.01±0.00 |
| | | TPR@80 TNR | Grp. 0 | 81.57±0.6 | 81.06±0.8 | 81.69±0.3 | 82.06±0.3 | 81.32±0.8 | 79.72±0.7 | 80.58±0.5 | 81.54±1.1 | 78.01±0.7 | 82.11±0.2 | 81.31±0.5 |
| | | | Grp. 1 | 82.91±1.0 | 82.34±0.4 | 82.65±0.4 | 83.05±0.2 | 82.21±1.0 | 79.53±0.4 | 81.44±0.4 | 82.40±0.3 | 79.73±0.2 | 83.47±0.5 | 82.83±0.8 |
| | | FPR | Grp. 0 | 8.05±0.7 | 8.73±1.3 | 8.62±0.6 | 7.14±0.6 | 8.63±0.4 | 9.61±1.6 | 8.97±1.0 | 8.44±0.2 | 9.77±1.0 | 7.90±0.5 | 8.04±1.3 |
| | | | Grp. 1 | 8.70±0.5 | 9.18±1.3 | 9.12±0.7 | 7.67±0.6 | 9.51±0.4 | 10.57±1.7 | 9.54±0.9 | 8.84±0.3 | 10.71±1.0 | 8.58±0.5 | 8.62±1.5 |
| | | FNR | Grp. 0 | 39.33±1.8 | 38.97±4.5 | 37.92±2.0 | 41.49±1.8 | 38.45±1.7 | 38.09±3.4 | 37.29±2.3 | 38.86±0.6 | 38.22±2.7 | 38.95±1.5 | 40.10±4.3 |
| | | | Grp. 1 | 37.64±2.7 | 36.78±4.0 | 36.94±2.7 | 41.54±2.6 | 35.58±0.8 | 38.23±4.6 | 37.17±2.5 | 38.43±1.1 | 37.21±2.7 | 37.24±0.9 | 38.63±4.8 |
| | | EqOdd | - | 98.83±0.5 | 98.68±0.2 | 99.26±0.4 | 99.41±0.3 | 98.13±0.4 | 99.07±0.3 | 99.65±0.1 | 99.55±0.2 | 99.02±0.2 | 98.81±0.3 | 98.97±0.4 |
| ADNI 1.5T | Age | BCE | Grp. 0 | 0.52±0.28 | 0.37±0.09 | 0.92±0.55 | 0.75±0.09 | 0.72±0.10 | 0.73±0.19 | 0.66±0.15 | 0.48±0.10 | 0.55±0.18 | 0.41±0.03 | 0.78±0.14 |
| | | | Grp. 1 | 0.33±0.11 | 0.52±0.09 | 0.56±0.09 | 0.43±0.12 | 0.55±0.35 | 0.33±0.11 | 0.51±0.06 | 0.43±0.03 | 0.33±0.01 | 0.41±0.12 | 0.60±0.13 |
| | | ECE | Grp. 0 | 0.22±0.08 | 0.17±0.02 | 0.25±0.06 | 0.26±0.02 | 0.30±0.06 | 0.26±0.01 | 0.28±0.04 | 0.20±0.02 | 0.23±0.06 | 0.16±0.01 | 0.29±0.10 |
| | | | Grp. 1 | 0.14±0.04 | 0.21±0.05 | 0.14±0.05 | 0.18±0.03 | 0.21±0.18 | 0.16±0.04 | 0.21±0.00 | 0.18±0.04 | 0.14±0.01 | 0.17±0.01 | 0.25±0.15 |
| | | TPR@80 TNR | Grp. 0 | 84.56±11.3 | 82.98±4.0 | 72.28±4.3 | 68.07±5.8 | 68.77±11.6 | 69.82±4.7 | 70.53±3.6 | 86.14±7.1 | 75.79±6.6 | 86.99±3.4 | 28.77±23.2 |
| | | | Grp. 1 | 72.22±9.2 | 56.00±12.5 | 68.44±12.0 | 74.67±21.9 | 60.67±31.9 | 92.00±13.9 | 50.00±8.7 | 47.23±13.9 | 80.67±1.2 | 66.12±23.4 | 33.33±25.8 |
| | | FPR | Grp. 0 | 16.67±8.3 | 8.33±8.3 | 25.00±14.4 | 19.44±9.6 | 8.33±8.3 | 22.22±4.8 | 38.89±17.3 | 13.89±12.7 | 16.67±8.3 | 13.89±4.8 | 50.00±38.2 |
| | | | Grp. 1 | 20.63±14.5 | 22.23±13.7 | 28.57±20.8 | 14.29±14.3 | 28.57±14.3 | 11.11±12.0 | 44.44±27.9 | 20.63±9.9 | 11.11±2.8 | 22.22±7.3 | 55.56±23.5 |
| | | FNR | Grp. 0 | 15.79±9.1 | 24.56±6.1 | 17.54±16.1 | 24.56±12.2 | 36.84±9.1 | 21.05±10.5 | 14.03±8.0 | 19.30±12.2 | 24.56±8.0 | 15.79±5.3 | 15.79±13.9 |
| | | | Grp. 1 | 20.00±0.0 | 26.67±11.5 | 20.00±0.0 | 40.00±20.0 | 26.67±23.1 | 26.67±11.5 | 33.33±30.6 | 13.33±11.5 | 20.00±0.0 | 13.33±11.5 | 20.00±20.0 |
| | | EqOdd | - | 91.24±4.3 | 87.79±0.9 | 91.98±6.9 | 84.65±3.8 | 81.99±7.6 | 87.43±3.8 | 81.91±9.3 | 86.35±4.0 | 93.14±2.8 | 93.20±2.9 | 86.05±0.4 |
| | Sex | BCE | Grp. 0 | 0.47±0.14 | 0.52±0.14 | 1.01±0.25 | 0.53±0.04 | 0.40±0.14 | 0.66±0.11 | 0.51±0.06 | 0.53±0.04 | 0.38±0.09 | 0.34±0.09 | 0.68±0.04 |
| | | | Grp. 1 | 0.48±0.01 | 0.63±0.15 | 1.03±0.44 | 0.80±0.28 | 0.42±0.05 | 0.81±0.16 | 0.52±0.02 | 1.03±0.30 | 0.38±0.05 | 0.51±0.18 | 0.73±0.09 |
| | | ECE | Grp. 0 | 0.16±0.03 | 0.17±0.07 | 0.23±0.05 | 0.19±0.02 | 0.17±0.06 | 0.19±0.03 | 0.19±0.04 | 0.14±0.02 | 0.16±0.03 | 0.13±0.02 | 0.22±0.04 |
| | | | Grp. 1 | 0.21±0.01 | 0.30±0.08 | 0.26±0.04 | 0.30±0.07 | 0.23±0.04 | 0.34±0.07 | 0.20±0.04 | 0.25±0.04 | 0.19±0.02 | 0.19±0.06 | 0.33±0.06 |
| | | TPR@80 TNR | Grp. 0 | 74.52±15.4 | 65.71±16.5 | 61.75±3.7 | 81.83±6.9 | 85.71±11.2 | 52.86±14.3 | 76.43±10.5 | 77.05±7.4 | 87.78±9.8 | 88.57±10.8 | 37.26±24.2 |
| | | | Grp. 1 | 76.33±4.5 | 66.00±13.7 | 57.67±14.8 | 73.67±11.6 | 74.67±8.3 | 49.67±16.5 | 68.56±23.0 | 70.33±9.7 | 59.83±18.0 | 71.67±7.5 | 60.00±13.7 |
| | | FPR | Grp. 0 | 20.84±7.2 | 16.67±21.7 | 9.72±6.4 | 18.05±4.8 | 22.22±10.5 | 72.22±21.4 | 22.22±25.5 | 6.95±8.7 | 23.61±10.5 | 16.67±7.2 | 48.61±32.4 |
| | | | Grp. 1 | 22.22±11.1 | 29.63±33.9 | 14.81±17.0 | 11.11±11.1 | 14.81±6.4 | 40.74±23.1 | 11.11±11.1 | 11.11±11.1 | 14.81±17.0 | 18.52±23.1 | 66.67±22.2 |
| | | FNR | Grp. 0 | 21.43±24.7 | 30.95±4.1 | 40.48±10.9 | 23.81±4.1 | 4.76±4.1 | 9.52±10.9 | 16.67±18.0 | 21.43±12.4 | 14.29±7.1 | 14.28±12.4 | 26.19±20.6 |
| | | | Grp. 1 | 13.33±15.3 | 30.00±17.3 | 46.67±5.8 | 23.33±11.5 | 30.00±10.0 | 16.67±20.8 | 33.33±5.8 | 33.33±15.3 | 6.67±5.8 | 16.67±11.5 | 3.33±5.8 |
| | | EqOdd | - | 85.79±8.6 | 84.03±8.8 | 91.58±5.1 | 92.49±1.4 | 83.21±5.9 | 73.54±23.3 | 86.11±13.7 | 88.72±4.4 | 88.99±3.6 | 86.60±7.4 | 76.77±12.4 |

Table A12: Results of other metrics for MIMIC-CXR, OCT, and Fitzpatrick17k dataset

| Dataset | Attr. | Metrics | Group | ERM | Resample | DomainInd | LAFTR | CFair | LNL | EnD | ODR | GroupDRO | SWAD | SAM |
|---|---|---|---|---|---|---|---|---|---|---|---|---|---|---|
| MIMIC-CXR | Age | BCE | Grp. 0 | 0.40±0.02 | 0.47±0.05 | 0.86±0.04 | 0.42±0.03 | 0.42±0.02 | 0.49±0.02 | 0.44±0.01 | 0.46±0.02 | 0.48±0.06 | 0.41±0.03 | 0.62±0.12 |
| | | | Grp. 1 | 0.43±0.01 | 0.48±0.02 | 1.02±0.01 | 0.45±0.02 | 0.44±0.00 | 0.48±0.00 | 0.46±0.01 | 0.46±0.01 | 0.49±0.01 | 0.43±0.01 | 0.62±0.13 |
| | | | Grp. 2 | 0.46±0.00 | 0.48±0.00 | 1.04±0.04 | 0.47±0.01 | 0.47±0.01 | 0.50±0.01 | 0.47±0.00 | 0.48±0.01 | 0.52±0.01 | 0.46±0.01 | 0.71±0.18 |
| | | | Grp. 3 | 0.46±0.00 | 0.47±0.00 | 1.05±0.05 | 0.46±0.00 | 0.47±0.01 | 0.49±0.01 | 0.48±0.01 | 0.47±0.01 | 0.54±0.03 | 0.46±0.00 | 0.72±0.18 |
| | | | Grp. 4 | 0.44±0.00 | 0.44±0.00 | 1.05±0.07 | 0.43±0.00 | 0.46±0.01 | 0.46±0.01 | 0.47±0.01 | 0.46±0.00 | 0.55±0.06 | 0.44±0.01 | 0.69±0.16 |
| | | ECE | Grp. 0 | 0.06±0.02 | 0.08±0.03 | 0.14±0.00 | 0.06±0.04 | 0.05±0.02 | 0.05±0.02 | 0.09±0.01 | 0.08±0.01 | 0.08±0.03 | 0.06±0.02 | 0.16±0.07 |
| | | | Grp. 1 | 0.02±0.00 | 0.05±0.01 | 0.15±0.00 | 0.04±0.02 | 0.02±0.01 | 0.02±0.00 | 0.04±0.02 | 0.06±0.02 | 0.05±0.02 | 0.03±0.01 | 0.16±0.08 |
| | | | Grp. 2 | 0.02±0.01 | 0.03±0.01 | 0.17±0.00 | 0.05±0.02 | 0.04±0.01 | 0.04±0.01 | 0.04±0.02 | 0.05±0.02 | 0.08±0.04 | 0.03±0.01 | 0.22±0.10 |
| | | | Grp. 3 | 0.02±0.00 | 0.03±0.01 | 0.17±0.01 | 0.03±0.01 | 0.05±0.01 | 0.04±0.02 | 0.07±0.02 | 0.05±0.02 | 0.13±0.04 | 0.03±0.01 | 0.26±0.11 |
| | | | Grp. 4 | 0.03±0.00 | 0.02±0.01 | 0.16±0.01 | 0.03±0.01 | 0.06±0.01 | 0.04±0.01 | 0.08±0.01 | 0.06±0.01 | 0.15±0.04 | 0.03±0.01 | 0.28±0.10 |
| | | TPR@80 TNR | Grp. 0 | 78.33±2.7 | 71.08±6.5 | 79.55±1.2 | 77.91±2.4 | 71.63±9.2 | 64.81±5.6 | 74.51±3.3 | 70.61±3.8 | 71.67±4.6 | 76.86±2.8 | 68.68±6.9 |
| | | | Grp. 1 | 77.24±1.2 | 72.14±0.4 | 74.12±0.1 | 73.74±3.1 | 75.03±1.4 | 67.98±1.7 | 73.69±2.3 | 74.78±1.3 | 68.81±1.1 | 77.67±0.6 | 62.41±1.7 |
| | | | Grp. 2 | 77.01±0.3 | 74.49±0.8 | 76.39±0.3 | 76.49±0.5 | 76.95±0.1 | 71.53±0.5 | 76.34±0.3 | 76.33±0.8 | 72.23±1.1 | 77.76±0.5 | 67.32±1.8 |
| | | | Grp. 3 | 73.30±0.6 | 71.88±0.4 | 73.49±0.5 | 73.29±0.2 | 72.74±0.7 | 69.30±0.5 | 72.57±0.4 | 72.69±0.4 | 69.71±0.6 | 73.60±0.4 | 65.93±1.6 |
| | | | Grp. 4 | 65.85±0.3 | 65.99±1.7 | 66.39±1.9 | 66.33±0.2 | 66.53±0.9 | 61.45±1.6 | 66.44±0.2 | 64.73±0.3 | 62.34±0.8 | 66.64±0.4 | 59.96±1.1 |
| | | FPR | Grp. 0 | 54.17±3.5 | 53.79±3.3 | 47.35±2.9 | 53.41±5.2 | 45.46±3.0 | 62.88±6.5 | 43.94±2.9 | 45.83±6.8 | 36.37±5.2 | 45.45±2.0 | 69.32±4.1 |
| | | | Grp. 1 | 39.68±1.3 | 42.69±1.2 | 40.89±0.9 | 42.21±3.0 | 37.16±1.5 | 46.40±3.7 | 36.00±2.7 | 38.14±2.2 | 35.04±1.7 | 38.97±1.0 | 50.70±1.0 |
| | | | Grp. 2 | 30.75±1.3 | 33.21±0.3 | 31.20±1.5 | 31.38±1.8 | 29.20±0.9 | 35.70±1.9 | 30.35±2.4 | 30.81±0.9 | 32.77±1.4 | 29.66±2.0 | 37.66±1.4 |
| | | | Grp. 3 | 21.55±1.1 | 23.92±1.5 | 21.89±2.4 | 21.12±1.2 | 21.51±0.8 | 23.79±0.8 | 24.78±1.5 | 22.61±1.2 | 29.65±2.0 | 20.71±1.9 | 26.87±1.5 |
| | | | Grp. 4 | 15.10±1.5 | 16.77±0.7 | 15.14±2.4 | 14.14±2.3 | 16.76±1.2 | 14.05±0.3 | 19.71±0.8 | 16.53±1.4 | 26.22±2.7 | 14.67±1.6 | 18.38±2.7 |
| | | FNR | Grp. 0 | 2.08±0.5 | 2.43±1.7 | 3.47±1.8 | 3.65±1.0 | 5.04±0.3 | 5.90±2.0 | 6.08±1.3 | 6.60±0.3 | 12.33±2.6 | 5.04±1.2 | 5.73±1.4 |
| | | | Grp. 1 | 5.83±0.2 | 5.99±0.7 | 5.82±0.4 | 5.37±0.4 | 6.90±0.3 | 6.48±1.3 | 8.57±1.4 | 7.00±0.6 | 12.17±0.5 | 6.11±0.6 | 6.13±0.4 |
| | | | Grp. 2 | 12.20±0.7 | 11.61±0.2 | 12.14±0.8 | 12.51±1.1 | 13.63±0.6 | 12.55±0.8 | 13.26±1.6 | 12.99±0.3 | 13.98±0.7 | 12.64±1.3 | 13.77±1.2 |
| | | | Grp. 3 | 24.79±1.0 | 22.93±1.9 | 23.81±2.6 | 25.16±1.9 | 25.13±1.2 | 25.62±1.3 | 21.60±1.1 | 23.80±1.8 | 19.28±1.9 | 25.26±2.0 | 24.78±1.4 |
| | | | Grp. 4 | 41.81±1.7 | 39.34±2.7 | 42.44±4.9 | 43.49±4.1 | 38.87±1.3 | 49.03±0.7 | 34.01±1.0 | 41.00±2.2 | 30.26±3.9 | 42.86±3.4 | 41.87±3.0 |
| | Sex | BCE | Grp. 0 | 0.45±0.00 | 0.46±0.00 | 0.55±0.01 | 0.46±0.00 | 0.46±0.01 | 0.47±0.00 | 0.46±0.01 | 0.46±0.00 | 0.46±0.00 | 0.44±0.00 | 1.11±0.10 |
| | | | Grp. 1 | 0.45±0.00 | 0.45±0.00 | 0.51±0.02 | 0.45±0.00 | 0.45±0.00 | 0.47±0.00 | 0.46±0.01 | 0.45±0.00 | 0.46±0.00 | 0.44±0.00 | 1.01±0.09 |
| | | ECE | Grp. 0 | 0.02±0.01 | 0.03±0.01 | 0.10±0.01 | 0.02±0.01 | 0.04±0.02 | 0.03±0.01 | 0.04±0.03 | 0.03±0.01 | 0.02±0.01 | 0.02±0.00 | 0.40±0.05 |
| | | | Grp. 1 | 0.01±0.01 | 0.02±0.02 | 0.09±0.01 | 0.02±0.02 | 0.02±0.01 | 0.02±0.00 | 0.04±0.02 | 0.03±0.02 | 0.02±0.01 | 0.02±0.00 | 0.35±0.04 |
| | | TPR@80 TNR | Grp. 0 | 76.88±0.7 | 75.92±0.2 | 76.11±0.2 | 76.40±0.1 | 76.87±0.1 | 73.92±0.4 | 76.61±0.3 | 77.15±0.3 | 75.49±0.2 | 78.28±0.2 | 71.02±0.5 |
| | | | Grp. 1 | 78.32±0.3 | 77.55±0.6 | 78.21±0.6 | 77.89±0.2 | 77.65±0.5 | 75.45±0.4 | 77.37±0.5 | 78.31±0.7 | 77.35±0.2 | 79.23±0.4 | 72.82±0.9 |
| | | FPR | Grp. 0 | 21.93±0.6 | 23.30±0.4 | 23.66±0.1 | 23.13±1.3 | 23.03±0.9 | 25.29±1.8 | 24.87±1.8 | 22.46±0.2 | 24.04±1.2 | 21.52±0.7 | 27.95±1.1 |
| | | | Grp. 1 | 25.64±1.5 | 26.66±0.8 | 26.69±0.8 | 26.98±1.0 | 24.13±1.9 | 28.66±2.2 | 25.18±2.3 | 26.08±0.6 | 27.71±0.4 | 25.16±0.6 | 29.13±0.8 |
| | | FNR | Grp. 0 | 20.87±0.0 | 20.06±0.4 | 19.82±0.2 | 19.99±1.2 | 19.93±0.7 | 20.02±1.9 | 18.29±1.3 | 19.93±0.1 | 20.02±1.0 | 20.07±0.6 | 20.03±0.7 |
| | | | Grp. 1 | 15.80±1.4 | 15.12±0.7 | 14.88±0.8 | 14.66±0.6 | 17.51±1.7 | 15.67±1.6 | 16.81±1.9 | 15.48±0.5 | 14.53±0.2 | 15.29±0.7 | 16.94±0.2 |
| | | EqOdd | - | 95.60±1.3 | 95.85±0.6 | 96.00±0.8 | 95.41±1.9 | 98.24±1.1 | 95.81±3.1 | 99.08±0.6 | 95.97±0.6 | 95.42±1.0 | 95.79±0.1 | 97.86±0.4 |
| | Race | BCE | Grp. 0 | 0.45±0.00 | 0.45±0.00 | 0.54±0.01 | 0.45±0.00 | 0.44±0.00 | 0.46±0.00 | 0.46±0.01 | 0.45±0.01 | 0.46±0.00 | 0.44±0.00 | 0.99±0.19 |
| | | | Grp. 1 | 0.46±0.00 | 0.47±0.01 | 0.55±0.01 | 0.46±0.01 | 0.46±0.00 | 0.48±0.01 | 0.47±0.01 | 0.47±0.01 | 0.47±0.00 | 0.46±0.00 | 0.95±0.17 |
| | | ECE | Grp. 0 | 0.01±0.01 | 0.03±0.02 | 0.10±0.01 | 0.02±0.01 | 0.02±0.01 | 0.02±0.01 | 0.03±0.02 | 0.04±0.02 | 0.03±0.01 | 0.02±0.00 | 0.40±0.06 |
| | | | Grp. 1 | 0.02±0.01 | 0.03±0.02 | 0.10±0.00 | 0.02±0.02 | 0.01±0.00 | 0.03±0.02 | 0.02±0.01 | 0.04±0.02 | 0.03±0.01 | 0.02±0.00 | 0.34±0.05 |
| | | TPR@80 TNR | Grp. 0 | 76.19±0.4 | 75.81±0.3 | 76.00±0.6 | 76.50±0.4 | 76.53±0.4 | 73.54±0.8 | 75.76±0.4 | 75.97±0.7 | 75.29±0.3 | 76.84±0.2 | 69.48±0.6 |
| | | | Grp. 1 | 76.92±0.4 | 76.43±0.5 | 77.01±0.6 | 77.10±0.5 | 77.39±0.7 | 74.47±0.7 | 76.34±0.9 | 77.16±0.5 | 75.84±0.4 | 77.83±0.4 | 69.36±1.9 |
| | | FPR | Grp. 0 | 21.20±1.1 | 21.16±0.2 | 22.74±2.3 | 21.57±0.9 | 21.76±1.3 | 23.60±2.0 | 21.97±2.0 | 21.38±1.7 | 22.21±1.1 | 20.94±0.4 | 25.55±0.9 |
| | | | Grp. 1 | 31.14±1.3 | 31.38±0.2 | 32.20±2.5 | 31.07±0.3 | 30.33±0.9 | 33.65±1.4 | 29.60±0.8 | 30.91±2.0 | 32.24±0.7 | 30.40±0.5 | 36.74±1.1 |
| | | FNR | Grp. 0 | 22.22±1.2 | 22.64±0.5 | 20.77±1.9 | 21.51±0.7 | 21.42±1.2 | 22.14±1.7 | 21.99±1.7 | 22.20±1.5 | 22.03±0.8 | 21.88±0.4 | 23.66±0.8 |
| | | | Grp. 1 | 12.38±1.0 | 12.60±0.3 | 11.81±1.4 | 12.38±0.3 | 13.05±0.7 | 12.76±1.3 | 14.53±1.0 | 13.01±0.5 | 12.39±0.5 | 12.25±0.4 | 14.18±1.0 |
| | | EqOdd | - | 90.11±0.1 | 89.87±0.3 | 90.79±0.2 | 90.69±0.7 | 91.53±0.8 | 90.28±2.3 | 92.45±1.5 | 90.64±0.3 | 90.17±0.5 | 90.45±0.2 | 89.66±0.6 |
| OCT | Age | BCE | Grp. 0 | 0.19±0.02 | 0.25±0.14 | 0.33±0.07 | 0.19±0.06 | 0.23±0.05 | 0.31±0.03 | 0.55±0.00 | 0.22±0.06 | 0.21±0.01 | 0.33±0.04 | 6.10±1.31 |
| | | | Grp. 1 | 0.04±0.02 | 0.11±0.03 | 0.02±0.00 | 0.04±0.01 | 0.07±0.02 | 0.11±0.03 | 0.52±0.01 | 0.07±0.08 | 0.17±0.11 | 0.21±0.04 | 3.65±0.85 |
| | | ECE | Grp. 0 | 0.09±0.01 | 0.10±0.03 | 0.11±0.01 | 0.07±0.02 | 0.11±0.02 | 0.13±0.05 | 0.33±0.03 | 0.09±0.01 | 0.10±0.03 | 0.15±0.03 | 0.39±0.00 |
| | | | Grp. 1 | 0.03±0.01 | 0.07±0.02 | 0.02±0.00 | 0.03±0.01 | 0.05±0.00 | 0.08±0.02 | 0.37±0.03 | 0.04±0.03 | 0.09±0.02 | 0.06±0.02 | 0.24±0.00 |
| | | TPR@80 TNR | Grp. 0 | 87.10±4.0 | 80.45±3.9 | 82.45±5.7 | 83.12±13.2 | 87.11±4.2 | 84.63±2.6 | 73.85±30.5 | 40.78±36.0 | 91.56±0.3 | 76.42±5.1 | 87.73±4.3 |
| | | | Grp. 1 | 80.00±0.0 | 67.30±41.0 | 20.00±0.0 | 20.00±0.0 | 43.91±41.4 | 41.16±36.7 | 67.53±41.2 | 20.00±0.0 | 41.63±37.5 | 90.05±5.1 | 82.35±21.2 |
| | | FPR | Grp. 0 | 5.88±5.9 | 5.88±0.0 | 5.88±5.9 | 0.00±0.0 | 3.92±6.8 | 5.88±5.9 | 5.88±5.9 | 0.00±0.0 | 3.92±3.4 | 1.96±3.4 | 1.96±3.4 |
| | | | Grp. 1 | 1.04±1.8 | 1.04±1.8 | 0.00±0.0 | 0.00±0.0 | 1.04±1.8 | 1.04±1.8 | 1.04±1.8 | 3.13±5.4 | 2.08±3.6 | 5.21±1.8 | 1.04±1.8 |
| | | FNR | Grp. 0 | 12.12±5.2 | 15.15±10.5 | 18.18±9.1 | 18.18±9.1 | 18.18±9.1 | 21.21±10.5 | 15.15±5.2 | 12.12±5.2 | 12.12±5.2 | 27.27±9.1 | 15.15±5.2 |
| | | | Grp. 1 | 0.00±0.0 | 6.67±5.8 | 0.00±0.0 | 3.33±5.8 | 3.33±5.8 | 3.33±5.8 | 6.67±5.8 | 3.33±5.8 | 10.00±10.0 | 20.00±10.0 | 20.00±10.0 |
| | | EqOdd | - | 91.52±2.3 | 93.03±4.9 | 87.97±4.0 | 92.27±6.7 | 91.66±4.9 | 88.34±6.1 | 92.30±2.5 | 94.04±2.8 | 93.96±5.8 | 94.13±5.4 | 94.39±0.3 |
| Fitzpatrick17k | Skin type | BCE | Grp. 0 | 27.45±5.7 | 38.66±13.2 | 165.16±59.7 | 31.00±4.5 | 27.13±2.0 | 30.07±2.3 | 56.64±1.4 | 24.13±2.7 | 32.86±3.4 | 31.19±4.0 | 34.10±8.7 |
| | | | Grp. 1 | 31.60±3.3 | 41.80±13.4 | 119.70±30.1 | 38.82±9.9 | 29.94±1.7 | 30.05±2.5 | 58.18±1.6 | 34.27±2.4 | 34.42±5.8 | 32.79±1.9 | 36.78±8.9 |
| | | | Grp. 2 | 29.18±6.7 | 37.36±20.9 | 152.13±57.5 | 28.49±5.5 | 22.50±0.3 | 24.95±0.8 | 56.74±1.3 | 27.41±2.7 | 27.92±4.5 | 25.45±3.7 | 33.20±10.9 |
| | | | Grp. 3 | 24.10±6.3 | 27.79±21.6 | 49.76±18.0 | 19.75±4.6 | 21.64±0.8 | 23.93±4.8 | 55.35±1.6 | 21.16±2.4 | 18.68±4.1 | 12.34±0.5 | 25.30±9.8 |
| | | | Grp. 4 | 20.78±7.0 | 25.96±16.5 | 80.73±41.9 | 18.18±6.4 | 17.84±0.8 | 19.07±4.0 | 54.41±1.4 | 18.62±3.1 | 15.54±1.9 | 14.07±0.5 | 22.77±10.2 |
| | | | Grp. 5 | 23.21±7.7 | 31.20±14.2 | 88.83±80.4 | 20.11±6.6 | 15.48±0.8 | 12.32±2.3 | 54.04±1.3 | 15.44±4.1 | 20.07±1.8 | 13.39±4.3 | 19.87±10.1 |
| | | ECE | Grp. 0 | 6.52±3.6 | 11.70±10.1 | 17.42±8.7 | 6.19±1.7 | 4.35±1.2 | 5.71±0.4 | 31.46±0.8 | 3.91±0.9 | 8.78±2.6 | 6.28±1.4 | 9.16±5.5 |
| | | | Grp. 1 | 5.78±2.5 | 11.72±9.0 | 15.58±5.7 | 7.95±2.6 | 3.87±1.0 | 5.34±1.0 | 29.55±1.2 | 6.56±0.9 | 6.88±1.9 | 5.92±0.8 | 10.05±5.3 |
| | | | Grp. 2 | 6.32±4.5 | 11.56±10.2 | 12.42±4.8 | 5.79±2.3 | 3.30±0.9 | 4.59±0.8 | 33.07±1.1 | 4.81±1.2 | 6.24±2.5 | 4.56±0.3 | 9.32±6.2 |
| | | | Grp. 3 | 6.58±5.1 | 11.47±11.9 | 9.94±4.8 | 3.57±2.3 | 3.51±0.4 | 4.75±2.0 | 34.39±0.7 | 4.76±0.0 | 5.85±2.9 | 3.08±0.0 | 7.99±8.2 |
| | | | Grp. 4 | 7.20±3.7 | 11.06±12.0 | 7.63±2.5 | 3.90±0.7 | 4.14±2.6 | 4.63±0.5 | 35.08±1.0 | 4.07±0.3 | 4.32±1.3 | 2.41±0.2 | 7.72±6.7 |
| | | | Grp. 5 | 11.68±6.5 | 14.29±10.4 | 9.93±3.5 | 6.65±1.5 | 5.68±2.1 | 4.73±0.5 | 37.13±1.1 | 7.27±1.9 | 5.88±1.4 | 5.84±1.9 | 9.48±5.1 |
| | | TPR@80 TNR | Grp. 0 | 84.21±3.1 | 87.39±3.3 | 80.89±4.7 | 77.55±2.2 | 80.64±4.0 | 80.15±6.4 | 77.77±0.8 | 83.60±5.5 | 89.45±2.3 | 86.75±2.2 | 77.20±2.8 |
| | | | Grp. 1 | 82.79±2.4 | 82.06±3.2 | 75.65±10.5 | 80.99±2.6 | 84.02±3.7 | 84.83±1.2 | 77.15±2.5 | 79.26±5.6 | 85.09±2.6 | 87.12±2.0 | 81.47±4.8 |
| | | | Grp. 2 | 84.94±2.0 | 84.48±8.9 | 78.37±3.8 | 87.10±5.4 | 87.11±2.2 | 88.32±1.2 | 81.78±2.2 | 83.15±3.8 | 88.12±3.0 | 91.97±1.8 | 83.87±1.2 |
| | | | Grp. 3 | 83.59±1.5 | 93.28±6.5 | 85.30±11.5 | 93.50±2.0 | 86.96±3.0 | 86.30±4.9 | 88.95±1.7 | 83.39±10.0 | 96.41±0.6 | 94.86±3.8 | 91.19±1.0 |
| | | | Grp. 4 | 81.76±5.8 | 79.55±14.4 | 86.48±10.3 | 83.02±8.0 | 74.54±11.6 | 74.53±14.1 | 69.73±5.6 | 69.91±14.2 | 87.89±5.1 | 83.99±3.4 | 75.49±4.0 |
| | | | Grp. 5 | 66.13±9.1 | 76.66±0.4 | 56.08±19.9 | 61.99±18.1 | 77.38±0.4 | 57.80±1.6 | 56.73±20.4 | 68.08±9.3 | 70.60±11.6 | 71.03±12.5 | 76.36±0.7 |
| | | FPR | Grp. 0 | 5.18±0.8 | 7.97±1.7 | 6.50±2.4 | 4.91±1.0 | 7.84±2.2 | 7.83±4.2 | 8.63±1.8 | 5.84±1.5 | 8.37±2.9 | 6.51±1.6 | 7.57±0.7 |
| | | | Grp. 1 | 5.39±1.5 | 7.58±3.5 | 5.56±2.3 | 5.05±0.5 | 7.32±1.8 | 6.65±2.0 | 11.20±3.5 | 7.66±1.5 | 6.82±2.4 | 5.22±1.2 | 5.47±1.9 |
| | | | Grp. 2 | 7.09±2.1 | 7.78±4.1 | 4.88±1.3 | 6.04±1.3 | 6.04±1.3 | 5.81±2.0 | 9.29±2.3 | 8.36±1.3 | 7.08±2.2 | 5.46±1.1 | 5.22±2.4 |
| | | | Grp. 3 | 4.74±2.6 | 7.18±3.3 | 2.58±0.2 | 5.56±2.1 | 4.74±1.8 | 5.15±3.3 | 4.61±0.2 | 5.28±0.8 | 4.88±0.4 | 4.47±0.8 | 5.69±0.0 |
| | | | Grp. 4 | 3.48±1.9 | 3.98±1.1 | 2.99±0.7 | 5.47±3.0 | 2.98±1.3 | 2.49±1.7 | 2.49±0.9 | 3.98±0.4 | 2.49±1.1 | 3.24±0.4 | 4.23±0.9 |
| | | | Grp. 5 | 4.76±3.2 | 7.94±0.0 | 4.23±1.8 | 6.35±4.2 | 3.70±1.8 | 3.17±3.2 | 2.65±1.8 | 5.29±0.9 | 3.70±0.9 | 5.82±2.4 | 6.35±1.6 |
| | | FNR | Grp. 0 | 31.62±1.5 | 26.50±3.0 | 35.04±5.3 | 37.61±1.5 | 28.21±6.8 | 38.46±7.7 | 33.34±4.4 | 29.92±1.5 | 25.64±6.8 | 25.64±8.9 | 38.46±0.0 |
| | | | Grp. 1 | 33.70±2.3 | 32.23±6.6 | 41.76±4.0 | 38.10±5.5 | 30.77±4.8 | 29.30±5.2 | 31.87±1.9 | 36.27±4.8 | 28.94±2.8 | 24.17±5.0 | 32.97±4.0 |
| | | | Grp. 2 | 30.67±4.6 | 27.33±8.1 | 34.67±4.2 | 32.00±3.5 | 28.00±4.0 | 29.33±3.1 | 28.67±6.4 | 30.67±9.9 | 24.00±5.3 | 18.67±8.3 | 34.67±5.8 |
| | | | Grp. 3 | 39.79±3.7 | 24.73±8.1 | 46.24±6.7 | 29.03±11.6 | 38.71±11.6 | 37.63±15.9 | 33.33±4.9 | 34.41±4.9 | 27.96±6.7 | 11.83±4.9 | 33.33±3.7 |
| | | | Grp. 4 | 56.67±5.8 | 36.67±11.5 | 56.67±5.8 | 33.33±15.3 | 46.67±5.8 | 50.00±10.0 | 66.67±5.8 | 50.00±20.0 | 36.67±15.3 | 20.00±0.0 | 50.00±10.0 |
| | | | Grp. 5 | 58.33±14.4 | 41.67±14.4 | 50.00±43.3 | 50.00±43.3 | 50.00±25.0 | 33.33±14.4 | 58.33±38.2 | 41.67±52.0 | 58.33±14.4 | 25.00±25.0 | 41.67±14.4 |

Table A13: Results of other metrics for COVID-CT-MD, PAPILA, and OL3I dataset.

| Dataset | Attr. | Metrics | Group | ERM | Resample | DomainInd | LAFTR | CFair | LNL | EnD | ODR | GroupDRO | SWAD | SAM |
|---|---|---|---|---|---|---|---|---|---|---|---|---|---|---|
| COVID-CT-MD | Age | BCE | Grp. 0 | 0.76±0.24 | 0.73±0.08 | 1.18±0.71 | 0.71±0.04 | 0.72±0.02 | 0.77±0.08 | 0.69±0.00 | 1.21±0.80 | 1.04±0.20 | 0.58±0.04 | 1.46±1.31 |
| | | | Grp. 1 | 0.72±0.03 | 0.77±0.16 | 1.30±0.55 | 0.77±0.08 | 0.78±0.15 | 0.87±0.09 | 0.74±0.01 | 1.06±0.35 | 0.95±0.18 | 0.60±0.07 | 1.24±0.90 |
| | | ECE | Grp. 0 | 0.26±0.09 | 0.22±0.09 | 0.26±0.09 | 0.19±0.05 | 0.18±0.03 | 0.23±0.05 | 0.17±0.04 | 0.31±0.17 | 0.34±0.04 | 0.16±0.02 | 0.30±0.14 |
| | | | Grp. 1 | 0.31±0.04 | 0.34±0.07 | 0.40±0.05 | 0.35±0.05 | 0.33±0.14 | 0.37±0.02 | 0.31±0.04 | 0.34±0.10 | 0.30±0.12 | 0.26±0.05 | 0.41±0.16 |
| | | TPR@80 TNR | Grp. 0 | 49.91±8.7 | 46.96±3.1 | 38.86±9.3 | 33.86±14.8 | 34.54±17.6 | 33.51±4.5 | 32.86±2.9 | 51.59±11.3 | 52.17±5.7 | 50.43±7.6 | 40.29±0.5 |
| | | | Grp. 1 | 38.67±8.3 | 27.56±13.9 | 37.33±9.2 | 24.00±13.9 | 32.00±13.9 | 37.60±10.0 | 29.33±2.3 | 46.67±2.3 | 30.93±15.0 | 45.33±2.4 | 36.44±9.6 |
| | | FPR | Grp. 0 | 47.62±20.8 | 74.60±11.0 | 58.73±2.7 | 74.61±27.5 | 79.36±7.3 | 63.49±18.0 | 77.78±2.7 | 52.38±9.5 | 50.79±7.3 | 50.79±9.9 | 55.55±21.5 |
| | | | Grp. 1 | 41.67±8.3 | 44.44±9.6 | 44.44±12.7 | 55.55±41.1 | 52.78±24.1 | 41.67±25.0 | 33.33±0.0 | 44.45±4.8 | 52.78±17.3 | 33.33±8.3 | 27.78±12.7 |
| | | FNR | Grp. 0 | 14.49±9.0 | 7.25±2.5 | 10.15±2.5 | 13.04±15.1 | 5.80±5.0 | 10.15±6.6 | 5.80±2.5 | 14.49±6.6 | 11.60±5.0 | 8.70±7.5 | 11.60±12.6 |
| | | | Grp. 1 | 40.00±20.0 | 33.33±11.5 | 40.00±0.0 | 26.67±46.2 | 33.33±11.5 | 26.67±11.5 | 40.00±0.0 | 33.33±11.5 | 26.67±11.5 | 33.33±11.5 | 46.67±23.1 |
| | | EqOdd | - | 80.30±12.7 | 71.88±6.1 | 77.93±5.1 | 80.77±19.9 | 72.94±11.8 | 80.83±6.0 | 60.68±0.1 | 86.61±4.8 | 88.30±3.9 | 78.95±3.6 | 68.57±10.8 |
| | Sex | BCE | Grp. 0 | 0.66±0.08 | 1.20±0.36 | 1.14±0.61 | 0.96±0.35 | 0.73±0.06 | 0.86±0.30 | 0.71±0.00 | 0.85±0.20 | 0.80±0.21 | 0.66±0.10 | 0.74±0.03 |
| | | | Grp. 1 | 0.63±0.06 | 0.96±0.49 | 1.06±0.43 | 0.99±0.63 | 0.78±0.17 | 0.73±0.10 | 0.69±0.00 | 0.69±0.13 | 0.64±0.13 | 0.59±0.12 | 0.57±0.06 |
| | | ECE | Grp. 0 | 0.23±0.03 | 0.33±0.05 | 0.30±0.09 | 0.28±0.07 | 0.22±0.04 | 0.27±0.10 | 0.25±0.10 | 0.30±0.09 | 0.25±0.11 | 0.22±0.05 | 0.28±0.01 |
| | | | Grp. 1 | 0.24±0.07 | 0.26±0.12 | 0.31±0.02 | 0.31±0.11 | 0.26±0.07 | 0.23±0.03 | 0.21±0.03 | 0.25±0.06 | 0.20±0.10 | 0.25±0.10 | 0.20±0.04 |
| | | TPR@80 TNR | Grp. 0 | 51.56±11.1 | 35.56±6.7 | 42.96±4.5 | 47.70±10.5 | 32.44±11.1 | 39.67±7.8 | 34.67±3.5 | 52.44±19.1 | 47.44±8.1 | 57.33±12.2 | 46.71±4.3 |
| | | | Grp. 1 | 42.05±12.3 | 61.71±10.1 | 40.00±5.3 | 61.88±5.6 | 43.59±10.2 | 42.05±4.4 | 47.18±10.8 | 50.43±28.5 | 53.33±16.0 | 48.63±10.6 | 53.33±15.8 |
| | | FPR | Grp. 0 | 44.44±9.9 | 63.49±12.0 | 53.97±15.3 | 57.14±9.5 | 68.25±7.3 | 63.49±7.3 | 58.73±2.7 | 41.27±30.2 | 52.38±23.8 | 41.27±12.0 | 55.55±2.7 |
| | | | Grp. 1 | 30.56±12.7 | 61.11±21.0 | 50.00±0.0 | 50.00±0.0 | 47.22±12.7 | 47.22±9.6 | 77.78±4.8 | 47.22±33.7 | 44.44±12.7 | 36.11±17.3 | 44.44±9.6 |
| | | FNR | Grp. 0 | 17.78±7.7 | 17.78±16.8 | 13.33±6.7 | 6.67±6.7 | 4.45±3.9 | 13.33±6.7 | 13.33±0.0 | 17.78±20.4 | 11.11±3.8 | 17.78±13.9 | 17.78±10.2 |
| | | | Grp. 1 | 25.64±4.4 | 7.69±0.0 | 10.25±4.4 | 10.25±4.4 | 23.07±13.3 | 23.07±13.3 | 5.13±4.4 | 23.07±20.4 | 12.82±4.4 | 20.51±8.9 | 20.51±16.0 |
| | | EqOdd | - | 89.12±6.4 | 86.84±5.0 | 92.28±4.8 | 91.36±4.1 | 80.17±5.4 | 90.78±6.2 | 86.37±1.9 | 81.22±7.0 | 93.59±3.0 | 92.82±0.8 | 85.21±12.2 |
| PAPILA | Age | BCE | Grp. 0 | 0.34±0.04 | 0.27±0.02 | 0.52±0.03 | 0.15±0.13 | 0.20±0.13 | 0.28±0.03 | 0.53±0.02 | 0.16±0.02 | 0.23±0.05 | 0.09±0.08 | 0.72±0.43 |
| | | | Grp. 1 | 0.74±0.05 | 0.82±0.04 | 1.36±0.15 | 1.42±0.65 | 0.86±0.17 | 0.87±0.18 | 0.68±0.01 | 0.67±0.06 | 0.76±0.03 | 4.50±2.60 | 1.25±0.22 |
| | | ECE | Grp. 0 | 0.15±0.04 | 0.17±0.06 | 0.11±0.00 | 0.08±0.04 | 0.10±0.03 | 0.15±0.01 | 0.41±0.01 | 0.12±0.02 | 0.12±0.02 | 0.04±0.03 | 0.11±0.00 |
| | | | Grp. 1 | 0.25±0.02 | 0.26±0.01 | 0.31±0.02 | 0.32±0.10 | 0.28±0.06 | 0.30±0.08 | 0.24±0.03 | 0.25±0.05 | 0.26±0.03 | 0.35±0.03 | 0.31±0.05 |
| | | TPR@80 TNR | Grp. 0 | 24.59±8.6 | 20.00±0.0 | 32.10±19.2 | 32.44±21.6 | 28.57±14.8 | 24.67±22.8 | 20.00±0.0 | 20.00±0.0 | 20.00±0.0 | 20.00±0.0 | 28.00±13.9 |
| | | | Grp. 1 | 23.27±4.7 | 21.90±3.3 | 17.02±3.6 | 16.67±7.9 | 19.21±5.1 | 30.48±16.2 | 15.24±5.9 | 16.49±11.0 | 25.08±8.3 | 25.08±3.1 | 19.81±2.9 |
| | | FPR | Grp. 0 | 6.25±6.2 | 0.00±0.0 | 6.25±6.2 | 2.08±3.6 | 2.08±3.6 | 18.75±27.2 | 2.08±3.6 | 14.58±13.0 | 0.00±0.0 | 4.17±7.2 | 6.25±6.2 |
| | | | Grp. 1 | 56.86±3.4 | 56.86±3.4 | 60.78±3.4 | 47.06±15.6 | 47.06±5.9 | 52.94±27.0 | 54.90±6.8 | 66.67±3.4 | 60.78±9.0 | 37.25±14.8 | 62.75±3.4 |
| | | FNR | Grp. 0 | 83.33±28.9 | 50.00±0.0 | 66.67±28.9 | 16.67±28.9 | 16.67±28.9 | 50.00±50.0 | 16.67±28.9 | 0.00±0.0 | 33.33±28.9 | 16.67±28.9 | 50.00±50.0 |
| | | | Grp. 1 | 23.81±8.2 | 19.05±8.2 | 33.33±8.3 | 28.57±14.3 | 28.57±0.0 | 28.57±14.3 | 33.33±8.3 | 19.05±8.2 | 19.05±8.2 | 33.33±8.3 | 19.05±8.2 |
| | | EqOdd | - | 44.93±18.2 | 56.09±5.8 | 56.07±12.4 | 64.42±1.3 | 64.42±2.0 | 67.43±10.1 | 58.11±2.7 | 64.43±10.1 | 57.70±2.1 | 72.74±10.4 | 46.75±17.5 |
| | Sex | BCE | Grp. 0 | 0.64±0.12 | 1.08±0.19 | 0.57±0.04 | 0.55±0.05 | 0.57±0.12 | 4.93±2.90 | 0.57±0.01 | 0.45±0.03 | 0.77±0.30 | 1.10±0.17 | 0.64±0.26 |
| | | | Grp. 1 | 0.77±0.53 | 1.13±0.59 | 0.53±0.06 | 0.47±0.00 | 0.56±0.10 | 3.61±2.61 | 0.59±0.02 | 0.42±0.05 | 0.81±0.32 | 1.08±0.14 | 0.61±0.27 |
| | | ECE | Grp. 0 | 0.19±0.03 | 0.30±0.04 | 0.23±0.04 | 0.27±0.02 | 0.20±0.01 | 0.25±0.02 | 0.27±0.04 | 0.15±0.06 | 0.22±0.05 | 0.23±0.05 | 0.23±0.02 |
| | | | Grp. 1 | 0.24±0.06 | 0.30±0.06 | 0.19±0.01 | 0.17±0.06 | 0.18±0.03 | 0.33±0.05 | 0.26±0.03 | 0.16±0.08 | 0.18±0.02 | 0.25±0.07 | 0.22±0.03 |
| | | TPR@80 TNR | Grp. 0 | 27.64±24.0 | 45.00±31.2 | 11.00±6.9 | 13.00±14.7 | 20.33±12.9 | 50.83±17.7 | 54.50±4.8 | 25.00±26.5 | 46.67±20.8 | 38.81±25.0 | 17.50±11.5 |
| | | | Grp. 1 | 46.67±6.1 | 35.00±9.0 | 37.16±8.0 | 30.67±4.6 | 34.11±12.3 | 56.67±18.6 | 60.27±6.7 | 40.89±7.8 | 19.71±15.7 | 40.00±10.6 | 43.67±7.5 |
| | | FPR | Grp. 0 | 20.83±3.6 | 20.83±7.2 | 39.58±25.3 | 45.83±7.2 | 50.00±38.0 | 16.67±7.2 | 8.33±9.5 | 22.92±3.6 | 18.75±6.2 | 22.92±7.2 | 39.58±9.5 |
| | | | Grp. 1 | 31.37±3.4 | 25.49±12.2 | 41.18±27.0 | 43.14±9.0 | 47.06±32.7 | 19.61±14.8 | 21.57±9.0 | 31.37±9.0 | 23.53±10.2 | 33.33±6.8 | 41.18±5.9 |
| | | FNR | Grp. 0 | 50.00±0.0 | 33.33±14.4 | 25.00±25.0 | 25.00±0.0 | 25.00±0.0 | 41.67±14.4 | 58.33±14.4 | 41.67±14.4 | 58.33±14.4 | 33.33±14.4 | 25.00±0.0 |
| | | | Grp. 1 | 13.33±11.5 | 33.33±11.5 | 33.33±23.1 | 0.00±0.0 | 26.67±30.6 | 33.33±11.5 | 20.00±20.0 | 13.33±23.1 | 13.33±11.5 | 20.00±0.0 | 13.33±11.5 |
| | | EqOdd | - | 76.40±5.8 | 92.43±2.2 | 82.59±5.6 | 83.95±3.3 | 84.39±7.2 | 92.16±0.4 | 74.22±13.2 | 76.61±12.1 | 74.74±11.2 | 87.51±8.5 | 92.39±5.5 |
| OL3I | Age | BCE | Grp. 0 | 0.49±0.48 | 0.10±0.01 | 0.15±0.07 | 0.10±0.01 | 0.17±0.12 | 0.15±0.04 | 0.67±0.01 | 0.11±0.01 | 0.53±0.31 | 0.10±0.01 | 0.15±0.02 |
| | | | Grp. 1 | 0.64±0.54 | 0.37±0.03 | 0.32±0.05 | 0.31±0.01 | 0.37±0.07 | 0.30±0.03 | 0.67±0.01 | 0.36±0.08 | 0.59±0.27 | 0.33±0.07 | 0.29±0.02 |
| | | ECE | Grp. 0 | 0.30±0.27 | 0.02±0.00 | 0.07±0.08 | 0.02±0.00 | 0.09±0.12 | 0.08±0.06 | 0.47±0.01 | 0.02±0.01 | 0.36±0.19 | 0.02±0.00 | 0.08±0.03 |
| | | | Grp. 1 | 0.28±0.31 | 0.07±0.01 | 0.09±0.07 | 0.05±0.01 | 0.10±0.07 | 0.06±0.04 | 0.41±0.01 | 0.06±0.03 | 0.32±0.18 | 0.06±0.04 | 0.05±0.02 |
| | | TPR@80 TNR | Grp. 0 | 41.37±10.1 | 67.76±4.2 | 30.45±14.1 | 66.27±17.6 | 50.28±21.7 | 49.12±15.6 | 28.07±1.3 | 51.00±10.6 | 36.75±30.0 | 56.81±11.7 | 53.02±7.7 |
| | | | Grp. 1 | 22.33±10.1 | 18.85±7.8 | 26.24±7.1 | 26.48±0.4 | 29.36±2.9 | 28.30±4.5 | 19.09±6.5 | 26.39±6.6 | 23.09±10.1 | 27.73±16.0 | 28.49±8.3 |
| | | FPR | Grp. 0 | 15.61±13.5 | 13.06±9.0 | 18.60±15.3 | 14.12±6.9 | 9.02±4.3 | 4.04±2.0 | 6.40±4.0 | 11.94±11.6 | 21.33±6.8 | 8.21±12.3 | 9.95±9.0 |
| | | | Grp. 1 | 27.32±20.5 | 30.60±22.6 | 35.65±16.3 | 34.97±12.3 | 17.90±12.4 | 12.43±4.0 | 5.19±2.7 | 24.18±21.0 | 34.02±8.6 | 27.46±29.6 | 11.61±8.9 |
| | | FNR | Grp. 0 | 63.89±17.3 | 50.00±22.0 | 63.89±17.3 | 38.89±19.2 | 66.67±22.1 | 80.56±9.6 | 80.56±9.6 | 66.67±25.0 | 58.33±30.0 | 69.44±17.3 | 66.67±22.0 |
| | | | Grp. 1 | 62.12±18.4 | 63.63±27.6 | 45.46±23.6 | 51.51±16.0 | 72.73±13.6 | 78.79±6.9 | 92.42±6.9 | 63.64±19.8 | 51.51±6.9 | 62.12±26.6 | 78.79±17.2 |
| | | EqOdd | - | 90.23±6.7 | 84.41±5.2 | 82.25±7.6 | 78.71±4.8 | 90.76±5.7 | 94.16±0.8 | 91.50±5.0 | 90.34±2.0 | 83.43±2.4 | 84.44±11.5 | 93.11±2.4 |
| | Sex | BCE | Grp. 0 | 0.33±0.14 | 0.31±0.11 | 0.52±0.24 | 0.25±0.02 | 0.28±0.08 | 0.27±0.04 | 0.67±0.01 | 0.30±0.09 | 0.28±0.06 | 0.25±0.03 | 0.24±0.01 |
| | | | Grp. 1 | 0.25±0.18 | 0.24±0.15 | 0.35±0.32 | 0.13±0.02 | 0.15±0.01 | 0.19±0.06 | 0.67±0.01 | 0.21±0.14 | 0.18±0.10 | 0.12±0.00 | 0.16±0.02 |
| | | ECE | Grp. 0 | 0.14±0.14 | 0.14±0.11 | 0.19±0.17 | 0.04±0.00 | 0.05±0.01 | 0.09±0.05 | 0.42±0.01 | 0.11±0.11 | 0.09±0.08 | 0.04±0.01 | 0.04±0.01 |
| | | | Grp. 1 | 0.15±0.15 | 0.14±0.14 | 0.18±0.25 | 0.03±0.03 | 0.04±0.01 | 0.11±0.06 | 0.46±0.01 | 0.11±0.14 | 0.08±0.10 | 0.03±0.01 | 0.06±0.03 |
| | | TPR@80 TNR | Grp. 0 | 40.07±9.8 | 42.58±1.8 | 23.43±11.3 | 39.86±4.5 | 35.99±12.1 | 36.35±9.1 | 16.04±6.7 | 34.76±8.4 | 33.07±8.7 | 38.08±4.0 | 37.44±2.5 |
| | | | Grp. 1 | 54.11±10.0 | 65.09±13.5 | 40.71±26.1 | 51.16±10.8 | 37.11±3.8 | 54.78±2.7 | 18.09±8.1 | 51.06±34.5 | 53.04±9.8 | 58.25±11.0 | 31.83±2.5 |
| | | FPR | Grp. 0 | 22.56±14.9 | 21.44±10.5 | 33.89±26.5 | 9.33±7.8 | 25.55±14.7 | 16.44±10.6 | 45.00±36.3 | 21.44±9.1 | 29.56±7.1 | 16.11±8.2 | 9.44±5.5 |
| | | | Grp. 1 | 13.82±10.7 | 17.64±11.7 | 34.38±33.4 | 5.42±3.7 | 19.72±14.6 | 10.90±7.0 | 43.12±35.2 | 11.53±7.3 | 18.06±4.1 | 10.07±6.6 | 6.81±5.8 |
| | | FNR | Grp. 0 | 51.67±28.4 | 46.67±22.5 | 61.67±25.7 | 75.00±13.2 | 53.33±5.8 | 70.00±8.7 | 51.67±37.9 | 60.00±20.0 | 48.33±2.9 | 68.33±7.6 | 73.33±5.8 |
| | | | Grp. 1 | 50.00±18.9 | 40.48±27.0 | 38.09±27.0 | 80.95±10.9 | 64.29±18.9 | 61.90±20.6 | 35.71±43.4 | 64.29±24.7 | 42.86±18.9 | 54.76±18.0 | 83.34±16.5 |
| | | EqOdd | - | 91.94±3.2 | 93.32±4.6 | 75.69±4.4 | 94.59±4.5 | 88.55±8.7 | 91.28±4.6 | 91.09±5.8 | 89.57±0.4 | 86.75±2.0 | 90.19±5.9 | 91.78±4.8 |

Table A14: Results of other metrics in out-of-distribution setting. In the dataset column, the dataset in the first row is the training domain, and the second row is the testing domain.

| Dataset | Attr. | Metrics | Group | ERM | Resample | DomainInd | LAFTR | CFair | LNL | EnD | ODR | GroupDRO | SWAD | SAM |
|---|---|---|---|---|---|---|---|---|---|---|---|---|---|---|
| CheXpert MIMIC-CXR | Age | BCE | Grp. 0 | 0.64±0.03 | 0.91±0.09 | 1.86±0.31 | 0.68±0.05 | 0.79±0.15 | 0.76±0.09 | 1.03±0.17 | 0.84±0.14 | 0.70±0.09 | 0.82±0.03 | 0.67±0.03 |
| | | | Grp. 1 | 0.70±0.01 | 0.90±0.08 | 2.07±0.30 | 0.71±0.03 | 0.82±0.14 | 0.74±0.07 | 0.97±0.17 | 0.86±0.10 | 0.72±0.09 | 0.93±0.01 | 0.72±0.03 |
| | | | Grp. 2 | 0.72±0.01 | 0.80±0.08 | 2.16±0.23 | 0.69±0.03 | 0.73±0.09 | 0.67±0.05 | 0.76±0.12 | 0.77±0.06 | 0.69±0.08 | 0.95±0.02 | 0.71±0.03 |
| | | | Grp. 3 | 0.73±0.01 | 0.74±0.06 | 2.33±0.22 | 0.69±0.05 | 0.68±0.06 | 0.63±0.04 | 0.65±0.08 | 0.71±0.03 | 0.66±0.06 | 0.95±0.02 | 0.70±0.02 |
| | | | Grp. 4 | 0.68±0.00 | 0.64±0.05 | 2.18±0.22 | 0.62±0.05 | 0.59±0.04 | 0.56±0.04 | 0.55±0.04 | 0.61±0.02 | 0.60±0.04 | 0.84±0.02 | 0.64±0.02 |
| | | ECE | Grp. 0 | 0.28±0.01 | 0.36±0.03 | 0.43±0.05 | 0.30±0.03 | 0.36±0.08 | 0.32±0.07 | 0.45±0.07 | 0.36±0.06 | 0.28±0.07 | 0.36±0.01 | 0.27±0.02 |
| | | | Grp. 1 | 0.29±0.00 | 0.35±0.04 | 0.43±0.03 | 0.29±0.02 | 0.34±0.07 | 0.29±0.06 | 0.40±0.07 | 0.35±0.04 | 0.27±0.08 | 0.38±0.00 | 0.29±0.02 |
| | | | Grp. 2 | 0.26±0.00 | 0.27±0.04 | 0.36±0.03 | 0.24±0.02 | 0.25±0.05 | 0.21±0.04 | 0.24±0.07 | 0.26±0.02 | 0.21±0.07 | 0.32±0.00 | 0.25±0.01 |
| | | | Grp. 3 | 0.23±0.00 | 0.22±0.03 | 0.30±0.02 | 0.21±0.02 | 0.20±0.03 | 0.17±0.03 | 0.16±0.06 | 0.21±0.01 | 0.17±0.05 | 0.27±0.00 | 0.21±0.01 |
| | | | Grp. 4 | 0.18±0.00 | 0.17±0.02 | 0.23±0.01 | 0.16±0.02 | 0.15±0.02 | 0.13±0.02 | 0.11±0.03 | 0.15±0.01 | 0.14±0.03 | 0.21±0.00 | 0.17±0.01 |
| | | TPR@80 TNR | Grp. 0 | 73.58±6.0 | 60.45±6.0 | 63.12±12.0 | 70.07±4.0 | 72.05±4.0 | 59.42±9.0 | 65.48±8.0 | 53.00±7.0 | 56.11±7.0 | 76.22±3.0 | 53.54±3.0 |
| | | | Grp. 1 | 67.32±1.0 | 58.35±3.0 | 57.29±8.0 | 65.94±2.0 | 64.42±2.0 | 55.56±3.0 | 58.52±4.0 | 56.04±6.0 | 57.77±1.0 | 69.43±1.0 | 61.52±0.0 |
| | | | Grp. 2 | 69.51±0.0 | 62.95±3.0 | 60.63±6.0 | 70.23±1.0 | 67.61±2.0 | 62.88±1.0 | 63.95±5.0 | 61.08±9.0 | 61.46±1.0 | 71.51±1.0 | 66.99±1.0 |
| | | | Grp. 3 | 64.15±0.0 | 60.47±3.0 | 57.93±6.0 | 65.12±1.0 | 63.23±1.0 | 59.91±2.0 | 61.36±4.0 | 56.24±8.0 | 56.35±1.0 | 65.89±1.0 | 63.84±0.0 |
| | | | Grp. 4 | 55.47±1.0 | 54.00±2.0 | 51.37±5.0 | 56.47±2.0 | 55.29±2.0 | 54.18±1.0 | 54.18±2.0 | 50.95±7.0 | 48.18±2.0 | 57.50±1.0 | 55.60±0.0 |
| | | FPR | Grp. 0 | 59.85±4.0 | 53.79±7.0 | 60.61±6.0 | 56.82±9.0 | 60.23±9.0 | 71.21±9.0 | 45.08±11.0 | 59.47±3.0 | 68.18±2.0 | 52.27±2.0 | 72.35±1.0 |
| | | | Grp. 1 | 52.75±1.0 | 48.85±2.0 | 53.50±1.0 | 49.07±4.0 | 47.43±6.0 | 58.15±3.0 | 42.23±5.0 | 54.97±5.0 | 60.03±3.0 | 46.60±1.0 | 57.02±1.0 |
| | | | Grp. 2 | 38.62±1.0 | 41.28±2.0 | 42.91±2.0 | 37.82±3.0 | 38.17±4.0 | 46.61±2.0 | 39.02±4.0 | 44.94±7.0 | 44.41±3.0 | 35.38±1.0 | 40.67±1.0 |
| | | | Grp. 3 | 24.22±1.0 | 29.32±2.0 | 30.27±3.0 | 25.07±1.0 | 28.53±2.0 | 30.35±1.0 | 31.80±4.0 | 34.99±8.0 | 28.59±2.0 | 23.94±0.0 | 25.55±1.0 |
| | | | Grp. 4 | 12.89±1.0 | 18.76±3.0 | 20.52±4.0 | 14.02±2.0 | 20.21±2.0 | 16.68±3.0 | 22.58±4.0 | 25.31±10.0 | 16.38±2.0 | 14.87±1.0 | 13.82±0.0 |
| | | FNR | Grp. 0 | 2.26±2.0 | 11.11±7.0 | 4.86±1.0 | 3.12±3.0 | 4.86±4.0 | 4.34±1.0 | 9.98±5.0 | 4.51±2.0 | 4.51±2.0 | 3.82±1.0 | 2.95±0.0 |
| | | | Grp. 1 | 4.00±0.0 | 8.92±3.0 | 5.87±2.0 | 4.86±1.0 | 6.21±3.0 | 4.69±1.0 | 11.98±3.0 | 6.56±1.0 | 4.93±1.0 | 5.40±0.0 | 4.23±0.0 |
| | | | Grp. 2 | 10.90±1.0 | 13.28±1.0 | 13.05±3.0 | 11.55±2.0 | 12.01±2.0 | 10.85±1.0 | 13.95±2.0 | 11.79±1.0 | 13.58±1.0 | 12.43±1.0 | 12.01±0.0 |
| | | | Grp. 3 | 29.16±2.0 | 27.08±2.0 | 28.58±3.0 | 27.37±2.0 | 24.63±2.0 | 26.12±3.0 | 23.33±2.0 | 24.07±2.0 | 31.66±2.0 | 28.31±1.0 | 27.74±1.0 |
| | | | Grp. 4 | 58.00±3.0 | 47.82±4.0 | 48.38±1.0 | 53.67±2.0 | 43.78±4.0 | 51.15±5.0 | 42.17±5.0 | 40.79±8.0 | 57.51±4.0 | 51.07±0.0 | 56.45±1.0 |
| | Sex | BCE | Grp. 0 | 0.70±0.04 | 0.63±0.08 | 0.92±0.05 | 0.72±0.07 | 0.70±0.07 | 0.65±0.00 | 0.64±0.05 | 0.69±0.11 | 0.73±0.01 | 0.90±0.01 | 0.65±0.02 |
| | | | Grp. 1 | 0.73±0.03 | 0.66±0.09 | 0.94±0.09 | 0.76±0.08 | 0.70±0.05 | 0.68±0.03 | 0.70±0.05 | 0.71±0.14 | 0.75±0.01 | 0.92±0.00 | 0.68±0.03 |
| | | ECE | Grp. 0 | 0.22±0.02 | 0.18±0.06 | 0.22±0.02 | 0.23±0.04 | 0.22±0.04 | 0.20±0.00 | 0.19±0.03 | 0.21±0.06 | 0.23±0.01 | 0.28±0.00 | 0.20±0.01 |
| | | | Grp. 1 | 0.25±0.02 | 0.21±0.06 | 0.25±0.04 | 0.26±0.04 | 0.24±0.03 | 0.23±0.02 | 0.23±0.03 | 0.23±0.07 | 0.26±0.01 | 0.30±0.01 | 0.24±0.02 |
| | | TPR@80 TNR | Grp. 0 | 68.33±1.0 | 69.40±2.0 | 69.25±1.0 | 68.06±2.0 | 68.37±2.0 | 67.97±1.0 | 69.98±0.0 | 70.41±1.0 | 67.32±1.0 | 71.88±0.0 | 71.14±0.0 |
| | | | Grp. 1 | 70.98±1.0 | 71.25±1.0 | 71.25±1.0 | 70.76±1.0 | 70.03±2.0 | 68.78±2.0 | 71.31±1.0 | 71.41±1.0 | 70.04±1.0 | 72.95±0.0 | 72.33±0.0 |
| | | FPR | Grp. 0 | 28.89±1.0 | 28.73±3.0 | 27.97±1.0 | 29.47±2.0 | 28.04±3.0 | 28.62±2.0 | 27.28±1.0 | 28.41±1.0 | 28.37±1.0 | 26.48±1.0 | 28.05±2.0 |
| | | | Grp. 1 | 31.91±1.0 | 30.96±2.0 | 31.05±1.0 | 32.19±2.0 | 36.26±2.0 | 33.96±3.0 | 27.28±1.0 | 32.02±1.0 | 31.83±2.0 | 30.20±1.0 | 31.09±2.0 |
| | | FNR | Grp. 0 | 21.03±1.0 | 20.65±1.0 | 21.47±1.0 | 20.86±3.0 | 21.63±2.0 | 21.81±2.0 | 22.49±1.0 | 20.06±0.0 | 23.08±2.0 | 20.71±1.0 | 20.16±2.0 |
| | | | Grp. 1 | 16.45±1.0 | 16.99±1.0 | 16.99±1.0 | 17.40±2.0 | 16.82±1.0 | 13.45±1.0 | 16.28±2.0 | 20.53±2.0 | 15.86±1.0 | 16.81±1.0 | 16.04±2.0 |
| | | EqOdd | - | 96.20±1.0 | 97.05±0.0 | 96.43±1.0 | 96.62±1.0 | 91.80±1.0 | 94.57±4.0 | 98.74±0.0 | 96.09±1.0 | 95.46±1.0 | 96.19±0.0 | 96.42±0.0 |
| | Race | BCE | Grp. 0 | 0.68±0.02 | 0.66±0.04 | 0.95±0.12 | 0.69±0.02 | 0.55±0.03 | 0.68±0.01 | 0.62±0.04 | 0.67±0.05 | 0.71±0.05 | 0.89±0.00 | 0.65±0.01 |
| | | | Grp. 1 | 0.72±0.04 | 0.72±0.08 | 0.95±0.12 | 0.75±0.04 | 0.60±0.05 | 0.75±0.01 | 0.71±0.06 | 0.74±0.06 | 0.75±0.03 | 0.97±0.01 | 0.71±0.01 |
| | | ECE | Grp. 0 | 0.20±0.01 | 0.19±0.02 | 0.23±0.02 | 0.21±0.01 | 0.10±0.04 | 0.21±0.01 | 0.18±0.02 | 0.20±0.01 | 0.21±0.01 | 0.26±0.00 | 0.20±0.01 |
| | | | Grp. 1 | 0.24±0.02 | 0.24±0.03 | 0.27±0.03 | 0.26±0.02 | 0.15±0.05 | 0.26±0.01 | 0.25±0.03 | 0.26±0.02 | 0.26±0.01 | 0.33±0.00 | 0.25±0.01 |
| | | TPR@80 TNR | Grp. 0 | 68.42±1.0 | 67.77±3.0 | 69.90±0.0 | 69.04±1.0 | 67.13±1.0 | 65.60±2.0 | 68.83±2.0 | 68.83±2.0 | 66.73±1.0 | 70.39±1.0 | 69.85±1.0 |
| | | | Grp. 1 | 68.99±1.0 | 68.24±4.0 | 70.34±0.0 | 68.47±1.0 | 67.01±1.0 | 65.95±2.0 | 68.04±3.0 | 68.96±2.0 | 68.33±1.0 | 70.96±0.0 | 69.97±0.0 |
| | | FPR | Grp. 0 | 24.51±1.0 | 27.57±4.0 | 24.10±1.0 | 24.98±1.0 | 27.67±3.0 | 29.97±3.0 | 24.57±1.0 | 27.55±3.0 | 26.56±1.0 | 24.69±1.0 | 24.05±2.0 |
| | | | Grp. 1 | 36.49±1.0 | 38.83±3.0 | 36.23±2.0 | 37.09±1.0 | 39.88±3.0 | 42.18±2.0 | 35.81±1.0 | 38.75±1.0 | 38.75±1.0 | 36.10±1.0 | 36.26±2.0 |
| | | FNR | Grp. 0 | 26.11±0.0 | 22.98±2.0 | 24.76±2.0 | 24.87±1.0 | 23.98±2.0 | 22.47±2.0 | 25.65±1.0 | 21.93±2.0 | 25.44±2.0 | 23.91±1.0 | 25.17±2.0 |
| | | | Grp. 1 | 13.78±1.0 | 12.75±0.0 | 13.60±2.0 | 13.19±1.0 | 13.83±1.0 | 12.30±0.0 | 15.17±1.0 | 12.49±1.0 | 13.46±1.0 | 13.26±0.0 | 13.49±1.0 |
| | | EqOdd | - | 87.84±1.0 | 89.25±1.0 | 88.35±1.0 | 88.10±0.0 | 88.82±1.0 | 88.81±1.0 | 89.13±1.0 | 89.64±0.0 | 87.91±1.0 | 88.97±0.0 | 88.05±0.0 |
| MIMIC-CXR CheXpert | Age | BCE | Grp. 0 | 0.48±0.03 | 0.63±0.10 | 0.80±0.11 | 0.52±0.05 | 0.48±0.04 | 0.64±0.08 | 0.47±0.04 | 0.44±0.02 | 0.50±0.02 | 0.87±0.03 | 0.91±0.18 |
| | | | Grp. 1 | 0.39±0.01 | 0.46±0.02 | 0.70±0.05 | 0.42±0.03 | 0.41±0.00 | 0.44±0.02 | 0.39±0.01 | 0.39±0.02 | 0.40±0.01 | 0.70±0.00 | 0.77±0.21 |
| | | | Grp. 2 | 0.33±0.01 | 0.35±0.01 | 0.55±0.04 | 0.35±0.02 | 0.35±0.01 | 0.36±0.02 | 0.35±0.02 | 0.33±0.02 | 0.34±0.01 | 0.61±0.00 | 0.74±0.23 |
| | | | Grp. 3 | 0.29±0.01 | 0.30±0.02 | 0.45±0.03 | 0.30±0.02 | 0.33±0.02 | 0.31±0.03 | 0.34±0.02 | 0.29±0.02 | 0.29±0.01 | 0.53±0.00 | 0.71±0.24 |
| | | | Grp. 4 | 0.23±0.01 | 0.23±0.02 | 0.25±0.01 | 0.24±0.02 | 0.28±0.03 | 0.25±0.03 | 0.30±0.02 | 0.23±0.02 | 0.23±0.01 | 0.43±0.01 | 0.63±0.20 |
| | | ECE | Grp. 0 | 0.15±0.03 | 0.17±0.03 | 0.16±0.01 | 0.18±0.04 | 0.17±0.02 | 0.26±0.05 | 0.11±0.04 | 0.13±0.02 | 0.15±0.02 | 0.34±0.01 | 0.42±0.08 |
| | | | Grp. 1 | 0.11±0.02 | 0.11±0.02 | 0.13±0.00 | 0.13±0.03 | 0.12±0.01 | 0.14±0.03 | 0.09±0.02 | 0.09±0.03 | 0.11±0.02 | 0.28±0.00 | 0.36±0.09 |
| | | | Grp. 2 | 0.11±0.02 | 0.12±0.02 | 0.10±0.00 | 0.13±0.03 | 0.13±0.02 | 0.14±0.02 | 0.13±0.02 | 0.11±0.03 | 0.12±0.01 | 0.27±0.00 | 0.38±0.10 |
| | | | Grp. 3 | 0.12±0.01 | 0.12±0.02 | 0.08±0.00 | 0.14±0.02 | 0.12±0.02 | 0.15±0.03 | 0.17±0.01 | 0.12±0.02 | 0.13±0.01 | 0.26±0.00 | 0.40±0.11 |
| | | | Grp. 4 | 0.13±0.01 | 0.12±0.02 | 0.04±0.00 | 0.14±0.02 | 0.16±0.03 | 0.15±0.03 | 0.19±0.01 | 0.13±0.02 | 0.13±0.01 | 0.25±0.00 | 0.39±0.10 |
| | | TPR@80 TNR | Grp. 0 | 70.66±4.0 | 66.39±8.0 | 70.65±4.0 | 71.05±1.0 | 72.94±4.0 | 56.94±5.0 | 69.71±4.0 | 71.65±5.0 | 66.65±6.0 | 75.09±3.0 | 59.25±3.0 |
| | | | Grp. 1 | 79.15±2.0 | 74.96±2.0 | 79.80±2.0 | 79.88±2.0 | 78.30±1.0 | 74.56±1.0 | 80.04±2.0 | 79.77±2.0 | 78.19±2.0 | 83.09±1.0 | 72.68±1.0 |
| | | | Grp. 2 | 79.56±1.0 | 76.94±1.0 | 79.11±1.0 | 78.12±1.0 | 77.38±3.0 | 75.84±1.0 | 76.69±1.0 | 78.49±1.0 | 77.97±1.0 | 81.50±0.0 | 71.92±1.0 |
| | | | Grp. 3 | 69.59±1.0 | 67.66±2.0 | 70.69±1.0 | 66.28±2.0 | 66.74±4.0 | 66.72±2.0 | 69.47±1.0 | 68.81±1.0 | 68.29±2.0 | 72.24±0.0 | 64.09±2.0 |
| | | | Grp. 4 | 63.10±3.0 | 63.24±3.0 | 62.55±2.0 | 60.50±5.0 | 61.11±2.0 | 63.16±2.0 | 63.92±2.0 | 61.32±3.0 | 59.01±3.0 | 66.67±4.0 | 63.60±1.0 |
| | | FPR | Grp. 0 | 22.28±2.0 | 25.16±6.0 | 19.39±6.0 | 25.32±2.0 | 23.08±6.0 | 44.07±7.0 | 16.83±4.0 | 18.75±6.0 | 23.88±6.0 | 19.55±2.0 | 34.62±11.0 |
| | | | Grp. 1 | 16.71±1.0 | 18.78±2.0 | 16.67±2.0 | 18.10±1.0 | 17.03±1.0 | 20.09±2.0 | 14.29±2.0 | 16.98±2.0 | 16.82±2.0 | 17.77±1.0 | 21.98±2.0 |
| | | | Grp. 2 | 11.30±1.0 | 13.05±2.0 | 10.94±1.0 | 11.31±0.0 | 12.65±1.0 | 12.19±1.0 | 10.90±1.0 | 12.41±1.0 | 11.18±2.0 | 11.87±1.0 | 15.60±2.0 |
| | | | Grp. 3 | 8.20±1.0 | 9.91±2.0 | 7.80±1.0 | 7.08±0.0 | 10.10±1.0 | 8.27±1.0 | 10.08±2.0 | 9.72±1.0 | 8.12±1.0 | 7.81±1.0 | 11.92±2.0 |
| | | | Grp. 4 | 4.26±1.0 | 5.76±1.0 | 3.77±1.0 | 3.04±1.0 | 7.46±2.0 | 3.77±1.0 | 7.93±2.0 | 6.25±1.0 | 3.88±1.0 | 3.65±1.0 | 6.76±2.0 |
| | | FNR | Grp. 0 | 26.67±3.0 | 28.89±7.0 | 28.33±5.0 | 25.56±2.0 | 24.45±7.0 | 21.11±8.0 | 36.11±8.0 | 27.22±5.0 | 30.56±7.0 | 23.33±3.0 | 30.00±3.0 |
| | | | Grp. 1 | 25.58±2.0 | 26.92±4.0 | 26.23±5.0 | 23.29±2.0 | 26.98±2.0 | 24.93±5.0 | 32.89±4.0 | 25.49±4.0 | 27.04±5.0 | 20.91±1.0 | 23.53±4.0 |
| | | | Grp. 2 | 36.34±2.0 | 36.82±2.0 | 37.16±3.0 | 39.17±2.0 | 37.77±5.0 | 37.99±3.0 | 39.80±3.0 | 35.63±3.0 | 39.30±4.0 | 33.26±1.0 | 36.14±4.0 |
| | | | Grp. 3 | 57.93±4.0 | 54.62±5.0 | 58.90±3.0 | 63.44±2.0 | 53.34±4.0 | 61.17±5.0 | 52.49±5.0 | 53.74±3.0 | 60.43±5.0 | 57.37±2.0 | 53.85±3.0 |
| | | | Grp. 4 | 76.89±2.0 | 75.13±4.0 | 77.91±5.0 | 83.59±4.0 | 66.92±4.0 | 81.56±8.0 | 66.79±7.0 | 72.47±4.0 | 81.94±4.0 | 81.31±2.0 | 70.20±6.0 |
| | Sex | BCE | Grp. 0 | 0.30±0.00 | 0.32±0.02 | 0.30±0.02 | 0.29±0.03 | 0.32±0.01 | 0.33±0.01 | 0.33±0.04 | 0.30±0.02 | 0.30±0.01 | 0.56±0.00 | 1.25±0.17 |
| | | | Grp. 1 | 0.30±0.00 | 0.32±0.02 | 0.28±0.02 | 0.29±0.04 | 0.31±0.01 | 0.32±0.03 | 0.30±0.03 | 0.29±0.02 | 0.29±0.01 | 0.55±0.01 | 1.18±0.18 |
| | | ECE | Grp. 0 | 0.12±0.00 | 0.13±0.02 | 0.06±0.01 | 0.10±0.03 | 0.13±0.01 | 0.14±0.01 | 0.14±0.04 | 0.11±0.02 | 0.10±0.01 | 0.26±0.00 | 0.54±0.07 |
| | | | Grp. 1 | 0.12±0.01 | 0.14±0.02 | 0.05±0.01 | 0.11±0.04 | 0.12±0.01 | 0.14±0.02 | 0.11±0.03 | 0.10±0.01 | 0.10±0.01 | 0.26±0.00 | 0.52±0.08 |
| | | TPR@80 TNR | Grp. 0 | 80.62±1.0 | 78.88±0.0 | 77.95±1.0 | 79.46±0.0 | 78.89±1.0 | 78.83±0.0 | 79.07±0.0 | 79.59±1.0 | 78.48±1.0 | 81.61±0.0 | 77.28±0.0 |
| | | | Grp. 1 | 76.90±1.0 | 76.31±0.0 | 75.77±1.0 | 77.08±1.0 | 76.86±2.0 | 76.19±0.0 | 76.37±1.0 | 76.50±2.0 | 75.80±1.0 | 78.75±0.0 | 74.92±2.0 |
| | | FPR | Grp. 0 | 9.24±1.0 | 9.05±1.0 | 10.36±1.0 | 9.59±1.0 | 10.63±1.0 | 10.85±2.0 | 10.34±1.0 | 10.38±0.0 | 11.22±1.0 | 8.82±1.0 | 12.10±1.0 |
| | | | Grp. 1 | 8.69±1.0 | 9.10±0.0 | 8.82±1.0 | 9.60±3.0 | 9.64±1.0 | 10.65±4.0 | 7.86±1.0 | 9.69±1.0 | 9.48±2.0 | 8.18±1.0 | 10.33±0.0 |
| | | FNR | Grp. 0 | 42.01±4.0 | 42.96±2.0 | 40.32±1.0 | 41.99±1.0 | 39.79±4.0 | 40.89±6.0 | 38.74±2.0 | 39.68±2.0 | 39.78±2.0 | 40.45±3.0 | 37.99±2.0 |
| | | | Grp. 1 | 43.65±3.0 | 42.55±0.0 | 44.56±3.0 | 41.57±6.0 | 41.60±2.0 | 41.53±8.0 | 46.32±4.0 | 42.22±1.0 | 42.04±4.0 | 43.58±3.0 | 41.56±1.0 |
| | | EqOdd | - | 98.42±1.0 | 98.94±1.0 | 97.11±2.0 | 97.49±2.0 | 97.91±1.0 | 95.27±3.0 | 94.97±1.0 | 98.38±0.0 | 97.96±1.0 | 98.12±0.0 | 97.33±1.0 |
| | Race | BCE | Grp. 0 | 0.31±0.02 | 0.29±0.01 | 0.28±0.01 | 0.31±0.02 | 0.30±0.02 | 0.32±0.00 | 0.35±0.04 | 0.28±0.01 | 0.31±0.01 | 0.57±0.00 | 1.09±0.22 |
| | | | Grp. 1 | 0.32±0.03 | 0.30±0.01 | 0.29±0.01 | 0.32±0.02 | 0.31±0.01 | 0.33±0.00 | 0.36±0.04 | 0.29±0.01 | 0.32±0.01 | 0.58±0.00 | 1.09±0.22 |
| | | ECE | Grp. 0 | 0.13±0.03 | 0.11±0.02 | 0.05±0.01 | 0.13±0.02 | 0.11±0.02 | 0.13±0.01 | 0.15±0.03 | 0.09±0.02 | 0.13±0.01 | 0.27±0.00 | 0.54±0.07 |
| | | | Grp. 1 | 0.12±0.03 | 0.11±0.02 | 0.05±0.01 | 0.13±0.02 | 0.11±0.02 | 0.13±0.00 | 0.15±0.04 | 0.08±0.01 | 0.13±0.01 | 0.27±0.00 | 0.54±0.07 |
| | | TPR@80 TNR | Grp. 0 | 78.40±1.0 | 77.71±1.0 | 79.24±0.0 | 78.88±1.0 | 78.61±2.0 | 75.59±2.0 | 76.24±2.0 | 76.21±3.0 | 79.38±0.0 | 80.59±0.0 | 73.78±1.0 |
| | | | Grp. 1 | 78.62±1.0 | 76.37±0.0 | 77.68±1.0 | 77.54±2.0 | 76.14±2.0 | 74.87±1.0 | 73.83±2.0 | 73.89±4.0 | 78.36±0.0 | 80.33±0.0 | 73.84±1.0 |
| | | FPR | Grp. 0 | 9.11±1.0 | 10.05±0.0 | 9.31±1.0 | 8.71±1.0 | 11.06±1.0 | 11.35±2.0 | 9.12±1.0 | 9.72±2.0 | 9.37±1.0 | 8.96±1.0 | 13.10±0.0 |
| | | | Grp. 1 | 9.45±1.0 | 10.27±0.0 | 9.58±1.0 | 9.06±2.0 | 11.29±1.0 | 11.71±1.0 | 9.49±1.0 | 9.94±2.0 | 10.47±1.0 | 9.43±0.0 | 13.68±0.0 |
| | | FNR | Grp. 0 | 42.12±3.0 | 40.93±1.0 | 42.98±3.0 | 43.03±5.0 | 37.98±3.0 | 39.11±4.0 | 43.78±2.0 | 43.39±2.0 | 39.63±4.0 | 39.58±1.0 | 39.00±1.0 |
| | | | Grp. 1 | 43.30±3.0 | 41.58±0.0 | 44.26±3.0 | 43.99±6.0 | 39.95±2.0 | 40.25±5.0 | 46.54±2.0 | 45.88±4.0 | 40.42±5.0 | 41.08±1.0 | 36.97±1.0 |
| | | EqOdd | - | 99.24±0.0 | 99.15±0.0 | 99.23±0.0 | 99.34±0.0 | 98.90±1.0 | 99.25±1.0 | 98.43±0.0 | 98.65±1.0 | 99.32±0.0 | 99.01±0.0 | 98.70±1.0 |

Table A15: Results of other metrics in out-of-distribution setting. In the dataset column, the dataset in the first row is the training domain, and the second row is the testing domain.

| Dataset | Attr. | Metrics | Group | ERM | Resample | DomainInd | LAFTR | CFair | LNL | EnD | ODR | GroupDRO | SWAD | SAM |
|---|---|---|---|---|---|---|---|---|---|---|---|---|---|---|
| ADNI 1.5T ADNI 3T | Age | BCE | Grp. 0 | 0.68±0.30 | 0.72±0.30 | 1.25±0.51 | 1.05±0.57 | 0.66±0.06 | 0.64±0.26 | 0.65±0.01 | 0.60±0.14 | 0.81±0.22 | 0.51±0.12 | 0.72±0.03 |
| | | | Grp. 1 | 0.44±0.11 | 0.48±0.14 | 0.49±0.13 | 0.56±0.13 | 0.51±0.09 | 0.34±0.04 | 0.58±0.04 | 0.44±0.06 | 0.54±0.08 | 0.56±0.17 | 0.72±0.09 |
| | | ECE | Grp. 0 | 0.23±0.08 | 0.24±0.07 | 0.29±0.05 | 0.26±0.05 | 0.19±0.05 | 0.21±0.06 | 0.22±0.10 | 0.22±0.04 | 0.26±0.04 | 0.16±0.03 | 0.19±0.02 |
| | | | Grp. 1 | 0.16±0.03 | 0.16±0.05 | 0.14±0.04 | 0.20±0.04 | 0.16±0.05 | 0.14±0.01 | 0.13±0.03 | 0.14±0.05 | 0.16±0.03 | 0.13±0.01 | 0.25±0.11 |
| | | TPR@80 | Grp. 0 | 88.88±5.0 | 79.80±12.0 | 72.16±10.0 | 66.48±11.0 | 55.28±20.0 | 78.69±17.0 | 55.28±17.0 | 77.96±8.0 | 68.08±23.0 | 78.17±9.0 | 31.94±10.0 |
| | | TNR | Grp. 1 | 81.89±6.0 | 82.21±7.0 | 80.21±7.0 | 68.75±21.0 | 68.24±16.0 | 92.10±8.0 | 60.00±3.0 | 77.59±1.0 | 69.63±13.0 | 84.70±13.0 | 26.11±12.0 |
| | | FPR | Grp. 0 | 16.67±9.0 | 14.94±7.0 | 13.79±8.0 | 17.82±13.0 | 32.76±30.0 | 11.49±2.0 | 39.08±24.0 | 18.39±17.0 | 18.96±15.0 | 9.19±5.0 | 81.61±17.0 |
| | | | Grp. 1 | 28.50±9.0 | 30.11±13.0 | 27.96±7.0 | 38.71±19.0 | 41.94±27.0 | 17.20±7.0 | 55.91±29.0 | 29.03±12.0 | 43.01±26.0 | 11.83±2.0 | 92.47±5.0 |
| | | FNR | Grp. 0 | 15.97±8.0 | 22.92±9.0 | 36.81±10.0 | 33.33±16.0 | 25.00±15.0 | 34.72±17.0 | 15.28±13.0 | 23.61±17.0 | 28.48±8.0 | 31.95±5.0 | 6.95±5.0 |
| | | | Grp. 1 | 5.56±0.0 | 5.56±0.0 | 3.70±5.0 | 9.26±7.0 | 10.18±13.0 | 3.71±3.0 | 14.82±21.0 | 12.97±13.0 | 7.41±6.0 | 14.82±3.0 | 3.71±3.0 |
| | | EqOdd | - | 88.12±5.0 | 83.74±7.0 | 76.37±8.0 | 77.52±10.0 | 83.30±8.0 | 80.26±6.0 | 88.11±7.0 | 87.00±13.0 | 77.44±10.0 | 88.75±3.0 | 92.48±8.0 |
| | Sex | BCE | Grp. 0 | 0.62±0.14 | 0.55±0.14 | 0.84±0.25 | 0.76±0.20 | 0.59±0.22 | 0.66±0.06 | 0.59±0.03 | 1.18±0.38 | 0.52±0.09 | 0.55±0.15 | 0.71±0.04 |
| | | | Grp. 1 | 0.63±0.14 | 0.50±0.10 | 1.00±0.48 | 0.70±0.20 | 0.66±0.24 | 0.66±0.04 | 0.54±0.01 | 1.16±0.42 | 0.53±0.10 | 0.59±0.06 | 0.73±0.05 |
| | | ECE | Grp. 0 | 0.21±0.03 | 0.22±0.03 | 0.27±0.07 | 0.27±0.06 | 0.21±0.05 | 0.21±0.08 | 0.11±0.02 | 0.28±0.07 | 0.19±0.05 | 0.16±0.04 | 0.21±0.04 |
| | | | Grp. 1 | 0.19±0.04 | 0.17±0.05 | 0.23±0.04 | 0.20±0.05 | 0.20±0.05 | 0.15±0.05 | 0.12±0.02 | 0.27±0.07 | 0.14±0.04 | 0.15±0.02 | 0.18±0.04 |
| | | TPR@80 | Grp. 0 | 59.78±8.0 | 69.99±14.0 | 54.89±18.0 | 51.67±11.0 | 64.66±23.0 | 30.44±19.0 | 60.86±12.0 | 80.07±10.0 | 62.22±21.0 | 75.72±15.0 | 26.89±9.0 |
| | | TNR | Grp. 1 | 71.23±11.0 | 75.58±8.0 | 67.33±14.0 | 62.35±16.0 | 64.69±10.0 | 40.41±19.0 | 74.81±10.0 | 80.66±5.0 | 78.27±12.0 | 77.17±4.0 | 17.65±5.0 |
| | | FPR | Grp. 0 | 28.03±11.0 | 24.24±9.0 | 38.63±9.0 | 42.42±11.0 | 32.58±14.0 | 60.60±32.0 | 19.70±26.0 | 28.79±16.0 | 26.51±10.0 | 19.70±5.0 | 87.88±13.0 |
| | | | Grp. 1 | 28.51±9.0 | 16.23±11.0 | 34.21±22.0 | 34.65±14.0 | 26.76±14.0 | 59.21±26.0 | 16.67±24.0 | 14.91±5.0 | 21.05±6.0 | 17.55±8.0 | 86.85±9.0 |
| | | FNR | Grp. 0 | 8.89±11.0 | 20.00±20.0 | 3.33±6.0 | 7.78±8.0 | 6.66±7.0 | 12.22±24.0 | 33.33±29.0 | 11.11±8.0 | 12.22±11.0 | 22.22±15.0 | 0.00±0.0 |
| | | | Grp. 1 | 19.13±12.0 | 21.60±11.0 | 14.81±4.0 | 17.90±7.0 | 23.46±11.0 | 22.22±18.0 | 20.99±12.0 | 19.75±9.0 | 19.13±7.0 | 22.22±7.0 | 8.64±6.0 |
| | | EqOdd | - | 89.40±5.0 | 90.89±5.0 | 87.66±5.0 | 89.33±7.0 | 86.49±6.0 | 88.02±7.0 | 89.84±3.0 | 88.74±4.0 | 89.93±3.0 | 93.55±4.0 | 93.96±3.0 |

## D  INCORPORATE NEW DATASETS AND ALGORITHMS IN MEDFAIR

We implement MEDFAIR using the PyTorch framework. We show example pseudo codes to demonstrate how to incorporate new datasets and algorithms. Detailed documentation can be found at https://ys-zong.github.io/MEDFAIR/.

### D.1  ADDING NEW DATASETS

We implement a base dataset class BaseDataset, and a new dataset can be added by creating a new file and inheriting it.

```python
from datasets.BaseDataset import BaseDataset
class DatasetX(BaseDataset):
    def __init__(self, metadata, path_to_images, sens_name, sens_classes,
                                    transform):
        super(DatasetX, self).__init__(metadata, sens_name, sens_classes,
                                    transform)

    def __getitem__(self, idx):
        item = self.metadata.iloc[idx]
        img = Image.open(path_to_images[idx])

        # apply image transform/augmentation
        img = self.transform(img)
        label = torch.FloatTensor([int(item['binaryLabel'])])

        # convert to sensitive attributes to numerical values
        sensitive = self.get_sensitive(self.sens_name, self.sens_classes,
                                    item)

        return img, label, sensitive, idx
```

### D.2  ADDING NEW ALGORITHM

We implement a base algorithm class BaseNet, which contains basic configuration and regular training/validation/testing loop. A new algorithm can be added by inheriting it and rewriting the training loop, loss, etc. if needed.

For example, SAM (Foret et al., 2020) algorithm can be added by re-implementing the training loop.

```python
class SAM(BaseNet):
    def __init__(self, opt):
        super(SAM, self).__init__(opt)
        def _train(self, loader):
            self.network.train()
            for i, (images, targets, sensitive_attr, index) in
                                            enumerate(loader):

                enable_running_stats(self.network)
                outputs, features = self.network(images)

                loss = self._criterion(outputs, targets)
                loss.mean().backward()
                self.optimizer.first_step(zero_grad=True)
                self.scheduler.step()

                disable_running_stats(self.network)
                outputs, features = self.network(images)
                self._criterion(outputs, targets).mean().backward()
                self.optimizer.second_step(zero_grad=True)
                self.scheduler.step()
```

# E   ANALYSIS OF SOURCE OF BIAS

We take two datasets HAM10000 and MIMIC-CXR as case studies to analyze the source of bias. As shown in Table A2, we can observe severe data and class imbalance as the direct sources of bias for both datasets. Thus, we utilize resampling strategies to explicitly mitigate the imbalance. We use three types of resampling to upsample the minority subgroup, class, or both subgroup and class so that all groups appear with equal chances during training, i.e. subgroup resampling, class resampling, and subgroup and class resampling, respectively.

For the HAM10000 dataset, we take the cartesian product of age and sex subgroups to construct 8 intersectional subgroups, whose statistics are shown in Table A16 (excluding age 0-20 as it has too few samples). The results of the ERM, resampling strategies, and the best-performing SWAD method are shown in Table A17. The worst-performing subgroup is "40-60 Male", which neither contains the least number of images nor not the most class-imbalanced subgroup. Also, it can be seen that for intersectional subgroups, different resampling strategies produce a similar performance as the ERM and sometimes even worse, where all of them are worse than SWAD. In other words, there are other sources of bias beyond the observable data/class imbalance, which explains why resampling methods do not lead to better performance. For example, the skin type can be a potential source of bias (not recorded by metadata) as it is a dermatology dataset, and lesions on darker skin are intrinsically more difficult to diagnose than that on lighter skin. Overall, they are difficult or impossible to disentangle given the metadata as the sensitive attributes of different datasets can be correlated in different ways. Therefore, methods specifically optimizing for one particular factor are usually less effective.

Table A16: The statistics of the intersectional subgroup HAM10000 dataset. M. represents Male and F. represents female.

| Subgroup | 20-40 M. | 40-60 M. | 60-80 M. | 80+ M. | 20-40 F. | 40-60 F. | 60-80 F. | 80+ F. |
|---|---|---|---|---|---|---|---|---|
| **% Images** | 7.27% | 9.52% | 23.09% | 23.05% | 18.97% | 10.96% | 5.16% | 1.99% |
| **% Malignant** | 4.53% | 6.60% | 10.93% | 8.09% | 25.75% | 20.77% | 31.14% | 34.72% |

Table A17: The performance of different methods on HAM10000 dataset.

| Attr. | Metric | ERM | Subgroup-R | Class-R | Subgroup_class-R | SWAD |
|---|---|---|---|---|---|---|
| **Sex** | Overall | 87.24 | 88.03 | 87.89 | 88.76 | **91.46** |
| | Min. | 86.48 | 86.23 | 87.42 | 88.11 | **91.14** |
| | Gap | 1.59 | 4.43 | 1.13 | 1.28 | **0.54** |
| **Age** | Overall | **90.00** | 89.72 | 87.38 | 88.60 | 89.61 |
| | Min. | 77.53 | 82.73 | 79.41 | 85.02 | **86.66** |
| | Gap | 14.72 | 9.68 | 8.85 | 5.52 | **5.72** |
| **Intersection** | Overall | 88.13 | 88.03 | 88.78 | 87.75 | **89.32** |
| | Min. | 78.60 | 69.33 | 79.59 | 75.12 | **81.25** |
| | Gap | 17.23 | 24.02 | 17.29 | 18.43 | **15.02** |

Following a similar procedure, we further analyze the intersectional subgroup of MIMIC-CXR dataset, where we obtain 20 subgroups by taking the cartesian product of the age, sex, and race subgroups. As it can be seen from Table A18, the number of images and the value of labels ("No Finding") vary greatly among different subgroups. The worst-performing subgroup is the "60-80 non-White Female", which is imbalanced in data and class, but also does not contain the least number of images nor not the most class-imbalanced subgroup.

From the results of Table A19, we do observe a slight performance increase of the worst-case subgroup after resampling the minority subgroup, class, or both, while there are also decreases in the overall performance. A possible reason is that the label noises in some subgroups are more severe

than the other subgroups as identified by Zhang et al. (2022), and the upsampled subgroups may contain more noisy labels to worsen the overall performance.

To summarize, most of the medical imaging datasets contain multiple sources of bias instead of only one, where they are correlated in different ways and it is difficult or impossible to fully disentangle for separate analyses. The mixture of multiple confounders makes the algorithms that specifically optimize for one particular factor (e.g. data imbalance) fail to succeed, i.e. do not outperforms ERM. This, to some extent, explains why the domain generalization method SWAD is the most consistently high-ranked algorithm as it does not have a specific assumption of the source of bias. Increasing the general notion of robustness may be beneficial to various kinds of confounders.

Table A18: The statistics of the intersectional subgroup MIMIC-CXR dataset.

| Subgroup | % Images | % "No Finding" |
|---|---|---|
| 0-20 non-White Female | 0.12% | 57.48% |
| 20-40 non-White Female | 0.22% | 72.18% |
| 40-60 non-White Female | 0.11% | 81.27% |
| 60-80 non-White Female | 0.3% | 80.72% |
| 80+ non-White Female | 2.50% | 50.69% |
| 0-20 White Female | 4.01% | 62.38% |
| 20-40 White Female | 2.58% | 59.98% |
| 40-60 White Female | 4.22% | 73.8% |
| 60-80 White Female | 10.18% | 40.51% |
| 80+ White Female | 6.75% | 47.58% |
| 0-20 non-White Male | 7.03% | 44.91% |
| 20-40 non-White Male | 7.17% | 55.74% |
| 40-60 non-White Male | 15.27% | 29.67% |
| 60-80 non-White Male | 6.12% | 34.17% |
| 80+ White Male | 11.35% | 32.43% |
| 0-20 White Male | 6.88% | 38.9% |
| 20-40 White Male | 5.41% | 22.04% |
| 40-60 White Male | 1.60% | 21.54% |
| 60-80 White Male | 6.03% | 23.64% |
| 80+ White Male | 2.17% | 27.14% |

Table A19: The performance of different methods on MIMIC-CXR dataset.

| Attr. | Metric | ERM | Subgroup-R | Class-R | Subgroup_class-R | SWAD |
|---|---|---|---|---|---|---|
| **Race** | Overall | 86.26 | 86.07 | 86.20 | 86.33 | **87.10** |
| | Min. | 85.52 | 85.31 | 85.46 | 85.56 | **86.38** |
| | Gap | **0.85** | 0.88 | 0.86 | 0.95 | 0.88 |
| **Sex** | Overall | 86.45 | 86.21 | 86.37 | 86.24 | **87.05** |
| | Min. | 85.62 | 85.31 | 85.45 | 85.39 | **86.23** |
| | Gap | 1.41 | 1.53 | 1.60 | 1.46 | **1.39** |
| **Age** | Overall | 86.40 | 85.53 | 86.02 | 85.43 | **87.08** |
| | Min. | 81.06 | 70.97 | 74.69 | 73.24 | **82.15** |
| | Gap | 5.32 | 6.99 | 5.87 | 6.08 | **5.10** |
| **Intersection** | Overall | 86.26 | 85.06 | 86.25 | 85.37 | **86.35** |
| | Min. | 72.43 | 74.31 | 73.31 | 74.18 | **75.96** |
| | Gap | 18.92 | **11.59** | 16.06 | 15.45 | 14.78 |

## F   ACKNOWLEDGEMENT TO DATA RESOURCES

For the usage of ADNI dataset:

Data used in preparation of this article were obtained from the Alzheimer's Disease Neuroimaging Initiative (ADNI) database (adni.loni.usc.edu). As such, the investigators within the ADNI contributed to the design and implementation of ADNI and/or provided data but did not participate in analysis or writing of this report. A complete listing of ADNI investigators can be found at: http://adni.loni.usc.edu/wp-content/uploads/how_to_apply/ADNI_Acknowledgement_List.pdf

The ADNI was launched in 2003 as a public-private partnership, led by Principal Investigator Michael W. Weiner,MD. The primary goal of ADNI has been to test whether serial magnetic resonance imaging (MRI), positron emission tomography (PET), other biological markers, and clinical and neuropsychological assessment can be combined to measure the progression of mild cognitive impairment (MCI) and early Alzheimer's disease (AD). For up-to-date information, see www.adni-info.org.

Data collection and sharing for this project was funded by the Alzheimer's Disease Neuroimaging Initiative (ADNI) (National Institutes of Health Grant U01 AG024904) and DOD ADNI (Department of Defense award number W81XWH-12-2-0012). ADNI is funded by the National Institute on Aging, the National Institute of Biomedical Imaging and Bioengineering, and through generous contributions from the following: AbbVie, Alzheimer's Association; Alzheimer's Drug Discovery Foundation; Araclon Biotech; BioClinica, Inc.; Biogen; Bristol-Myers Squibb Company; CereSpir, Inc.; Cogstate; Eisai Inc.; Elan Pharmaceuticals, Inc.; Eli Lilly and Company; EuroImmun; F. Hoffmann-La Roche Ltd and its affiliated company Genentech, Inc.; Fujirebio; GE Healthcare; IXICO Ltd.;Janssen Alzheimer Immunotherapy Research & Development, LLC.; Johnson & Johnson Pharmaceutical Research & Development LLC.; Lumosity; Lundbeck; Merck & Co., Inc.;Meso Scale Diagnostics, LLC.; NeuroRx Research; Neurotrack Technologies; Novartis Pharmaceuticals Corporation; Pfizer Inc.; Piramal Imaging; Servier; Takeda Pharmaceutical Company; and Transition Therapeutics. The Canadian Institutes of Health Research is providing funds to support ADNI clinical sites in Canada. Private sector contributions are facilitated by the Foundation for the National Institutes of Health (www.fnih.org). The grantee organization is the Northern California Institute for Research and Education, and the study is coordinated by the Alzheimer's Therapeutic Research Institute at the University of Southern California. ADNI data are disseminated by the Laboratory for Neuro Imaging at the University of Southern California.

