# OpenReview forum: "MEDFAIR: Benchmarking Fairness for Medical Imaging"
_ICLR.cc/2023/Conference — ICLR 2023 notable top 25%_

### Official Review · Reviewer_sgSV · 2022-10-21

**Confidence:** 4
**Correctness:** 3
**Technical Novelty And Significance:** 2
**Empirical Novelty And Significance:** 3
**Recommendation:** 6

**Clarity, Quality, Novelty And Reproducibility:**

Clarity: The writing of this paper needs further improvement.

Quality: High and easy to follow.

Novelty: High.

Reproducibility: High.

**Strength And Weaknesses:**

The topic of this paper is significant and valuable. But there are many issues.

1. Page 1 (bottom), "label noise (e.g., Zhang et al. (2022) find that label noise in...".
2. Figure 1, why is this figure not mentioned in the text?
3. Page 2 (bottom), "(E.g., data collected at hospital A is used to train a model deployed at hospital B). " I have never seen it written like this before.
4. Page 2 (bottom) "Bias widely exists in ERM models... " and Page 3 (top) "...the empirical risk minimization (ERM)
with statistical significance...",  why does the latter ERM have the full name and not the preceding one?
5. Page 3 (middle), "...e.g., there are data of patients’ age and sex. ", statements are not concise enough.
6. Page 3 (bottom), "Our framework can be easily extended to non-binary classification." There is no explanation in the full text or even in the supplementary material on how to extend to non-dichotomous classification.
7. Page 3 (middle), "We train models for each combination of algorithms × datasets × sensitive attributes." What role does the model selection in Section 3.3 play?
8. Section 3.2 lacks the reasons for choosing these algorithms.




**Summary Of The Paper:**

This paper propose a framework, named MEDFAIR, to benchmark the fairness of machine learning models for medical imaging. Through extensive experiments, the authors find that the under-studied issue of model selection criterion can have a significant impact on fairness outcomes. The recommendations are made for different medical application scenarios.

**Summary Of The Review:**

As an empirical evaluation paper, this paper has conducted a lot of experiments and obtained some valuable conclusions, but the writing needs to be improved.

---

> ### Author Response · Authors · 2022-11-16
> **Response (1/2)**
>
> Thank you very much for reviewing our paper and providing helpful comments.
>
> ## **General Response Re: Lack of novelty**
>
>
> As a general response, we would like to highlight that as an empirical evaluation paper, our contributions do not lie in the novelty of proposing a new method, but (1) in the comprehensiveness and thoroughness of our benchmark, and (2) the significance of the new conclusions that can be drawn from that evaluation. We would kindly ask you to re-evaluate our paper from this perspective. As mentioned in the ICLR reviewer guide (https://iclr.cc/Conferences/2023/ReviewerGuide), experimental rigour and new empirical knowledge are considered valuable as well as novel algorithms.
>
> To recap our contributions: We (1) reveal the novel empirical insight that model selection strategies can significantly impact max-min fairness. This has not been previously highlighted, but it is immensely valuable knowledge for practitioners due to its ease of implementation compared to specialized fairness interventions, (2) We conduct the largest-scale study to date of fairness interventions in medical imaging, showing rigorously that no method in the substantial set evaluated outperforms ERM. This is an important and novel empirical insight, because going on prior published work alone practitioners and clinicians might inappropriately rely on existing published algorithms for fairness and inadvertently perpetuate socially unfair outcomes. (See also Reviewer q4yd Q2). (3) We provide a major benchmark platform to improve the quality of future research in this area by enabling future researchers to conveniently evaluate algorithms on a large number of datasets to achieve more reliable conclusions, and conveniently control for model selection strategy. This is in contrast to the status quo today, where algorithms cannot be reliably compared due to inconsistent model selection and the use of too few datasets to draw reliable conclusions.
>
> We emphasise that there is extensive of precedent for such rigorous benchmark work to be published at ICLR, demonstrating the value of such contributions to the community e.g. [A,B,C,D].
>
> [A] Gulrajani et al, ICLR'21, In serach of lost Domain Generalization.
>
> [B] Toneva et al, ICLR'19, An Empirical Study of Example Forgetting during Deep Neural Network Learning.
>
> [C] Triantafillou et al, ICLR'20, Meta-Dataset: A Dataset of Datasets for Learning to Learn from Few Examples.
>
> [D] Chen et al, ICLR'19, A Closer Look at Few-shot Classification.
>
> ## **Specific Concerns**
>
> We have made changes to our paper and provided our response to your specific concerns below.
> - Thank you for pointing this out. We have corrected the grammar: "label noise'' $\rightarrow$ "label noises''.
> - Figure 1. Thanks for pointing that out. We added the reference to Figure 1 in our revised version (last paragraph on Page 2).
> - Through this example, we try to convey that a trained model may be deployed in different institutes due to the limited availability of data, patient privacy, etc., where there might be a domain shift. And it is essential to evaluate whether the fairness achieved in in-domain holds under the shift, thus identifying the necessity of the out-of-distribution experiments. We have modified our example to: ``A model trained on the data from hospital A may be also deployed in hospital B.'' This is common practice for medical data as identified by many studies [1][2][3]. It is also a scenario that machine learning communities try to solve, such as domain adaptation and domain generalization.
> - Thanks for pointing out. We have deleted the full name in the latter one, as we have identified the full name of ERM in the abstract.
> - Through this example, we want to explain our experimental settings in that we only evaluate one sensitive attribute at a time. For example, we have the metadata of both the age and sex of patients. In experiment A, we evaluate the fairness regarding patients' age. And in experiment B, we evaluate the fairness regarding patients' sex. We reworded it to "e.g. there are metadata of both patients' age and sex''.
> -  Basically, only two simple steps are required to change to a non-binary classification problem. 1) Change the loss function from binary cross entropy (BCE) to cross entropy (CE). 2) Set the output layer to the number of classes instead of 1. The changes do not affect the core idea of algorithms and can be modified by simple engineering. We have added concrete steps in our documentation (supplementary materials).
> - For each combination of (algorithms × datasets × sensitive attributes) (e.g. CheXpert, ERM, Sex), we consider multiple hyper-parameters and checkpoints. Model selection consists of choosing the best hyper-parameter and checkpoint combination according to a given validation metric. Model selection is conducted separately for each combination algorithm-dataset-attribute.

---

> > ### Author Response · Authors · 2022-11-16
> > **Response (2/2)**
> >
> > **Continuing specific concerns**
> >
> > - We first review widely-used in-processing bias mitigation algorithms of several categories in related work (Detailed in Appendix Sec A2): subgroup rebalancing, domain-independence, adversarial training, disentanglement, and select typical algorithms from each category. We also select methods from the area of domain generalization, which is related to fairness because their common goal is to be robust to distribution changes across different sub-population (discussion in Appendix A.5). All the selected algorithms have official open-source code, which can ensure that our re-implementation is as close as the original ones. We have added more discussion about the choices of methods in section 3.2.
> >
> > We would love to hear from you if you have any further concerns. Thank you!
> >
> > Ref:
> >
> > [1] Zhang, Angela, et al. ``Shifting machine learning for healthcare from development to deployment and from models to data.'' Nature Biomedical Engineering (2022): 1-16.
> >
> > [2] Wiens, Jenna, et al. ``Do no harm: a roadmap for responsible machine learning for health care.'' Nature medicine 25.9 (2019): 1337-1340.
> >
> > [3] Finlayson, Samuel G., et al. ``The clinician and dataset shift in artificial intelligence.'' The New England journal of medicine 385.3 (2021): 283.

---

### Official Review · Reviewer_q4yd · 2022-10-23

**Confidence:** 5
**Correctness:** 3
**Technical Novelty And Significance:** 2
**Empirical Novelty And Significance:** 2
**Recommendation:** 5

**Clarity, Quality, Novelty And Reproducibility:**

> All of the datasets we use are publicly available

I am not aware that the CheXpert and MIMIC datasets have race publicly available. Was this information privately obtained? Can you add a breakdown of these subgroups in the paper?

In the discussion the authors mention sources of bias include "subgroup size, imbalance in subgroup disease prevalence, difference in imaging protocols/time/location, spurious correlations" but these are not included in the evaluation or explored in this work.

The experiments don't appear to have multiple trials so it is hard to draw conclusions from these experiments, even negative conclusions.


**Strength And Weaknesses:**

The paper is well written and articulates existing methods of fairness very well.

Details about the experiments are not clear. Such as how the age was divided into subgroups and how many subgroups exist for each category are not mentioned. Knowing the number of subgroups where the min and the max performance are calculated would help to interpret the significance of the spread in Figure 3.

The main weakness is that there is not much of an actionable conclusion. The experiments contain runs of models on many datasets without a deep focus on any one of them, not focusing on the specifics of each one. The breakdown of biased variables is very limited to sex and age which doesn't really expose subtle issues.

I find the paper spends more time discussing fairness concepts than studying it on medical data. The conclusions drawn from the experiments on medical data seem limited and not detailed enough to reflect deeply on the specifics of each domain.


**Summary Of The Paper:**

The paper performs an analysis of fairness training methods on multiple medical imaging tasks. They conclude that no method is significantly better than basic ERM training.


**Summary Of The Review:**

The paper is well written and clearly discusses fairness but does not provide significant insight into the fairness issues of medical datasets. For this reason I do not recommend acceptance.

---

> ### Author Response · Authors · 2022-11-16
> **Response (1/3)**
>
> Thank you very much for the thoughtful review and excellent questions. Here are our responses to your comments.
>
> ### **Q1. Details about the division of subgroups**
>
> Different ages are divided into subgroups of a certain range, e.g. 20-40, 40-60, 60-80, and 80+. It depends on the specific datasets and the number of samples in the datasets (ensuring each subgroup contains enough samples). Statistics of all subgroups of all datasets can be found in Appendix Table A2-A5. We also report the specific value of minimum performance and the gap between minimum and maximum performance in Appendix Table A8 and A9. Section 3.1 of the manuscript points the reader to these details in the appendix.
>
> ### **Q2. Clarify the actionable conclusions**
>
> From our experiments, we make two sets of actionable conclusions for both practitioners and researchers.
>
> - For practitioners/clinicians.
>   - The empirical results show that state-of-the-art fairness interventions are not consistently effective, which is contrary to the claims made by individual fairness intervention papers, each of which is based on more limited experiments. This is an important outcome for clinicians who need to know that they cannot rely on such interventions to solve fairness issues in general. Without this knowledge, clinicians are at risk of inadvertently allowing biased algorithms to perpetuate socially unfair diagnosis outcomes.
>    - That said, the empirical results also advise practitioners that among the existing algorithms, SWAD is perhaps the best to improve outcomes among the imperfect set of options.
>    - Model selection. Our analysis shows that model selection indeed statistically significantly impacts Max-Min fairness. This is valuable knowledge as it is easy to implement and does not require practitioners to invest in implementing complex and specialized fairness intervention algorithms. This is a novel outcome of our analysis that has not been discussed before in the fairness literature.
>
> - For researchers.
>   - Previous fairness works tend to claim their methods work better than the others based on the results of a few selected datasets and potentially inconsistent hyperparameter tuning across methods. Our extensive experiments from a much larger number of datasets show that though some methods do perform slightly better, they do not have a statistically significant improvement on ERM. This ``negative'' finding is an actionable conclusion that future fairness studies need to include more datasets to draw statistically significant general conclusions, and that they need to evaluate in settings where all competitors have common hyperparameter tuning. Going beyond a mere actionable conclusion, we further contribute a major actionable *tool*: Our codebase will allow future fairness intervention algorithms to be directly tested on a much larger number of datasets than is used by fairness studied in the literature today, given the difficulty of collating and standardising lots of datasets prior to this paper. This will enable higher quality future research in the fairness community by enabling more reliable statistical comparisons across algorithms.
>   - Our "positive'' finding that model selection strategy can statistically significantly impact outcomes is also an actionable conclusion, because it means that algorithm papers must explicitly report on this experimental condition (which is not standard practice today!) in order to make meaningful comparisons. It also highlights novel model selection strategies as a target for future research that's orthogonal to the common learning algorithm focus today. Our codebase again provides a tool to action these insights by providing a properly engineered library that standardizes model selection across algorithms.
>
> ### **Q3. More sensitive attributes?**
>
> Our sensitive attributes are not limited to age and sex. We also include race and skin type as the sensitive attributes, which cover most of the common sensitive attributes in clinical scenarios.
>
> ## Clarity, Quality, Novelty And Reproducibility
>
>
> ### **Q4. Availability of race metadata.**
> For CheXpert, the race data is obtained from the public Stanford AIMI database (https://stanfordaimi.azurewebsites.net/datasets/192ada7c-4d43-466e-b8bb-b81992bb80cf) reported in paper [1], and incorporated to the original metadata based on the patient ID. For the MIMIC-CXR, the race data is available via MIMIC-IV [2] dataset, which is also deposited in the PhysioNet database. We merge it to the original metadata based on ``subject ID''. We have added the relevant details in Appendix B.1.2. The breakdown (number of samples and number of classes in each subgroup) of the subgroups is available in Appendix Table A2.
>
> [1] Gichoya, Judy Wawira, et al. "AI recognition of patient race in medical imaging: a modelling study.'' The Lancet Digital Health (2022).
>
> [2] Johnson, A. et al. MIMIC-IV (version 0.4). PhysioNet https://doi.org/10.13026/a3wn-hq05 (2020).

---

> > ### Author Response · Authors · 2022-11-16
> > **Response (2/3)**
> >
> > ### **Q5. Sources of bias**
> >
> > We discuss the *potential* sources of bias at the end of our paper in order to try to provide potential explanations for the failure of the bias mitigation algorithms to outperform ERM -- not because we claim to have completely analysed all bias sources. This is impossible as several potential sources of bias, such as differences in imaging time/protocols/location, would require meta-data that is not generally available, and thus can not be analysed systematically given available data.
> >
> > To recap the analysis that we do have, we find that 1) most of the medical imaging datasets contain multiple sources of bias instead of only one (Table 1), where they are correlated in different ways and it is difficult or impossible to fully disentangle them for separate analysis. The mixture of multiple confounders makes algorithms that specifically optimize for one particular factor (e.g. data imbalance) fail to outperform ERM. 2) This, to some extent, explains why the domain generalization method SWAD is the most consistently high-ranked algorithm as it does not have a specific assumption of the source of bias. Increasing general model robustness may be beneficial to various kinds of confounders. Through the analysis and our empirical results, we suggest the potential research direction for fairness via optimizing robustness.
> >
> > However, we can address your concern by better analysing the intersection of available sensitive attributes. We tried different resampling strategies as a direct approach to mitigate the cause of the bias demonstrated in the metadata. The results of the experiments further confirm the conjecture we made in the discussion section from the observation of our meta-analysis (statistical results): the bias is caused by a mixture of confounding factors, which is difficult to directly disentangle. Thus, algorithms that specifically optimize for one particular factor often fail to succeed. Below are our concrete experiment setting and results. Further details can be found in Appendix E.
> >
> > 1) We show the further breakdown of the demographics of the intersectional subgroups below (also in Appendix Table A17) by taking the cartesian-product combination of age and sex subgroup. Compared with the statistics of separate age or sex subgroups (Appendix Table A2), it can be seen that the more fine-grained subgroups are even more imbalanced in terms of the number of images and disease prevalence among subgroups.
> >
> > | **Subgroup**       |  **20-40 M.** | **40-60 M.** | **60-80 M.** |  **80+ M.** |  **20-40 F.** |  **40-60 F.** | **60-80 F.** | **80+ F.** |
> > |-------------|-----------|---------|----------|-------|------------|--------|----------|--------|
> > | **\% Images**      | 7.27\%    | 9.52\%  | 23.09\%        | 23.05\%      | 18.97\%        | 10.96\%        | 5.16\%         | 1.99\%       |
> > | **\% Malignant** | 4.53\%     | 6.60\%    | 10.93\%        | 8.09\%       | 25.75\%        | 20.77\%        | 31.14\%        | 34.72\%      |
> >
> > 2) Therefore, we use three different resampling strategies to balance the appearance of the images and disease prevalence during training to explicitly mitigate the cause of bias, which are subgroup resampling (``resampling'' we used in the main text), class resampling (balance the disease prevalence), and subgroup-class resampling (balance both subgroup and disease prevalence). The results of them together with ERM and SWAD are shown below.
> >
> > | **Attr.**          | **Metric** | **ERM**   | **Subgroup-R** | **Class-R** | **Subgroup\_class-R** | **SWAD**  |
> > |---------------|----------|------|-----------|-------------|-------------------|----------------|
> > |                | Overall         | 87.24          | 88.03               | 87.89            | 88.76        | **91.46** |
> > | **Sex**            | Min.            | 86.48          | 86.23               | 87.42      | 88.11    | **91.14** |
> > |                         | Gap             | 1.59           | 4.43                | 1.13         | 1.28          | **0.54**  |
> > |                         | Overall         | **90.00** | 89.72               | 87.38            | 88.60           | 89.61          |
> > | **Age**            | Min.            | 77.53          | 82.73               | 79.41            | 85.02     | **86.66** |
> > |             | Gap             | 14.72          | 9.68                | 8.85             | 5.52       | **5.72**  |
> > |           | Overall         | 88.13          | 88.03               | 88.78            | 87.75            | **89.32** |
> > | **Intersection** | Min.      | 78.60          | 69.33               | 79.59            | 75.12         | **81.25** |
> > |                         | Gap      | 17.23          | 24.02               | 17.29            | 18.43       | **15.02** |
> >
> > To be continued.

---

> > > ### Author Response · Authors · 2022-11-16
> > > **Response (3/3)**
> > >
> > > **Continuing Q5**
> > >
> > > 3) As can be seen from the Table above, the performance of different resampling is similar to ERM (slightly better or worse), but all lower than the best-performing SWAD. It indicates that the bias is not simply caused by the data or class imbalance. Also, the worst-performing subgroup is ``40-60 Male'', which neither contains the least number of images nor not the most class-imbalanced subgroup. In other words, there are other sources of bias beyond the observable data/class imbalance. For example, the label noises in some subgroups are more severe than the other subgroups where we resample the noisy group because they contain fewer samples, which may lead to an even worse performance. Also, the skin type can also be another potential source of bias (not recorded) as it is a dermatology dataset. Overall, they are difficult or impossible to disentangle given the metadata is not complete. Therefore, methods specifically optimizing for one particular factor are usually less effective. Additionally, we perform a similar experiment for MIMIC-CXR dataset, which indicates the same conclusion (Appendix E Table A19-20).
> > >
> > > 4) As revealed by our experiments and analysis, it is difficult to disentangle different sources of bias given available meta-data. That is why we believe it beneficial to provide a meta-analysis with rigorous statistical tests in order to give an overview of the aggregate effectiveness of the current bias mitigation algorithms among various datasets as an initial step for future research.
> > >
> > > ### **Q6. Experiments lack multiple trials?**
> > >
> > > - Our experiments do in fact have multiple trials, as already stated at the end of Sec 3. Specifically, we run each experiment using three different random seeds to reduce the impact of the randomness. We report the specific mean values and standard deviations of all experiments in Appendix Table A11-A16. The variance among different seeds is low enough that we can safely perform statistical tests by taking the ranks of the mean values.
> > > - Our experiments are conducted on the combination of algorithm $\times$ dataset $\times$ sensitive attribute. For each combination and seed, we perform extensive hyper-parameter searches to achieve optimal performance, resulting in over 7000 models trained.
> > > - Finally, our main effort in producing reliable results from which strong conclusions can be drawn was to include a much larger number of datasets and algorithms than prior work. This increases our dataset-wise sample size, which is more important for drawing general conclusions than the number of trials per dataset.
> > >
> > >
> > > We hope our responses address your concern and would love to hear from you if you have any further concerns. Thank you!

---

> ### Author Response · Authors · 2022-11-18
> **Sincerely expecting further discussions from Reviewer q4yd**
>
> Dear Reviewer q4yd,
>
> We want to thank you here, again, for the constructive comments and acknowledgment of this paper. We have conducted additional experiments and provided detailed explanations to address your concerns. Could you please kindly check our revised paper and our responses, to see if your concerns are solved? We would really like to hear if you have any further questions before the discussion window is over. And if no more questions, please could you consider updating the score?
>
> Sincerely,
>
> Authors

---

### Official Review · Reviewer_zqn8 · 2022-10-24

**Confidence:** 4
**Correctness:** 4
**Technical Novelty And Significance:** 3
**Empirical Novelty And Significance:** 3
**Recommendation:** 8

**Clarity, Quality, Novelty And Reproducibility:**

The paper is of good quality and should be easy to reproduce, and is an original work.

**Strength And Weaknesses:**

The main strength of this paper:
1. The problem setup is interesting and of significant importance. Fairness issue in medical image systems can be harmful when put into practical use;
2. The experiments are extensive and well-designed. The authors choose different datasets and algorithms to finish the benchmarking, which makes it more reliable;
3. The paper is well-written and easy to follow;
4. The results are significant and the limitations are clarified. In the results shown in the paper, there does exist bias issue in various tasks related to medical images and cannot be ignored.

Does not have to be weakness, but some concerns are:
1. For now this paper only considers one sensitive attribute at a time, I believe it's important to also understand the correlation within the sensitive attributes, since in most datasets that's also imbalanced. Therefore some future work may also include a multi-factor analysis, for example adding a regression;
2. Maybe I am missing it -- what is the diagnose label used in the experiments reported in the main body? Were other label/tasks also considered?

**Summary Of The Paper:**

In this paper, the authors focus on benchmarking the fairness issue in the field of medical images, taking different algorithms, datasets and models into consideration. They first choose 9 different datasets, covering the different categories of medical images, and then implement 11 different algorithms that aim to mitigate the bias/fairness issue in machine learning systems, and finally conduct model selection for evaluation. The main contributions and conclusions are:
1. By extensive experiments, they show that the bias issue widely exist in the area of medical images;
2. Model selection introduces significant difference in the performance for different subgroups;
3. Current bias mitigation algorithms fail to address the fairness issue in medical images

**Summary Of The Review:**

This paper provides a benchmarking of fairness in medical image systems, with thorough experiments and significant results, I recommend this paper to be accepted.

---

> ### Author Response · Authors · 2022-11-16
> **Response (1/2)**
>
> Thank you very much for the insightful review and valuable suggestions! Here are our responses to your comments.
>
> **Q1. Correlation within sensitive attributes.**
>
> Thank you for your perceptive suggestions. We agree that it is important to consider the correlation or intersection of the sensitive attributes. One important reason we only consider one sensitive attribute at a time is that most of the current algorithms are specifically designed for one sensitive attribute with some underlying assumptions for one sensitive attribute. Developing algorithms for intersectional subgroups is a promising direction for future research.
>
> To address your concern, we take MIMIC-CXR dataset as an example to analyze the correlation as it has the largest number of images and 20 subgroups if we take the cartesian product of race, age, and sex subgroups. A detailed breakdown of the intersectional subgroups is shown below and in Table A19 of the revised paper and below:
>
> |  **Subgroup**         | **% Images** | **% "No Finding"** |
> |------------------------|-----------------|-----------------------|
> | 0-20 non-White Female  | 0.12\%          | 57.48\%               |
> | 20-40 non-White Female | 0.22\%          | 72.18\%               |
> | 40-60 non-White Female | 0.11\%          | 81.27\%               |
> | 60-80 non-White Female | 0.3\%           | 80.72\%               |
> | 80+ non-White Female   | 2.50\%          | 50.69\%               |
> | 0-20 White Female      | 4.01\%          | 62.38\%               |
> | 20-40 White Female     | 2.58\%          | 59.98\%               |
> | 40-60 White Female     | 4.22\%          | 73.8\%                |
> | 60-80 White Female     | 10.18\%         | 40.51\%               |
> | 80+ White Female       | 6.75\%          | 47.58\%               |
> | 0-20 non-White Male    | 7.03\%          | 44.91\%               |
> | 20-40 non-White Male   | 7.17\%          | 55.74\%               |
> | 40-60 non-White Male   | 15.27\%         | 29.67\%               |
> | 60-80 non-White Male   | 6.12\%          | 34.17\%               |
> | 80+ White Male         | 11.35\%         | 32.43\%               |
> | 0-20 White Male        | 6.88\%          | 38.9\%                |
> | 20-40 White Male       | 5.41\%          | 22.04\%               |
> | 40-60 White Male       | 1.60\%          | 21.54\%               |
> | 60-80 White Male       | 6.03\%          | 23.64\%               |
> | 80+ White Male         | 2.17\%          | 27.14\%               |
>
> It can be seen that the number of images and the value of labels (”No Finding”) vary greatly among different subgroups. To understand whether directly balancing the number of images or the labels (disease prevalence) will be helpful to mitigate the bias, i.e. if different sensitive attributes are only correlated in terms of the number of images and labels, we utilize resampling strategies to explicitly mitigate the imbalance. We use three types of resampling to upsample the minority subgroup, class, or both subgroup and class so that all groups appear with equal probability during training. I.e. subgroup resampling, class resampling, and subgroup and class resampling, respectively. The results are shown in the table below.
>
> | **Attr.**          | **Metric** | **ERM**  | **Subgroup-R** | **Class-R** | **Subgroup\_class-R** | **SWAD** |
> |-----------|-----------------|---------------|---------------------|------------------|----------------------------|----------------|
> |                       | Overall         | 86.26         | 86.07    | 86.20            | 86.33        | **87.10** |
> | **Race**         | Min.            | 85.52         | 85.31     | 85.46            | 85.56          | **86.38** |
> |                       | Gap             | **0.85** | 0.88    | 0.86             | 0.95          | 0.88         |
> |                       | Overall         | 86.45         | 86.21   | 86.37            | 86.24           | **87.05** |
> | **Race**         | Min.            | 85.62         | 85.31     | 85.45            | 85.39                      | **86.23** |
> |                       | Gap             | 1.41          | 1.53    | 1.60             | 1.46           | **1.39**  |
> |                       | Overall         | 86.40         | 85.53       | 86.02            | 85.43      | **87.08** |
> | **Age**          | Min.            | 81.06         | 70.97       | 74.69            | 73.24      | **82.15** |
> |                       | Gap             | 5.32          | 6.99      | 5.87             | 6.08          | **5.10**  |
> |                       | Overall         | 86.26         | 85.06     | 86.25            | 85.37            | **86.35** |
> | **Intersection** | Min.            | 72.43         | 74.31      | 73.31            | 74.18       | **75.96** |
> |                       | Gap             | 18.92         | **11.59**   | 16.06   | 15.45    | 14.78 |

---

> > ### Author Response · Authors · 2022-11-16
> > **Response (2/2)**
> >
> > **Continuing Q1**
> >
> > The worst-performing subgroup is the “60-80 non-White Female”, which performs worse than any other subgroup if only considering one sensitive attribute at a time. But this subgroup does not contain the smallest number of images nor is it the most class-imbalanced subgroup. Also, although we observe a slight performance increase of the worst-case subgroup after resampling the minority subgroup, class, or both, there are also decreases in the overall performance. A possible reason is that the label noises in some subgroups are more severe than the other subgroup and the upsampled subgroups may contain more noisy labels to worsen the overall performance.
> >
> > We agree that a multi-factor analysis would be interesting. However, one difficulty is that the different datasets we use do not always contain the same set of sensitive attributes, which makes the number of samples too small to perform an effective regression study. We are happy to include it in the future when we have gradually added more datasets.
> >
> >
> > **Q2. Diagnosis labels.**
> >
> >  - We include a diverse set of diagnosis labels of various diseases in our benchmark. For chest X-ray datasets CheXpert and MIMIC-CXR, we use the label of "No Finding'' to summarize the absence of all 13 detailed pathology labels. For fundus image dataset PAPILA, we use the label of "glaucomatous'' and "non-glaucomatous''. For skin dermatology datasets HAM10000 and Fitzpatrick17k, we classify whether the lesions are "benign'' or "malignant''. For the eye SD-OCT dataset, the task is to predict if the patients have age-related macular degeneration (AMD). For CT dataset COVID-CT-MD, We design the task to predict whether the patients are infected by COVID-19. For the brain MRI dataset ADNI, the diagnosis label is whether the patient has Alzheimer’s disease (AD).
> > - Additionally, we have added another cardiology dataset OL3I into our framework. While the tasks of previously included datasets can be generally classified as diagnosis, we introduce a new prognosis task through this dataset, which is to predict whether the individual would be diagnosed with ischemic heart disease one year after the scan according to the labels provided. We believe this will introduce more diversity to our framework.
> >
> > We specify the concrete diagnosis label used for each dataset in Appendix B.1.2, and updated the results after incorporating the new datasets.
> >
> > We would love to hear from you if you have any further concerns. Thank you!

---

### Official Review · Reviewer_aeZu · 2022-10-25

**Confidence:** 4
**Correctness:** 4
**Technical Novelty And Significance:** 2
**Empirical Novelty And Significance:** 3
**Recommendation:** 8

**Clarity, Quality, Novelty And Reproducibility:**

The experiments have been described appropriately. Code and instructions were provided for proper reproducibility.

**Strength And Weaknesses:**

*Strengths
- The background on fairness research was deeply evaluated, not focusing only on medical papers but also on other fairness work applied to computer vision, which were included in the proposed framework.
- The proposed framework has the potential to standardize comparison among different debiasing strategies in medical imaging
- Aside from the contribution of the framework, the experiments and discussion highlight several key points and challenges for fairness awareness.
- The appendix provides an extensive and explained report of the experiments, metrics, hyperparameters, as well as instructions for adding new algorithms to the proposed framework.

*Weaknesses
- The proposed framework is still missing several fairness strategies such as the self-supervised ones, as well as datasets from other medical data modalities, and also does not include segmentation, which is an essential task in medical imaging.

**Summary Of The Paper:**

This paper introduces MEDFAIR, a framework to benchmark fairness in machine learning models for medical imaging. It provides a reproducible environment for developing and evaluating bias mitigation algorithms for deep learning applied to medical imaging. The authors use it to evaluate fairness in multiple scenarios and make recommendations for different medical cases.

**Summary Of The Review:**

The framework contribution for fairness in medical imaging is relevant and an active concern in the field. Experiments are extensive, detailed, and possibly reproducible. I am overall quite happy with this contribution as it may help other researchers in the field, all code is available and it is possible to add new algorithms, metrics, and datasets into the framework.

---

> ### Author Response · Authors · 2022-11-16
> **Response**
>
> Thank you very much for the supportive comments and valuable suggestions!
>
> **Q1. Exhaustiveness of coverage**
>
> We agree with you that our framework is not exhaustive as we identified in our discussion section. We will keep our codebase alive and gradually add more algorithms and datasets to it. To address your concern, we have added a new cardiology dataset OL3I into our framework about ischemic heart disease risk assessment with CT to further increase the diversity of the medical modalities, thus having increased our total number of datasets to 10. Detailed information about this dataset can be found in Table 1 and Appendix Table A5. We have updated the corresponding results in Figure 3-5 and Appendix Table A8, A9, and A14. From the experiment of this dataset, we have also observed bias between the male/female group and young/old groups. We have also re-run the statistical tests and our conclusions about both model selection and the effectiveness of bias mitigation algorithms remain unchanged, which further supports the reliability of our results.
>
> With regard to self-supervised approaches to improving fairness. We agree this is very interesting, and will extend the benchmark to include such algorithms in a future codebase update. However, this is non-possible to conduct the full experiments in the revision time-window, because of the difficulty and cost of conducting self-supervised pre-training of 3D CNNs for application to our 3D image datasets, as we need to keep the consistency of the SSL algorithm used for both 2D and 3D CNNs, as well as the same pretrained dataset as the supervised methods.
>
> We also agree that segmentation is an important problem in medical imaging. It is will be an important part of future iterations of our benchmark, but for now, we leave it for future work.
>
> We would love to hear from you if you have any further concerns. Thank you!

---

> > ### Comment · Reviewer_aeZu · 2022-11-29
> > **Regarding the authors' response**
> >
> > I thank you the authors for their response. I am fully satisfied with all their efforts.

---

### Author Response · Authors · 2022-11-16
**General Response from Authors**

Dear AC and reviewers,

We sincerely thank AC and all reviewers’ time and efforts in reviewing our paper. The constructive suggestions have helped us to improve our paper further. We appreciate that reviewers find the problem we study of significant importance and our experiments extensive.

We have conducted several additional experiments and updated to the manuscript according to the reviewers' comments and suggestions. Here is a summary of our updates:

- [Adding more algorithms and datasets] We have additionally added another cardiology dataset to further diversify the composition of MEDFAIR and conducted the experiments. Our conclusion from the experiments remains unchanged: no methods outperform ERM with statistical significance.
- [Analysis of the intersectional subgroups] We decompose the dataset into more fine-grained subgroups (e.g. 60-80 year old white female) and analyze the performance in Appendix E.
- [Analysis of the source of the bias] We use different resampling strategies to explicitly mitigate those sources of bias that can be observed from the dataset statistics (ie, data/class imbalance), and analyze the performance with ERM and SWAD in Appendix E.
- [Clarification of writing] We have further explained the details that reviewers are interested in or feel unclear about. The corresponding contents have been added or modified in our revised version.

We hope that the new experiments can address the concerns of the reviewers and that the modifications explain the unclarity. We are happy to answer any other additional questions and provide more information.

Sincerely,

Authors

---

### Decision · Program_Chairs · 2023-01-20

**Decision:**

Accept: notable-top-25%

**Justification For Why Not Higher Score:**

The paper introduces a framework and presents an extensive set of experimental evaluations, but the methodological contributions are modest.

**Justification For Why Not Lower Score:**

Authors addressed the concerns raised by the reviewers during the rebuttal period. They also conducted additional sets of experiments as requested.

**Metareview: Summary, Strengths And Weaknesses:**

This work introduces a framework to evaluate fairness of machine learning in medical imaging applications. This a timely topic and fills an existing gap. The reviewers emphasized that code and instructions provided ensured proper reproducibility. Empirical evaluation is extensive and conclusions valuable to the community.


**Note From Pc:**

if the above contains the word "oral" or "spotlight" please see: "oral" presentation means -> notable-top-5% and "spotlight" means -> notable-top-25%. As stated in our emails, we are disassociating presentation type from AC recommendations

**Summary Of Ac-Reviewer Meeting:**

N/A